# Parallel Bayesian Optimization of Multiple Noisy Objectives with Expected Hypervolume Improvement

**Samuel Daulton**
Facebook, University of Oxford
sdaulton@fb.com

**Maximilian Balandat**
Facebook
balandat@fb.com

**Eytan Bakshy**
Facebook
ebakshy@fb.com

## Abstract

Optimizing multiple competing black-box objectives is a challenging problem in many fields, including science, engineering, and machine learning. Multi-objective Bayesian optimization (MOBO) is a sample-efficient approach for identifying the optimal trade-offs between the objectives. However, many existing methods perform poorly when the observations are corrupted by noise. We propose a novel acquisition function, NEHVI, that overcomes this important practical limitation by applying a Bayesian treatment to the popular expected hypervolume improvement (EHVI) criterion and integrating over this uncertainty in the Pareto frontier. We argue that, even in the noiseless setting, generating multiple candidates in parallel is an incarnation of EHVI with uncertainty in the Pareto frontier and therefore can be addressed using the same underlying technique. Through this lens, we derive a natural parallel variant, $q$NEHVI, that reduces computational complexity of parallel EHVI from *exponential* to *polynomial* with respect to the batch size. $q$NEHVI is one-step Bayes-optimal for hypervolume maximization in both noisy and noiseless environments, and we show that it can be optimized effectively with gradient-based methods via sample average approximation. Empirically, we demonstrate not only that $q$NEHVI is substantially more robust to observation noise than existing MOBO approaches, but also that it achieves state-of-the-art optimization performance and competitive wall-times in large-batch environments.

## 1 Introduction

Black-box optimization problems that involve multiple competing noisy objectives are ubiquitous in science and engineering. For example, a real-time communications service may be interested in tuning the parameters of a control policy to adapt video quality in real time in order to maximize video quality and minimize latency [10, 17]. In robotics, scientists may seek to design hardware components that maximize locomotive speed and minimize energy expended [8, 38]. In agriculture, development agencies may seek to balance crop yield and environmental impact [28]. For such multi-objective optimization (MOO) problems, there typically is no single solution that is best with respect to all objectives. Rather, the goal is to identify the *Pareto frontier*: a set of optimal trade-offs such that improving one objective means deteriorating another. In many cases, the objectives are expensive to evaluate. For instance, randomized trials used in agriculture and the internet industry may take weeks or months to conduct and incur opportunity costs, and manufacturing and testing hardware is both costly and time-consuming. Therefore, it is imperative to be able to identify good trade-offs with as few objective evaluations as possible.

Bayesian optimization (BO), a method for efficient global black-box optimization, is often used to tackle such problems. BO employs a probabilistic surrogate model in conjunction with an acquisition function to navigate the trade-off between exploration (evaluating designs with high uncertainty) and exploitation (evaluating designs that are believed to be optimal). Although a significant number of works have explored multi-objective Bayesian optimization (MOBO), most available methods

[3, 39, 51, 60] do not take into account the fact that, in practice, observations are often subject to noise. For example, results of an A/B test are highly variable due to heterogeneity in the underlying user population and other factors. Agricultural trials are affected by the stochastic nature of plant growth and environmental factors such as soil composition or wind currents. In robotics, devices are subject to manufacturing tolerances, and observations of quantities such as locomotive speed and efficiency may be corrupted by measurement error from noisy sensors and environmental factors such as temperature or surface friction. While previous work has shown that a principled treatment of noisy observations can significantly improve optimization performance in the single-objective case [24, 37], this issue is understudied in the multi-objective setting. Furthermore, many applications in which evaluations take a long time require evaluating large *batches* of candidates in parallel in order to achieve reasonable throughput. For example, when firms optimize systems via A/B tests, it may take several weeks to test any particular configuration. Because of this, large batches of candidate policies are tested simultaneously [36]. In biochemistry and materials design, dozens of tests can be conducted parallel on a single microplate [63]. Even in sophisticated high throughput chemistry settings, these batches may take several hours or days to set up and evaluate [42]. Most existing MOBO methods, however, are either designed for purely sequential optimization [3, 51] or do not scale well to large batch sizes [11].

**Contributions:** In this work, we propose a novel MOBO algorithm, based on expected hypervolume improvement (EHVI), that scales to highly parallel evaluations of noisy objectives. Our approach is made possible by a general-purpose, differentiable, cached box decomposition (CBD) implementation that dramatically speeds up critical computations needed to account for uncertainty introduced by noisy observations and generate new candidate points for highly parallel batch or asynchronous evaluation. In particular, our CBD-based approach solves the fundamental problem of scaling parallel EHVI-based methods to large batch sizes, *reducing time and space complexity from exponential to polynomial*. Our proposed algorithm, *noisy* expected hypervolume improvement (NEHVI), is the one-step Bayes-optimal policy for hypervolume improvement and provides state-of-the-art performance across a variety of benchmarks. To our knowledge, our work provides the most extensive evaluation of noisy parallel MOBO to date. A high-quality implementation of $q$NEHVI, as well as many of the baselines considered here, will be made available as open-source software upon publication.

## 2   Preliminaries

Our goal is to find the set of optimal designs $\boldsymbol{x}$ over a bounded set $\mathcal{X} \subset \mathbb{R}^d$ that maximize one or more objectives $\boldsymbol{f}(\boldsymbol{x}) \in \mathbb{R}^M$, with no known analytical expression nor gradient information of $\boldsymbol{f}$.

**Multi-Objective Optimization (MOO)** aims to identify the set of *Pareto optimal* objective trade-offs. We say a solution $\boldsymbol{f}(\boldsymbol{x}) = \left[ f^{(1)}(\boldsymbol{x}), ..., f^{(M)}(\boldsymbol{x}) \right]$ *dominates* another solution $\boldsymbol{f}(\boldsymbol{x}) \succ \boldsymbol{f}(\boldsymbol{x}')$ if $f^{(m)}(\boldsymbol{x}) \geq f^{(m)}(\boldsymbol{x}')$ for $m = 1, ..., M$ and $\exists\, m \in \{1, ..., M\}$ s.t. $f^{(m)}(\boldsymbol{x}) > f^{(m)}(\boldsymbol{x}')$. We define the *Pareto frontier* as $\mathcal{P}^* = \{\boldsymbol{f}(\boldsymbol{x}) : \boldsymbol{x} \in \mathcal{X},\ \nexists\, \boldsymbol{x}' \in \mathcal{X}\ s.t.\ \boldsymbol{f}(\boldsymbol{x}') \succ \boldsymbol{f}(\boldsymbol{x})\}$, and denote the set of Pareto optimal designs as $\mathcal{X}^* = \{\boldsymbol{x} : \boldsymbol{f}(\boldsymbol{x}) \in \mathcal{P}^*\}$. Since the Pareto frontier (PF) is often an infinite set of points, MOO algorithms usually aim to identify a finite approximate PF $\mathcal{P}$. A natural measure of the quality of a PF is the hypervolume of the region of objective space that is dominated by the PF and bounded from below by a reference point. Provided with the approximate PF, the decision-maker can select a particular Pareto optimal trade-off according to their preferences.

**Bayesian Optimization (BO)** is a sample-efficient optimization method that leverages a probabilistic surrogate model to make principled decisions to balance exploration and exploitation [19, 50]. Typically, the surrogate is a Gaussian Process (GP), a flexible, non-parametric model known for its well-calibrated predictive uncertainty [47]. To decide which points to evaluate next, BO employs an acquisition function $\alpha(\cdot)$ that specifies the value of evaluating a set of new points $\boldsymbol{x}$ based on the surrogate's predictive distribution at . While evaluating the true black-box function $\boldsymbol{f}$ is time-consuming or costly, evaluating the surrogate is cheap and relatively fast; therefore, numerical optimization can be used to find the maximizer of the acquisition function $\boldsymbol{x}^* = \arg\max_{\boldsymbol{x} \in \mathcal{X}} \alpha(\boldsymbol{x})$ to evaluate next on the black-box function. BO sequentially selects new points to evaluate and updates the model to incorporate the new observations.

Evolutionary algorithms (EAs) such as NSGA-II [12] are a popular choice for solving MOO problems (see Zitzler et al. [67] for a review of various other approaches). However, EAs generally suffer from high sample complexity, rendering them infeasible for optimizing expensive-to-evaluate black-box

functions. Multi-objective Bayesian optimization (MOBO), which combines a Bayesian surrogate with an acquisition function designed for MOO, provides a much more sample-efficient alternative.

## 3 Related Work

Methods based on hypervolume improvement (HVI) seek to expand the volume of the objective space dominated by the Pareto frontier. Expected hypervolume improvement (EHVI) [16] is a natural extension of the popular expected improvement (EI) [29] acquisition function to the MOO setting. Recent work has led to efficient computational paradigms using box decomposition algorithms [59] and practical enhancements such as support for parallel candidate generation and gradient-based acquisition optimization [11, 58]. However, EHVI still suffers from some limitations, including (i) the assumption that observations are noise-free, and (ii) the exponential scaling of its batch variant, $q$EHVI, in the batch size $q$, which precludes large-batch optimization. DGEMO [39] is a recent method for parallel MOBO that greedily maximizes HVI while balancing the diversity of the design points being sampled. Although DGEMO scales well to large batch sizes, it does not account for noisy observations. TSEMO [5] is a Thompson sampling (TS) heuristic that can acquire batches of points by optimizing a random fourier feature (RFF) [46] approximation of a GP surrogate using NSGA-II and selecting a subset of points from the EA's population to sequentially greedily maximize HVI. This heuristic approach for maximizing HVI currently has no theoretical guarantees and relies on zeroth-order optimization methods, which tend to be slower and exhibit worse optimization performance than gradient-based approaches.

Entropy-based methods such as PESMO [25], MESMO [3], and PFES [51] are an alternative to EHVI. Of these three methods, PESMO is the only one that accounts for observation noise. However, PESMO involves intractable entropy computations and therefore relies on complex approximations, as well as challenging and time-consuming numerical optimization procedures [25]. Garrido-Merchán & Hernández-Lobato [21] recently proposed an extension to PESMO that supports parallel candidate generation. However, the authors of this work provide limited evaluation and have not provided code to reproduce their results.[1]

MOO can also be cast into a single-objective problem by applying a random scalarization of the objectives. ParEGO maximizes the expected improvement using random augmented Chebyshev scalarizations [32]. MOEA/D-EGO [64] extends ParEGO to the batch setting using multiple random scalarizations and the genetic algorithm MOEA/D [65] to optimize these scalarizations in parallel. Recently, $q$ParEGO, another batch variant of ParEGO was proposed that uses compositional Monte Carlo objectives and sequential greedy candidate selection [11]. Additionally, the authors proposed a noisy variant, $q$NParEGO, but the empirical evaluation of that variant was limited. TS-TCH [45] combines random Chebyshev scalarizations with Thompson sampling [54], which is naturally robust to noise when the objective is scalarized. Golovin & Zhang [23] propose to use a hypervolume scalarization with the property that the expected value of the scalarization over a specific distribution of weights is equivalent to the hypervolume indicator. The authors propose a upper confidence bound algorithm using randomly sampled weights, but provide a very limited empirical evaluation.

Many prior attempts by the simulation community to handle MOO with noisy observations found that accounting for the noise did not improve optimization performance: Horn et al. [26] suggest that the best approach is to ignore noise, and Koch et al. [33] concluded that further research was needed to determine if modeling techniques such as re-interpolation could improve BO performance with noisy observations. In contrast, we find that accounting for noise *does substantially* improve performance in noisy settings.

Lastly, previous works have considered methods for quantifying and monitoring uncertainty in the Pareto frontiers during the optimization [4, 7]. In contrast, we provide a solution to performing MOBO in noisy settings, rather than purely reasoning about the uncertainty in the Pareto frontier.

## 4 Background on Expected Hypervolume Improvement

In this section, we review hypervolume, hypervolume improvement, and expected hypervolume improvement as well as efficient methods for computing these metrics using box decompositions.

---

[1] We contacted the authors twice asking for code to reproduce their results, but they graciously declined.

**Definition 1.** *The hypervolume indicator (*HV*) of a finite approximate Pareto frontier $\mathcal{P}$ is the $M$-dimensional Lebesgue measure $\lambda_M$ of the space dominated by $\mathcal{P}$ and bounded from below by a reference point. $\mathbf{r} \in \mathbb{R}^M$: $\mathrm{HV}(\mathcal{P}|\mathbf{r}) = \lambda_M\left(\bigcup_{\mathbf{v} \in \mathcal{P}}[\mathbf{r}, \mathbf{v}]\right)$, where $[\mathbf{r}, \mathbf{v}]$ denotes the hyper-rectangle bounded by vertices $\mathbf{r}$ and $\mathbf{v}$.*

As in previous work, we assume that the reference point $\mathbf{r}$ is known and specified by the decision maker [58].

**Definition 2.** *The hypervolume improvement (*HVI*) of a set of points $\mathcal{P}'$ w.r.t. an existing approximate Pareto frontier $\mathcal{P}$ and reference point $\mathbf{r}$ is defined as*[2] $\mathrm{HVI}(\mathcal{P}'|\mathcal{P}, \mathbf{r}) = \mathrm{HV}(\mathcal{P} \cup \mathcal{P}'|\mathbf{r}) - \mathrm{HV}(\mathcal{P}|\mathbf{r})$.

Computing HV requires calculating the volume of a typically non-rectangular polytope and is known to have time complexity that is super-polynomial in the number of objectives [59]. An efficient approach for computing HV is to (i) decompose the region that is dominated by the Pareto frontier $\mathcal{P}$ and bounded from below by the reference point $\mathbf{r}$ into disjoint axis-aligned hyperrectangles [34], (ii) compute the volume of each hyperrectangle in the decomposition, and (iii) sum over all hyperrectangles. So-called box decomposition algorithms have also been applied to partition the region that is *not dominated* by the Pareto frontier $\mathcal{P}$, which can be used to compute the HVI from a set of new points [15, 59]. See Appendix B for further details.

**Expected Hypervolume Improvement:** Since function values at unobserved points are unknown in black-box optimization, so is the HVI of an out-of-sample point. However, in BO the probabilistic surrogate model provides a posterior distribution $p(\mathbf{f}(\mathbf{x})|\mathcal{D})$ over the function values for each $\mathbf{x}$, which can be used to compute the expected hypervolume improvement (EHVI) acquisition function: $\alpha_{\mathrm{EHVI}}(\mathbf{x}|\mathcal{P}) = \mathbb{E}\big[\mathrm{HVI}(\mathbf{f}(\mathbf{x})|\mathcal{P})\big]$. Although $\alpha_{\mathrm{EHVI}}$ can be expressed analytically when (i) the objectives are assumed to be conditionally independent given $\mathbf{x}$ and (ii) the candidates are generated and evaluated sequentially [58], Monte Carlo (MC) integration is commonly used since it does not require either assumption [16]. The more general parallel variant using MC integration is given by

$$\alpha_{q\mathrm{EHVI}}(\mathcal{X}_{\mathrm{cand}}|\mathcal{P}) \approx \hat{\alpha}_{q\mathrm{EHVI}}(\mathcal{X}_{\mathrm{cand}}|\mathcal{P}) = \frac{1}{N}\sum_{t=1}^{N}\mathrm{HVI}(\tilde{\mathbf{f}}_t(\mathcal{X}_{\mathrm{cand}})|\mathcal{P}), \tag{1}$$

where $\tilde{\mathbf{f}}_t \sim p(\mathbf{f}|\mathcal{D})$ for $t = 1, ..., N$ and $\mathcal{X}_{\mathrm{cand}} = \{x_i\}_{i=1}^q$ [11]. The same box decomposition algorithms used to compute HVI can be used to compute EHVI (either analytic or via MC) using piece-wise integration. EHVI computation is agnostic to the choice of box decomposition algorithm (and can also use approximate methods [9]). Similar to EI in the single-objective case, EHVI is a one-step Bayes-optimal algorithm for maximizing hypervolume in the MOO setting under the following assumptions: (i) only a single design will be generated and evaluated, (ii) the observations are noise-free, (iii) the final approximate Pareto frontier (and final design that will be deployed) will be drawn from the set of observed points [19].

## 5   Expected Hypervolume Improvement with Noisy Observations

We consider the case that frequently arises in practice where we only receive noisy observations $\mathbf{y}_i = \mathbf{f}(\mathbf{x}_i) + \boldsymbol{\epsilon}_i$, $\boldsymbol{\epsilon}_i \sim \mathcal{N}(0, \Sigma_i)$, where $\Sigma_i$ is the noise covariance. In this setting, EHVI is no longer (one-step) Bayes-optimal. This is because we can no longer compute the true Pareto frontier $\mathcal{P}_n = \{\mathbf{f}(\mathbf{x}) \mid \mathbf{x} \in X_n, \nexists \mathbf{x}' \in X_n \ s.t. \ \mathbf{f}(\mathbf{x}') \succ \mathbf{f}(\mathbf{x})\}$ over the previously evaluated points $X_n = \{\mathbf{x}_i\}_{i=1}^n$. Simply using the observed Pareto frontier, $\mathcal{Y}_n = \{\mathbf{y} \mid \mathbf{y} \in Y_n, \nexists \mathbf{y}' \in Y_n \ s.t. \ \mathbf{y}' \succ \mathbf{y}, y\}$ where $Y_n = \{\mathbf{y}_i\}_{i=1}^n$, can have strong detrimental effects on optimization performance. This is illustrated in Figure 1, which shows how EHVI is misled by noisy observations that appear to be Pareto optimal. EHVI proceeds to spend its evaluation budget trying to optimize noise, resulting in a clumped Pareto frontier that lacks diversity. Although the posterior mean could serve as a "plug-in" estimate of the true function values at the observed points and provide some regularization [61], we find that this heuristic also leads to clustered Pareto frontiers (EHVI-PM in Fig. 1). Similar patterns emerge with DGEMO (which does not account for noise), and other baselines that utilize the posterior mean rather than the observed values when computing hypervolume improvement (see Appendix H). To our knowledge, all previous work on EHVI assumes that observations are noiseless [16, 58] or imputes the unknown true function values with the posterior mean.

---

[2]For brevity we omit the reference point $\mathbf{r}$ when referring to HVI.

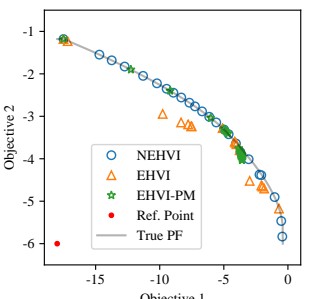

Figure 1: An illustration of the effect of noisy observations on the true noiseless Pareto frontiers identified by NEHVI (our proposed algorithm), EHVI, and EHVI-PM, which uses the modeled posterior mean as point estimate of the true in-sample function values. All algorithms are tested on a BraninCurrin synthetic problem, where observations are corrupted with zero-mean, additive Gaussian noise with a standard deviation of 5% of the range of respective objective. All methods use sequential ($q = 1$) optimization. See Appendix G for details.

## 5.1 A Bayes-optimal algorithm for hypervolume maximization in noisy environments

In contrast with EHVI(-PM), we instead approach the problem of hypervolume maximization under noisy observations from a Bayesian perspective and derive a novel one-step Bayes-optimal expected hypervolume improvement criterion that iterates the expectation over the posterior distribution $p(\boldsymbol{f}(X_n)|\mathcal{D}_n)$ of the function values at the previously evaluated points $X_n$ given noisy observations $\mathcal{D}_n = \{\boldsymbol{x}_i, \boldsymbol{y}_i, (\Sigma_i)\}_{i=1}^n$. Our acquisition function, *noisy expected hypervolume improvement* (NEHVI), is defined as

$$\alpha_{\text{NEHVI}}(\boldsymbol{x}) = \int \alpha_{\text{EHVI}}(\boldsymbol{x}|\mathcal{P}_n)p(\boldsymbol{f}|\mathcal{D}_n)d\boldsymbol{f} \qquad (2)$$

where $P_n$ denotes the Pareto frontier over $\boldsymbol{f}(X_n)$.

By integrating over the uncertainty in the function values at the observed points, NEHVI retains one-step Bayes-optimality in noisy environments (in noiseless environments, NEHVI is equivalent to EHVI). Empirically, Figure 1 shows that NEHVI is robust to noise and identifies a well-distributed Pareto frontier with no signs of clumping, even under very noisy observations.[3]

The integral in (2) is analytically intractable, but can easily be approximated using MC integration. Let $\tilde{\boldsymbol{f}}_t \sim p(\boldsymbol{f}|D_n)$ for $t = 1, ...N$ be samples from the posterior, and let $\mathcal{P}_t = \{\tilde{\boldsymbol{f}}_t(\boldsymbol{x}) \mid \boldsymbol{x} \in X_n, \tilde{\boldsymbol{f}}_t(\boldsymbol{x}) \succ \tilde{\boldsymbol{f}}_t(\boldsymbol{x}') \forall \boldsymbol{x}' \in X_n\}$ be the Pareto frontier over the previously evaluated points under the sampled function $\tilde{\boldsymbol{f}}_t$. Then, $\alpha_{\text{NEHVI}}(\boldsymbol{x}) \approx \frac{1}{N}\sum_{t=1}^N \alpha_{\text{EHVI}}(\boldsymbol{x}|\mathcal{P}_t)$. Using MC integration, we can compute the inner expectation in $\alpha_{\text{EHVI}}$ simultaneously using samples from the joint posterior $\tilde{\boldsymbol{f}}_t(X_n, \boldsymbol{x}) \sim p(\boldsymbol{f}(X_n, \boldsymbol{x})|\mathcal{D}_n)$ over $\boldsymbol{x}$ and $X_n$:

$$\hat{\alpha}_{\text{NEHVI}}(\boldsymbol{x}) = \frac{1}{N}\sum_{t=1}^N \text{HVI}(\tilde{\boldsymbol{f}}_t(\boldsymbol{x})|\mathcal{P}_t). \qquad (3)$$

See Appendix B for details on computing (3) using box decompositions. Note that this "full-MC" variant of NEHVI does not require objectives to be modeled independently, and supports multi-task covariance functions across correlated objectives.

## 5.2 Parallel Noisy Expected Hypervolume Improvement

Generating and evaluating batches of candidates is imperative to achieving adequate throughput in many real-world scenarios. qNEHVI can naturally be extended to the parallel (asynchronous or batch) setting by evaluating HVI with respect to a batch of $q$ points $\mathcal{X}_{\text{cand}} = \{\boldsymbol{x}_i\}_{i=1}^q$

$$\alpha_{q\text{NEHVI}}(\mathcal{X}_{\text{cand}}) = \int \alpha_{q\text{EHVI}}(\mathcal{X}_{\text{cand}}|\mathcal{P}_n)p(\boldsymbol{f}|\mathcal{D}_n)d\boldsymbol{f} \approx \hat{\alpha}_{q\text{NEHVI}}(\mathcal{X}_{\text{cand}}) = \frac{1}{N}\sum_{t=1}^N \text{HVI}(\tilde{\boldsymbol{f}}_t(\mathcal{X}_{\text{cand}})|\mathcal{P}_t)$$
$$(4)$$

Since optimizing $q$ candidates jointly is a difficult numerical optimization problem over a $qd$-dimensional domain, we use a sequential greedy approximation in the parallel setting and solve a sequence of $q$ simpler optimization problems with $d$ dimensions, which been shown empirically to improve optimization performance [57]. While selecting candidates according to a "sequential greedy" policy does not guarantee that the selected batch of candidates is a maximizer of the $\alpha_{q\text{NEHVI}}$, the submodularity of $\alpha_{q\text{NEHVI}}$ allows us to *bound the regret* of this approximation to be no more than $\frac{1}{e}\alpha_{q\text{NEHVI}}^*$, where $\alpha_{q\text{NEHVI}}^* = \max_{\mathcal{X}_{\text{cand}} \in \mathcal{X}} \alpha_{q\text{NEHVI}}(\mathcal{X}_{\text{cand}})$ (see Appendix F).

---

[3]This noise level is 5x greater than the ones considered by previous works that evaluate noisy MOBO [25].

# 6 Efficient Evaluation with Cached Box Decompositions

Although $\hat{\alpha}_{\text{NEHVI}}(\boldsymbol{x})$ in (3) has a concise mathematical form, computing it requires determining the Pareto frontier $\mathcal{P}_t$ under each sample $\tilde{\boldsymbol{f}}_t$ for $t = 1, ..., N$ and then partitioning the region that is not dominated by $\mathcal{P}_t$ into disjoint hyperrectangles $\{S_{k_t}\}_{k_t=1}^{K_t}$. Optimizing the unbiased MC estimator of $\alpha_{\text{NEHVI}}$ would require re-sampling $\{\tilde{\boldsymbol{f}}_t\}_{t=1}^N$ at each evaluation of $\alpha_{\text{NEHVI}}$. However, computing the Pareto frontier and performing a box decomposition under each of the $N$ samples during every evaluation of $\alpha_{\text{NEHVI}}$ in the inner optimization loop ($\boldsymbol{x}^* = \arg\max_{\boldsymbol{x}} \alpha_{\text{NEHVI}}(\boldsymbol{x}|\mathcal{D}_n)$) would be prohibitively expensive. This is because box decomposition algorithms have super-polynomial time complexity in the number of objectives [59]. We instead propose an efficient alternative computational technique for repeated evaluations of EHVI with uncertain Pareto frontiers.

**Cached Box Decompositions:** For repeated evaluations of the integral in (2), we use a set of fixed samples $\{\tilde{\boldsymbol{f}}_t(X_n)\}_{t=1}^N$, which allows us to compute the Pareto frontiers and box decompositions once, and cache them for the entirety of the acquisition function optimization, thereby making those two computationally intensive operations a one-time cost per BO iteration.[4] We refer to this approach as using *cached box decompositions* (CBD). The method of optimizing over fixed random samples is known as sample average approximation (SAA) [2].

**Conditional Posterior Sampling:** Under the CBD formulation, computing $\hat{\alpha}_{\text{NEHVI}}(\boldsymbol{x})$ with joint samples from $\tilde{\boldsymbol{f}}_t(X_n, \boldsymbol{x}) \sim p(\boldsymbol{f}(X_n, \boldsymbol{x})|\mathcal{D}_n)$ requires sampling from the conditional distributions

$$\tilde{\boldsymbol{f}}_t(\boldsymbol{x}) \sim p\big(\boldsymbol{f}(\boldsymbol{x})|\boldsymbol{f}(X_n) = \tilde{\boldsymbol{f}}_t(X_n), \mathcal{D}_n\big), \tag{5}$$

where $t = 1, ..., N$ and $\{\tilde{\boldsymbol{f}}_t(X_n)\}_{t=1}^N$ are the realized samples at the previously evaluated points. For multivariate Gaussian posteriors (as is the case with GP surrogates), we can sample from $p(\boldsymbol{f}(X_n)|\mathcal{D}_n)$ via the reparameterization trick [30] by evaluating $\tilde{\boldsymbol{f}}_t(\boldsymbol{x}) = \boldsymbol{\mu}_n + L_n^T \boldsymbol{\zeta}_{n,t}$, where $\boldsymbol{\zeta}_{n,t} \sim \mathcal{N}(\boldsymbol{0}, I_{nM})$, $\boldsymbol{\mu}_n \in \mathbb{R}^{nM}$ is the posterior mean, and $L_n \in \mathbb{R}^{nM \times nM}$ is a lower triangular root decomposition of the posterior covariance matrix, typically a Cholesky decomposition. Given $L_n$, we can obtain a root decomposition $L_n'$ of the covariance matrix of the joint posterior $p(\boldsymbol{f}(X_n, \boldsymbol{x})|\mathcal{D}_n)$ by performing efficient low-rank updates [44]. Given $L_n'$ and the posterior mean of $p(\boldsymbol{f}(X_n, \boldsymbol{x})|\mathcal{D}_n)$, we can sample from (5) via the reparameterization trick by augmenting the existing base samples $\boldsymbol{\zeta}_{n,t}$ with $M$ new base samples for the new point.

## 6.1 Efficient Sequential Greedy Batch Selection using CBD

The CBD technique addresses the general problem of inefficient repeated evaluations of EHVI with uncertain Pareto frontiers. In this section, we show that sequential greedy batch selection (with both $q$EHVI and $q$NEHVI) is an incarnation of EHVI with uncertain Pareto frontiers.

The original formulation of parallel EHVI in Daulton et al. [11] uses the inclusion-exclusion principle (IEP), which involves computing the volume jointly dominated by each of the $2^q - 1$ nonempty subsets of points in $\mathcal{X}_{\text{cand}}$. However, using large batch sizes is *not computationally feasible* under this formulation because time and space complexity are exponential in $q$ and multiplicative in the number of hyperrectangles in the box decomposition [11] (see Appendix D for a complexity analysis). Although $q$EHVI is optimized using sequential greedy batch selection, the IEP is used over all candidates $\boldsymbol{x}_1, ..., \boldsymbol{x}_i$ when selecting candidate $i$. Although the IEP could similarly be used to compute $q$NEHVI, we instead leverage CBD, which yields a sequential greedy approximation of the joint (noisy) EHVI that is *mathematically equivalent* to the IEP formulation, but *significantly reduces computational overhead*. That is, the IEP and CBD approaches produce exactly the same acquisition value for a given set of points $\mathcal{X}_{\text{cand}}$, but the IEP and the CBD approaches have *exponential* and *polynomial* time complexities in $q$, respectively.

When selecting $\boldsymbol{x}_i$ for $i \in \{2, \ldots, q\}$, all $\boldsymbol{x}_j$ for which $j < i$ have already been selected and are therefore held constant. Thus, we can decompose $q$NEHVI into the $q$NEHVI from the previously selected candidates $\boldsymbol{x}_1, \ldots, \boldsymbol{x}_{i-1}$ and NEHVI from $\boldsymbol{x}_i$ given the previously selected candidates

---

[4]For greater efficiency, we may also prune $X_n$ to remove points that are dominated with high probability, which we estimate via MC.

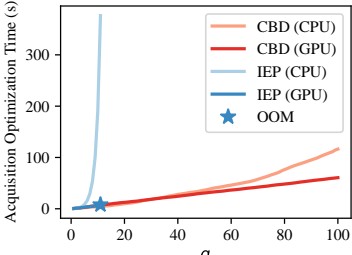

Figure 2: Acquisition optimization wall time under a sequential greedy approximation using L-BFGS-B. CBD enables scaling to much larger batch sizes $q$ than using the IEP and avoids running out-of-memory (OOM) on a GPU. Independent GPs are used for each outcome. The Pareto frontier of of the 2-objective, 6-dimensional DTLZ2 problem [13] is initialized with 20 points. Wall times were measured on a Tesla V100 SXM2 GPU (16GB RAM) and a 2x Intel Xeon 6138 CPU @ 2GHz (251GB RAM). See Appendix H.2 for results with more objectives.

$$\hat{\alpha}_{q\mathrm{NEHVI}}(\{\boldsymbol{x}_j\}_{j=1}^i) = \frac{1}{N}\sum_{t=1}^N \mathrm{HVI}\big(\{\tilde{\boldsymbol{f}}_t(\boldsymbol{x}_j)\}_{j=1}^{i-1}\,|\,\mathcal{P}_t\big) + \frac{1}{N}\sum_{t=1}^N \mathrm{HVI}\big(\tilde{\boldsymbol{f}}_t(\boldsymbol{x}_i)\,|\,\mathcal{P}_t \cup \{\tilde{\boldsymbol{f}}_t(\boldsymbol{x}_j)\}_{j=1}^{i-1}\big)$$

$$(6)$$

Note that the first term on the right hand side is constant, since $\{\boldsymbol{x}_j\}_{j=1}^{i-1}$ and $\{\tilde{\boldsymbol{f}}_t(\boldsymbol{x}_j)\}_{j=1}^{i-1}$ are fixed for all $t = 1, ..., N$. The second term is $\hat{\alpha}_{\mathrm{NEHVI}}(\boldsymbol{x}_i)$, where the NEHVI is taken with respect to the Pareto frontier across $\boldsymbol{f}(X_n, \boldsymbol{x}_1, ..., \boldsymbol{x}_{i-1})$ and computed using the fixed samples $\{\tilde{\boldsymbol{f}}_t(X_n, \boldsymbol{x}_1, ...\boldsymbol{x}_{i-1})\}_{t=1}^N$. To compute the second term when selecting candidate $\boldsymbol{x}_i$, the $N$ Pareto frontiers and CBDs are updated to include $\{\tilde{\boldsymbol{f}}_t(X_n, \boldsymbol{x}_1, ...\boldsymbol{x}_{i-1})\}_{t=1}^N$. As in the sequential $q = 1$ setting, the box decompositions are only computed and cached while selecting each candidate point. See Appendix C.2 for a derivation of (6). Although we have focused on $q$NEHVI in the above, the CBD formulation for $q$EHVI is obtained by simply replacing $\mathcal{P}_t$ with the Pareto frontier over the observed values $\mathcal{Y}_n$.

Despite computing $N$ box decompositions when selecting each candidate $\boldsymbol{x}_i$ for $i = 2, ..., q$, the CBD approach reduces the time and space complexity from exponential (under the IEP) to polynomial in $q$ (see Appendix D for details on time and space complexity). Figure 2 shows the total acquisition optimization time (including box decompositions) for various batch sizes and demonstrates that using CBD allows to scale to batch sizes that are *completely infeasible* when using IEP.

## 7 Optimizing NEHVI

**Differentiability**: Importantly, $\hat{\alpha}_{\mathrm{NEHVI}}(\boldsymbol{x})$ is differentiable w.r.t. $\boldsymbol{x}$. Although determining the Pareto frontier and computing the box decompositions are non-differentiable operations, these operations do not involve $\boldsymbol{x}$, even when re-sampling from the joint posterior $p(\boldsymbol{f}(X_n, \boldsymbol{x})|\mathcal{D}_n)$. Exact sample-path gradients of $\nabla_{\boldsymbol{x}}\hat{\alpha}_{\mathrm{NEHVI}}(\boldsymbol{x})$ can easily be computed using auto-differentiation in modern computational frameworks. This enables efficient gradient-based optimization of $q$NEHVI.[5]

**SAA Convergence Results:** In addition to approximating the outer expectation over $\boldsymbol{f}(X_n)$ with fixed posterior samples, we can similarly fix the base samples used for the new candidate point $\boldsymbol{x}$. This approach yields a deterministic acquisition function, which enables using (quasi-) higher-order optimization methods to obtain fast convergence rates for acquisition optimization [2]. Importantly, we prove that the theoretical convergence guarantees on acquisition optimization under the SAA approach proposed by Balandat et al. [2] also hold for NEHVI.

**Theorem 1.** *Suppose $\mathcal{X}$ is compact and $\boldsymbol{f}$ has a multi-output GP prior with continuously differentiable mean and covariance functions. Let $X_n = \{\boldsymbol{x}_i\}_{i=1}^n$ denote the previously evaluated points and $\{\boldsymbol{\zeta}\}_{t=1}^N$ be base samples $\boldsymbol{\zeta} \sim \mathcal{N}(\boldsymbol{0}, I_{(n+1)M})$. Let $\hat{\alpha}_{\mathrm{NEHVI}}$ denote the deterministic acquisition function computed using $\{\boldsymbol{\zeta}\}_{t=1}^N$ as $\hat{\alpha}_{\mathrm{NEHVI}}^N$ and define $S^* := \arg\max_{\boldsymbol{x}\in\mathcal{X}}\alpha_{\mathrm{NEHVI}}(\boldsymbol{x})$ to be the set of maximizers of $\alpha_{\mathrm{NEHVI}}(\boldsymbol{x})$ over $\mathcal{X}$. Suppose $\hat{\boldsymbol{x}}_N^* \in \arg\max_{\boldsymbol{x}\in\mathcal{X}}\hat{\alpha}_{\mathrm{NEHVI}}^N(\boldsymbol{x})$. Then (1) $\hat{\alpha}_{\mathrm{NEHVI}}^N(\hat{\boldsymbol{x}}_N^*) \to \alpha_{\mathrm{NEHVI}}(\boldsymbol{x}_N^*)$ almost surely, and (2) $\mathrm{dist}(\hat{\boldsymbol{x}}_N^*, S^*) \to 0$, where $\mathrm{dist}(\hat{\boldsymbol{x}}_N^*, \mathcal{S}^*) := \inf_{\boldsymbol{x}\in S^*}||\hat{\boldsymbol{x}}_N^* - \boldsymbol{x}||$ is the Euclidean distance between $\hat{\boldsymbol{x}}_N^*$ and the set $S^*$.*

Theorem 1 also holds in the parallel setting, so $q$NEHVI enjoys the same convergence guarantees as NEHVI on acquisition optimization under the SAA. See Appendix E for further details and proof.

---

[5]One can also show that the gradient of the full MC estimator $\hat{\alpha}_{q\mathrm{NEHVI}}$ is an unbiased estimator of the gradient of the true joint noisy expected hypervolume improvement $\alpha_{q\mathrm{NEHVI}}$. However, this result is not necessary for our SAA approach.

# 8 Approximation of $q$NEHVI using Approximate GP Sample Paths

Although CBD yields polynomial complexity of $q$NEHVI with respect to $q$ (rather than exponential complexity with the IEP), it still requires computing $N$ box decompositions and repeatedly evaluating the joint posterior over $\boldsymbol{f}(X_n, \{\boldsymbol{x}_j\}_{j=1}^{i-1})$ for selecting each candidate $\boldsymbol{x}_i$ for $i = 1, ..., q$. A cheaper alternative is to approximate the integral in (4) using a single approximate GP sample path $\tilde{\boldsymbol{f}}_i$ using RFFs when optimizing candidate $\boldsymbol{x}_i$. A single-sample approximation of $q$NEHVI, which we refer to as $q$NEHVI-1, can be computed by using $\tilde{\boldsymbol{f}}_i$ as the sampled GP in (6). Since the RFF is a deterministic model, it is much less computationally expensive to evaluate than the GP posterior on out-of-sample points, and exact gradients of $q$NEHVI-1 with respect to current candidate $\boldsymbol{x}_i$ can be computed and used for efficient multi-start optimization of $q$NEHVI-1 using second-order gradient methods. $q$NEHVI-1 requires CBD for efficient sequential greedy batch selection and gradient-based optimization, but does not use a sample average approximation for optimizing a new candidate $\boldsymbol{x}_i$; instead, it uses an approximate *sample path*. See Rahimi & Recht [46] for details on RFFs.

$q$NEHVI-1 is related to TSEMO in that both use sequential greedy batch selection using HVI based on RFF samples. However, TSEMO does not directly maximize HVI when selecting candidate $\boldsymbol{x}_i$, where $i = 1, ..., q$; rather, it relies on a heuristic approach of running NSGA-II on an RFF sample of each objective to create a discrete population of candidates and then selecting the point from the discrete population that maximizes HVI under the RFF sample. In contrast, $q$NEHVI-1 *directly optimizes* HVI under the RFF using exact sample-path gradients, which leads to improved optimization performance (see Appendix H). Furthermore, we find that $q$NEHVI-1 is significantly faster than TSEMO, because rather than using NSGA-II it uses second order gradient methods to optimize HVI (see Appendix H). Gradient-based optimization is only possible because CBD enables scalable, differentiable HVI computation. While the primary goal of this work is to develop a principled, scalable method for parallel EHVI in noisy environments, we include empirical comparisons with $q$NEHVI-1 throughout the appendix to demonstrate the generalizablility of the CBD approach and practical performance of the $q$NEHVI-1 approximation. $q$NEHVI-1 achieves the fastest batch selection timesof any method tested on a GPU on every problem; in many cases, this is an order of magnitude speed-up over $q$NEHVI. Moreover, $q$NEHVI-1 has a remarkable ability to scale to large batch sizes when the dimensionality of optimization problem is modest. Further investigation of $q$NEHVI-1 is needed, but we hope that the readers can recognize the ways in which $q$NEHVI can create broader opportunities for research into hypervolume improvement based acquisition functions.

# 9 Experiments

We empirically evaluate $q$NEHVI on a set of synthetic and real-world benchmark problems. We compare it against the following recently proposed methods from the literature: PESMO, MESMO (which we extend to the handle noisy observations using the noisy information gain from Takeno et al. [52]), PFES, DGEMO, MOEA/D-EGO, TSEMO, TS-TCH, $q$EHVI (and $q$EHVI-PM-CBD, which uses the posterior mean as a plug-in estimate for the function values at the in-sample points, along with CBD to scale to large batch sizes), and qNParEGO. We optimize all methods using multi-start L-BFGS-B with exact gradients (except for PFES, which uses gradients approximated via finite differences), including TS-TCH where we optimize approximate function samples using RFFs with 500 basis functions. We model each outcome with an independent GP with a Matérn 5/2 ARD kernel and infer the GP hyperparameters via maximum a posteriori (MAP) estimation. For all problems, we assume that the noise variances are observed (except ABR, where we infer the noise level). See Appendix G for more details on the experiments and acquisition function implementations.

We evaluate all methods using the logarithm of the difference in hypervolume between the true Pareto frontier and the approximate Pareto frontier recovered by the algorithm. Since evaluations are noisy, we compute the hypervolume dominated by the noiseless Pareto frontier across the observed points for each method.

**Synthetic Problems:** We consider a noisy variants of the *BraninCurrin* problem ($M = 2, d = 2$) and the *DTLZ2* problem ($M = 2, d = 6$) [13], in which observations are corrupted with zero-mean additive Gaussian noise with standard deviation of 5% of the range of each objective for *BraninCurrin* and 10% for *DTLZ2*.

**Adaptive Bitrate (ABR) Control Policy Optimization:** ABR controllers are used for real-time communication and media streaming applications. Policies for these controllers must be tuned to

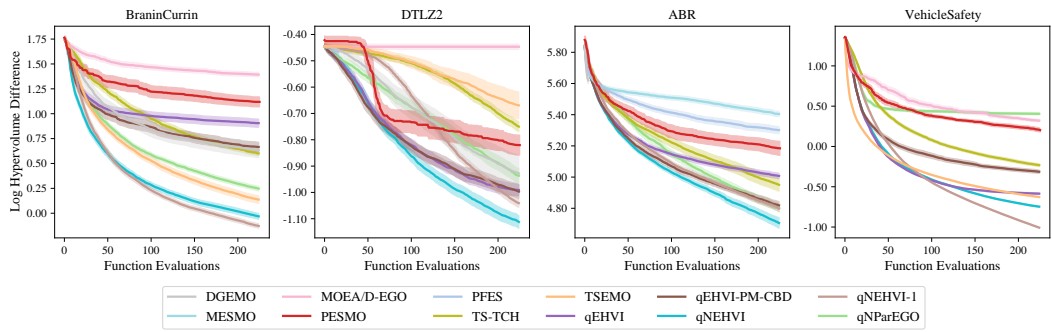

Figure 3: Sequential optimization performance. The shaded region indicates two standard errors of the mean over 100 replications (only 20 replications were feasible for PESMO due to large runtimes).

deliver a high quality of experience with respect to multiple objectives [40]. In industry settings, A/B tests with dozens of policies are tested simultaneously since each policy may take days or weeks to evaluate, producing noisy measurements across multiple objectives. In this experiment, we tune policies to maximize video quality (bitrate) and minimize stall time. The policy has $d = 4$ parameters, which are detailed in Appendix G. We use the Park simulator [41] and sample a random set of 100 traces to obtain noisy measurements of the objectives under a given policy. For comparing the performance of different methods, we estimate the true noiseless objective using mean objectives across 300 traces. We infer a homoskedastic noise level jointly with the GP hyperparameters via MAP estimation.

**Vehicle Design Optimization:** Optimizing the design of the frame an automobile is important to maximizing passenger safety, vehicle durability and fuel efficiency. Evaluating a vehicle design is time-consuming, since either a vehicle must manufactured and crashed, or a nonlinear finite element-based crash analysis must be run to simulate a collision (which can take over 20 hours per run) [62]. Hence, evaluating many designs in parallel is critical for reducing end-to-end optimization time. Observations are often noisy due to manufacturing imperfections, measurement error, or non-deterministic simulations. In this experiment, we tune the $d = 5$ widths of various components of a vehicle's frame to minimize proxy metrics for (1) fuel consumption, (2) passenger trauma in a full frontal collison, and (3) vehicle fragility [53]. See Appendix G for details. For this demonstration, we add zero-mean Gaussian noise with a standard deviation of 1% of the objective range, which roughly corresponds to the manufacturing noise level used in previous work [62].

### 9.1 Summary of Results:

We find that $q$NEHVI and $q$NEHVI-1 outperform all other methods on the noisy benchmarks, both in the sequential and parallel setting. In the sequential setting (Fig 3), $q$NEHVI and $q$NEHVI-1 are followed closely by $q$EHVI-PM, and in some cases, even $q$EHVI. TS-TCH is firmly in the middle of the pack, while information-theoretic acquisition functions appear to perform the worst. This is consistent across noise levels; for experiments where we add noise to the objectives, we consider noise levels ranging from 1% to 10% of the range of each objective (these are magnitudes of the noise often seen in practice). Previous works have only evaluated MOBO algorithms with noise levels of 1% [25]. In Appendix H, we perform a study showing that $q$NEHVI consistently performs best with increasing noise levels up to 30% of the range of each objective.

While parallel evaluation can provide optimization speedups on order of the batch size $q$, these evaluations do affect the overall sample complexity of the algorithm, since less information is available within the synchronous batch setting compared with fully sequential optimization. We find that, by and large, $q$NEHVI achieves the greatest hyper-volume for increasingly large batch sizes, and scales more elegantly relative to TS-TCH and the ParEGO variants (Fig 4). $q$NEHVI also consistently outperforms $q$EHVI-PM-CBD. In Appendix H, we observe that $q$NEHVI and $q$NEHVI-1 provides excellent anytime performance all values of $q$ that we tested. We provide results on 4 additional test problems in Appendix H.3, and in Appendix H.8, we demonstrate that leveraging CBD and a single sample path approximation, $q$NEHVI-1 enables scaling to 5-objective problems, which is a first for an HVI-based method, to our knowledge.

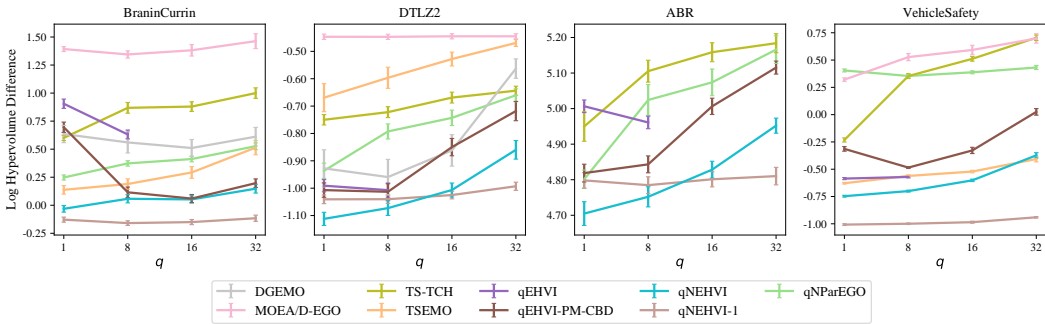

Figure 4: The quality of the final Pareto frontier identified by each method with increasing batch sizes $q$ given a budget of 224 function evaluations. $q$EHVI is only included for $q = 1$ and $q = 8$ because the IEP scales exponential with $q$. DGEMO is omitted on the ABR problem because it was prohibitively slow with time-consuming ABR simulations and on the VehicleSafety problem because DGEMO consistently crashed in the graph cutting algorithm.

In our experiments, we find that $q$NEHVI-1 is among the top performers on relatively low-dimensional problems. Given the strong performance of $q$NEHVI-1, we examine its performance as the dimensionality of the search space increases in Appendix H.5. We find that $q$NEHVI is more robust than $q$NEHVI-1 in higher-dimensional search spaces, but further investigation is needed into how the number of the Fourier basis functions affects the performance of $q$NEHVI-1 in high-dimensional search spaces.

**Optimization wall time:** Across all experiments, we observe competitive wall times for optimizing $q$NEHVI and $q$NEHVI-1 (all wall time comparisons are provided in Appendix H). On a GPU, optimizing $q$NEHVI-1 *incurs the lowest wall time of any method that we tested on every single problem* and optimizing $q$NEHVI is faster than optimizing information-theoretic methods on all problems. Using efficient low-rank Cholesky updates, $q$NEHVI is often faster than the $q$NParEGO implementation in BoTorch on a GPU.

## 10 Discussion

We proposed NEHVI, a novel acquisition function that provides a principled approach to parallel and noisy multi-objective Bayesian optimization. NEHVI is a one-step Bayes-optimal policy for maximizing the hypervolume dominated by the Pareto frontier in noisy and noise-free settings. NEHVI is made feasible by a new approach to computing joint hypervolumes (CBD), and we demonstrated that CBD enables scalable, parallel candidate generation with both noiseless $q$EHVI and $q$NEHVI. We provide theoretical results on optimizing a MC estimator of $q$NEHVI using sample average approximation and demonstrate significant improvements in optimization performance over state-of-the-art MOBO algorithms.

Yet, our work has some limitations. While the information-theoretic acquisition functions tested here perform poorly on our benchmarks, they do allow for decoupled evaluations of different objectives in cases where querying one objective may be more resource-intensive than querying other objectives. Optimizing such acquisition functions is a non-trivial task, and it is possible that with improved procedures, such acquisition functions could yield improved performance and provide a principled approach to selecting evaluation sources on a budget. Although practically fast enough for most Bayesian optimization tasks, exact hypervolume computation has super-polynomial complexity in the number of objectives. Combining $q$NEHVI with differentiable approximate methods for computing hypervolume (e.g. Couckuyt et al. [9], Golovin & Zhang [23]) could lead to further speed-ups.

We hope that the core ideas presented in this work, including the CBD approach, can provide a framework to support the development of new computationally efficient MOBO methods.

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
