# Appendix to:

# Parallel Bayesian Optimization of Multiple Noisy Objectives with Expected Hypervolume Improvement

## A  Potential Societal Impact

Bayesian Optimization specifically aims to increase sample efficiency for hard optimization algorithms, and consequently can help achieve better solutions without incurring large societal costs. For instance, as demonstrated in this work, automotive design problems may be solved much faster, reducing the amount of computationally costly simulations and thus the energy footprint during development. At the same time, improved solutions mean that high crash safety can be achieved with lighter cars, resulting in fewer resources required for their production and, importantly, improving fuel economy of the whole vehicle fleet. Increased robustness to noisy observations further helps reduce the resources spent on evaluating regions of the search space that are not promising. Improvements to the optimization performance and practicality of multi-objective Bayesian optimization have the potential to allow decision makers to better understand and make more informed decisions across multiple trade-offs. We expect these directions to be particularly important as Bayesian optimization is increasingly used for applications such as recommender systems [35], where auxiliary goals such as fairness must be accounted for. Of course, at the end of the day, exactly what objectives decision makers choose to optimize, and how they balance those trade-offs (and whether that is done in equitable fashion) is up to the individuals themselves.

## B  Computing Hypervolume Improvement with Box Decompositions

**Definition 3.** *For a set of objective vectors $\{\boldsymbol{f}(\boldsymbol{x}_i)\}_{i=1}^q$, a reference point $\boldsymbol{r} \in \mathbb{R}^M$, and a Pareto frontier $\mathcal{P}$, let $\Delta(\{\boldsymbol{f}(\boldsymbol{x}_i)\}_{i=1}^q, \mathcal{P}, \boldsymbol{r}) \subset \mathbb{R}^M$ denote the set of points (1) that are dominated by $\{\boldsymbol{f}(\boldsymbol{x}_i)\}_{i=1}^q$, (2) that dominate $\boldsymbol{r}$, and (3) that are not dominated by $\mathcal{P}$.*

Let $\{S_1, ..., S_K\}$ be a set of $K$ disjoint axis-aligned rectangles where each $S_k$ is defined by a pair of lower and upper vertices $\boldsymbol{l}_k \in \mathbb{R}^M$ and $\boldsymbol{u}_k \in \mathbb{R}^M \cup \{\infty\}$. Figure 5 shows an example decomposition. Such a partitioning allows for efficient piece-wise computation of the hypervolume improvement from a new point $\boldsymbol{f}(\boldsymbol{x}_i)$ by computing the volume of the intersection of the region dominated exclusively by the new point with $\Delta(\{f(\boldsymbol{x}_i), \mathcal{P}, \boldsymbol{r})$ (and not dominated by the $P$) with each hyperrectangle $S_k$. Although $\Delta(\boldsymbol{f}(\boldsymbol{x}_i), \mathcal{P}, \boldsymbol{r})$ is a non-rectangular polytope, the intersection of $\Delta(\boldsymbol{f}(\boldsymbol{x}_i), \mathcal{P}, \boldsymbol{r})$ with each rectangle $S_k$ is a rectangular polytope and the vertices bounding the hyperrectangle corresponding to $\Delta(\boldsymbol{f}(\boldsymbol{x}_i), \mathcal{P}, \boldsymbol{r}) \cap S_k$ can be easily computed: the lower bound vertex is $\boldsymbol{l}_k$ and the upper bound vertex is the component-wise minimum of $\boldsymbol{u}_k$ and the new point $\boldsymbol{f}(\boldsymbol{x})$: $\boldsymbol{z}_k := \min \left[ \boldsymbol{u}_k, \boldsymbol{f}(\boldsymbol{x}) \right]$. The hypervolume improvement can be computed by summing over the volume of $\Delta(\boldsymbol{f}(\boldsymbol{x}_i), \mathcal{P}, \boldsymbol{r}) \cap S_k$ over all $S_k$

$$\mathrm{HVI}\big(\boldsymbol{f}(\boldsymbol{x}), \mathcal{P}\big) = \sum_{k=1}^{K} \mathrm{HVI}_k\big(\boldsymbol{f}(\boldsymbol{x}), \boldsymbol{l}_k, \boldsymbol{u}_k\big) = \sum_{k=1}^{K} \prod_{m=1}^{M} \big[ z_k^{(m)} - l_k^{(m)} \big]_+, \qquad (7)$$

where $[\cdot]_+$ denotes the $\max(\cdot, 0)$ operation.

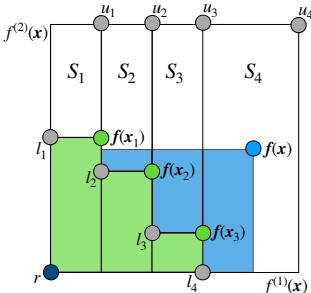

Figure 5: The hypervolume improvement from a new point $\boldsymbol{f}(\boldsymbol{x})$ is shown in blue. The current Pareto frontier $\mathcal{P}$ is given by the green points, the green area is the hypervolume of the Pareto frontier $\mathcal{P}$ given reference point $\boldsymbol{r}$. The white rectangles $S_1, ..., S_k$ are a disjoint, box decomposition of the non-dominated space that can be used to efficiently compute the hypervolume improvement.

## C $q$NEHVI under Different Computational Approaches

### C.1 Derivation of IEP formulation of $q$NEHVI

From (4), the expected noisy joint hypervolume improvement is given by

$$\hat{\alpha}_{q\text{NEHVI}}(\mathcal{X}_{\text{cand}}) = \frac{1}{N} \sum_{t=1}^{N} \text{HVI}(\tilde{\boldsymbol{f}}_t(\mathcal{X}_{\text{cand}})|\mathcal{P}_t)$$

Recall that the joint HVI formulation under the IEP derived by Daulton et al. [10] is given by

$$\text{HVI}(\boldsymbol{f}(\mathcal{X}_{\text{cand}})|\mathcal{P}) = \sum_{k=1}^{K} \sum_{j=1}^{q} \sum_{X_j \in \mathcal{X}_j} (-1)^{j+1} \prod_{m=1}^{M} \left[ z_{k,X_j}^{(m)} - l_k^{(m)} \right]_+ \tag{8}$$

where $\mathcal{X}_j := \{X_j \subseteq \mathcal{X}_{\text{cand}} : |X_j| = j\}$ and $z_{k,t,X_j}^{(m)} := \min[u_{k,t}^{(m)}, f^{(m)}(\boldsymbol{x}_{i_1}), ..., f^{(m)}(\boldsymbol{x}_{i_j})]$ for $X_j = \{\boldsymbol{x}_{i_1}, ..., \boldsymbol{x}_{i_j}\}$. In $q$NEHVI, the lower and upper bounds and the number of rectangles in each box decomposition depend $\mathcal{P}_t$. Hence,

$$\hat{\alpha}_{q\text{NEHVI}}(\mathcal{X}_{\text{cand}}) = \frac{1}{N} \sum_{t=1}^{N} \sum_{k=1}^{K_t} \sum_{j=1}^{q} \sum_{X_j \in \mathcal{X}_j} (-1)^{j+1} \prod_{m=1}^{M} \left[ z_{k,t,X_j}^{(m)} - l_{k,t}^{(m)} \right]_+$$

where $z_{k,t,X_j}^{(m)} := \min[u_{k,t}^{(m)}, \tilde{f}_t^{(m)}(\boldsymbol{x}_{i_1}), ..., \tilde{f}_t^{(m)}(\boldsymbol{x}_{i_j})]$ for $X_j = \{\boldsymbol{x}_{i_1}, ..., \boldsymbol{x}_{i_j}\}$.

### C.2 Derivation of CBD formulation of $q$NEHVI

Using Definition 2, we rewrite (4) as

$$\hat{\alpha}_{q\text{NEHVI}}(\mathcal{X}_{\text{cand}}) = \frac{1}{N} \sum_{t=1}^{N} \text{HVI}(\tilde{\boldsymbol{f}}_t(\mathcal{X}_{\text{cand}})|\mathcal{P}_t)$$

$$= \frac{1}{N} \sum_{t=1}^{N} \left[ \text{HV}(\tilde{\boldsymbol{f}}(\mathcal{X}_{\text{cand}}) \cup P_t) - \text{HV}(P_t) \right]$$

Adding and subtracting $\text{HV}(\tilde{\boldsymbol{f}}(\{\boldsymbol{x}_1\}, ..., \tilde{\boldsymbol{f}}(x_{q-1})\}) \cup P_t)$ yields

$$\hat{\alpha}_{q\text{NEHVI}}(\mathcal{X}_{\text{cand}}) = \frac{1}{N} \sum_{t=1}^{N} \left[ \text{HV}(\tilde{\boldsymbol{f}}(\mathcal{X}_{\text{cand}}) \cup P_t) - \text{HV}(\tilde{\boldsymbol{f}}(\{\boldsymbol{x}_1\}, ..., \tilde{\boldsymbol{f}}(x_{q-1})\}) \cup P_t) \right.$$

$$\left. + \text{HV}(\{\tilde{\boldsymbol{f}}(\boldsymbol{x}_1), ..., \tilde{\boldsymbol{f}}(x_{q-1})\} \cup P_t) - \text{HV}(P_t) \right].$$

Applying Definition 2 again leads to (6):

$$\hat{\alpha}_{q\text{NEHVI}}(\mathcal{X}_{\text{cand}}) = \frac{1}{N} \sum_{t=1}^{N} \text{HVI}(\tilde{\boldsymbol{f}}(\boldsymbol{x}_q)|\{\tilde{\boldsymbol{f}}(\boldsymbol{x}_1), ..., \tilde{\boldsymbol{f}}(\boldsymbol{x}_{q-1})\} \cup P_t)$$

$$+ \frac{1}{N} \sum_{t=1}^{N} \text{HVI}(\{\tilde{\boldsymbol{f}}(\boldsymbol{x}_1), ..., \tilde{\boldsymbol{f}}(\boldsymbol{x}_{q-1})\})|P_t).$$

Note that using the method of common random numbers, the CBD formulation is mathematically equivalent to IEP formulation, but the computing $q$NEHVI with the CBD trick is much more efficient.

## D  Complexity Analysis

### D.1  Complexity of Computing $q$NEHVI

In this section we study the complexity of computing the acquisition function. For brevity, we omit the cost of posterior sampling, which is the same for the CBD and IEP approaches.[6]

The CBD approach requires recomputing box decompositions when generating each new candidate. In the worst case, each new candidate is Pareto optimal under the fixed posterior samples, which leads to a time complexity of $O\big(N(n+i)^M\big)$ for computing the box decompositions in iteration $i$ [58]. Note that there are $O\big((n+i)^M\big)$ rectangles in each box decomposition. Given box decompositions and posterior samples at the new point, the complexity of computing the acquisition function on a single-threaded machine is $O\big(MN(n+i)^M\big)$. Hence, the total time complexity for generating $q$ candidates (ignoring potentially additional time complexity for automated gradient computations) is

$$O\left(N \sum_{i=1}^{q}(n+i)^M\right) + O\left(N_{\text{opt}}MN \sum_{i=1}^{q}(n+i)^M\right) = O\left(N_{\text{opt}}NM(n+q)^M q\right), \tag{9}$$

$$O\big(Nn^M\big) + O\left(N_{\text{opt}}MNn^M \sum_{i=1}^{q} 2^{i-1}\right) = O\left(N_{\text{opt}}NMn^M 2^q q\right). \tag{10}$$

The second term on the left hand side in both (9) and (10) is the acquisition optimization complexity, which boils down to $O(N_{\text{opt}})$ given infinite computing cores because the acquisition computation is completely parallelizable. However, as shown in Figure 2, even for relatively small values of $q$, CPU cores become saturated and GPU memory limits are reached.

Everything else fixed, the asymptotic relative time complexity of using CBD over IEP is therefore $q^{-M}2^q \to \infty$ as $q \to \infty$.

Similarly, the space complexity under the CBD formulation, $O\big(MN(n+q)^M\big)$, is also polynomial in $q$, whereas the space complexity is exponential in $q$ under the IEP formulation: $O\big(MNn^M q2^q\big)$.

Everything else fixed, the asymptotic relative complexity (both in terms of time and space) of using CBD over IEP is therefore $q^M 2^{-q} \to 0$ as $q \to \infty$.

### D.2  Efficient Batched Computation

As noted above, using either the IEP or CBD approach, the acquisition computation given the box decompositions is highly parallelizable. However, since the number of hyperrectangles $K_t$ in the box decompoosition can be different under each posterior sample $\tilde{\boldsymbol{f}}_t$, stacking the box decompositions does not result in a rectangular matrix; the matrix is ragged. In order to leverage modern batched

---

[6]Sampling from $p(\boldsymbol{f}(X_n)|\mathcal{D}_n)$ incurs a one-time cost of $O(Mn^3)$ if each of the $M$ outcomes is modeled by an independent GP, as it involves computing a Cholesky decomposition of the $n \times n$ posterior covariance (at the $n$ observed points) for each. Using low-rank updates of the Cholesky factor to sample from $p(\boldsymbol{f}(X_n, \boldsymbol{x}_0, ..., \boldsymbol{x}_i)|\mathcal{D}_n)$ has a time complexity of $O(M(n+i-1)^2)$ for $1 \le i \le q$ since each triangular solve has quadratic complexity. Sampling is more costly when using a multi-task GP model, as it requires a root decomposition of the $Mn \times Mn$ posterior covariance across data points and tasks.

tensor computational paradigms, we pad the box decompositions with empty hyperrectangles (e.g. $l = 0, u = 0$) such that the box decomposition under every posterior sample contains exactly $K = \max_t K_t$ hyperrectangles, which allows us to define a $t \times K$ dimensional matrix of box decompositions for use in batched tensor computation.

In the 2-objective case, instead of padding the box decomposition, the Pareto frontier under each posterior sample can be padded instead by repeating a point on the Pareto Frontier such that the padded Pareto frontier under every posterior sample has exactly $\max_t |\mathcal{P}_t|$ points. This enables computing the box decompositions analytically for all posterior samples in parallel using efficient batched computation. The resulting box decompositions all have $K = \max_t |\mathcal{P}_t| + 1$ hyperrectangles (some of which may be empty).

# E   Theoretical Results

Let $\boldsymbol{x}_{\text{prev}} \in \mathbb{R}^{nd}$ denote the stacked set of previously evaluated points in $X_n$: $\boldsymbol{x}_{\text{prev}} := [\boldsymbol{x}_1^T, ..., \boldsymbol{x}_n^T]^T$. Similarly, let $\boldsymbol{x}_{\text{cand}} \in \mathbb{R}^{qd}$ denote the stacked set of candidates in $\mathcal{X}_{\text{cand}}$: $\boldsymbol{x}_{\text{cand}} := [\boldsymbol{x}_{n+1}^T, ..., \boldsymbol{x}_{n+q}^T]^T$. Let $\tilde{\boldsymbol{f}}_t(\boldsymbol{x}_{\text{prev}}, \boldsymbol{x}_{\text{cand}}) := [\tilde{\boldsymbol{f}}_t(\boldsymbol{x}_1)^T, ..., \tilde{\boldsymbol{f}}_t(\boldsymbol{x}_{n+q})^T]^T$ denote the $t^{\text{th}}$ sample of the corresponding objectives, which we write using the parameterization trick as

$$\boldsymbol{f}_t(\boldsymbol{x}_{\text{prev}}, \boldsymbol{x}_{\text{cand}}) = \mu(\boldsymbol{x}_{\text{prev}}, \boldsymbol{x}_{\text{cand}}) + L(\boldsymbol{x}_{\text{prev}}, \boldsymbol{x}_{\text{cand}})\boldsymbol{\zeta}_t,$$

where $\mu(\boldsymbol{x}_{\text{prev}}, \boldsymbol{x}_{\text{cand}}) : \mathbb{R}^{(n+q)d} \to \mathbb{R}^{(n+q)M}$ is the multi-output GP's posterior mean and $L(\boldsymbol{x}_{\text{cand}}, \boldsymbol{x}_{\text{prev}}) \in \mathbb{R}^{(n+q)M \times (n+q)M}$ is a root decomposition (often a Cholesky decomposition) of the multi-output GP's posterior covariance $\Sigma(\boldsymbol{x}_{\text{cand}}, \boldsymbol{x}_{\text{prev}}) \in \mathbb{R}^{(n+q)M \times (n+q)M}$, and $\boldsymbol{\zeta}_t \in \mathbb{R}^{(n+q)M}$ with $\boldsymbol{\zeta}_t \sim \mathcal{N}(0, I_{(n+q)M})$.[7]

*Proof of Theorem 1.* Since the sequential NEHVI is equivalent to the qNEHVI with $q = 1$, we prove Theorem 1 for the general $q > 1$ case. Recall from Section C.2, that using the method of common random numbers to fix the base samples, the IEP and CBD formulations are equivalent. Therefore, we proceed only with the IEP formulation for this proof.

We closely follow the proof of Theorem 2 in Daulton et al. [10]. We consider the setting from Balandat et al. [2, Section D.5]. Let $f_t^{(m)}(\boldsymbol{x}_i, \boldsymbol{\zeta}_t) = S_{\{i,m\}}(\mu(\boldsymbol{x}_{\text{cand}}, \boldsymbol{x}_{\text{prev}}) + L(\boldsymbol{x}_{\text{cand}}, \boldsymbol{x}_{\text{prev}})\boldsymbol{\zeta}_t)$ denote the posterior distribution over the $m^{\text{th}}$ outcome at $\boldsymbol{x}_i$ as a random variable, where $S_{\{i,m\}}$ denotes the selection matrix ($\|S_{\{i,m\}}\|_\infty \le 1$ for all $i = 1, ..., n+q$ and $m = 1, ..., M$), to extract the element corresponding to outcome $m$ for the point $\boldsymbol{x}_i$. The HVI under a single posterior sample is given by

$$A(\boldsymbol{x}_{\text{cand}}, \boldsymbol{\zeta}_t; \boldsymbol{x}_{\text{prev}}) = \sum_{k=1}^{K_t} \sum_{j=1}^{q} \sum_{X_j \in \mathcal{X}_j} (-1)^{j+1} \prod_{m=1}^{M} \left[ z_{k,X_j}^{(m)}(\boldsymbol{\zeta}_t) - l_k^{(m)} \right]_+$$

where $\mathcal{X}_j := \{X_j = \{\boldsymbol{x}_{i_1}, ... \boldsymbol{x}_{i_j}\} \subseteq \mathcal{X}_{\text{cand}} : |X_j| = j, n+1 \le i_1 \le i_j \le n+q\}$ and $z_{k,X_j}^{(m)}(\boldsymbol{\zeta}_t) = \min\left[u_k^{(m)}, f^{(m)}(\boldsymbol{x}_{i_1}, \boldsymbol{\zeta}_t), ..., f^{(m)}(\boldsymbol{x}_{i_j}, \boldsymbol{\zeta}_t)\right]$. Note that the box decomposition of the non-dominated space $\{S_1, ..., S_{K_t}\}$ and the number of rectangles in the box decomposition depend on $\boldsymbol{\zeta}_t$. Importantly, the number of hyperrectangles $K_t$ in the decomposition is a finite and bounded by $O(|\mathcal{P}_t|^{\lfloor \frac{M}{2} \rfloor + 1})$ [33, 58], where $|\mathcal{P}_t| \le n$.

To satisfy the conditions of [2, Theorem 3], we need to show that there exists an integrable function $\ell : \mathbb{R}^{q \times M} \mapsto \mathbb{R}$ such that for almost every $\boldsymbol{\zeta}_t$ and all $\boldsymbol{x}_{\text{cand}}, \boldsymbol{y}_{\text{cand}} \subseteq \mathcal{X}$,

$$|A(\boldsymbol{x}_{\text{cand}}, \boldsymbol{\zeta}_t; \boldsymbol{x}_{\text{prev}}) - A(\boldsymbol{y}_{\text{cand}}, \boldsymbol{\zeta}_t; \boldsymbol{x}_{\text{prev}})| \le \ell(\boldsymbol{\zeta}_t)\|\boldsymbol{x}_{\text{cand}} - \boldsymbol{y}_{\text{cand}}\|. \tag{11}$$

We note that $\boldsymbol{x}_{\text{prev}}$ is fixed and omit $\boldsymbol{x}_{\text{prev}}$ for brevity, except where necessary.

Let

$$\tilde{a}_{k,m,j,X_j}(\boldsymbol{x}_{\text{cand}}, \boldsymbol{\zeta}_t) := \left[\min\left[u_{k,t}^{(m)}, f^{(m)}(\boldsymbol{x}_{i_1}, \boldsymbol{\zeta}_t), ..., f^{(m)}(\boldsymbol{x}_{i_j}, \boldsymbol{\zeta}_t)\right] - l_{k,t}^{(m)}\right]_+.$$

---

[7]Theorem 1 can be extended to handle non-*iid* base samples from a family of quasi-Monte Carlo methods as in Balandat et al. [2].

Because of linearity, it suffices to show that this condition holds for

$$\tilde{A}(\boldsymbol{x}_{\text{cand}}, \boldsymbol{\zeta}_t) := \prod_{m=1}^{M} \tilde{a}_{k,m,j,X_j}(\boldsymbol{x}_{\text{cand}}, \boldsymbol{\zeta}_t) = \prod_{m=1}^{M} \left[ \min \left[ u_{k,t}^{(m)}, f^{(m)}(\boldsymbol{x}_{i_1}, \boldsymbol{\zeta}_t), \dots, f^{(m)}(\boldsymbol{x}_{i_j}, \boldsymbol{\zeta}_t) \right] - l_{k,t}^{(m)} \right]_+$$

(12)

for all $k, j$, and $X_j$. Note that we can bound $\tilde{a}_{k,m,j,X_j}(\boldsymbol{x}_{\text{cand}}, \boldsymbol{\zeta}_t)$ by

$$\tilde{a}_{k,m,j,X_j}(\boldsymbol{x}_{\text{cand}}, \boldsymbol{\zeta}_t) \leq \left| \min \left[ u_{k,t}^{(m)}, f^{(m)}(\boldsymbol{x}_{i_1}, \boldsymbol{\zeta}_t), \dots, f^{(m)}(\boldsymbol{x}_{i_j}, \boldsymbol{\zeta}_t) \right] - l_{k,t}^{(m)} \right|$$
$$\leq |l_{k,t}^{(m)}| + \left| \min \left[ u_{k,t}^{(m)}, f^{(m)}(\boldsymbol{x}_{i_1}, \boldsymbol{\zeta}_t), \dots, f^{(m)}(\boldsymbol{x}_{i_j}, \boldsymbol{\zeta}_t) \right] \right|.$$

(13)

Consider the case where $u_{k,t}^{(m)} = \infty$. Then

$$\min[u_{k,t}^{(m)}, f(\boldsymbol{x}_{i_1}, \boldsymbol{\zeta}_t)^{(m)}, ..., f^{(m)}(\boldsymbol{x}_{i_j}, \boldsymbol{\zeta}_t)] = \min[f^{(m)}(\boldsymbol{x}_{i_1}, \boldsymbol{\zeta}_t), ..., f^{(m)}(\boldsymbol{x}_{i_j}, \boldsymbol{\zeta}_t)].$$

Now suppose $u_{k,t}^{(m)} < \infty$. Then

$$\min[u_{k,t}^{(m)}, f^{(m)}(\boldsymbol{x}_{i_1}, \boldsymbol{\zeta}_t), ... f^{(m)}(\boldsymbol{x}_{i_j}, \boldsymbol{\zeta}_t)] < \left|\min[f^{(m)}(\boldsymbol{x}_{i_1}, \boldsymbol{\zeta}_t), ... f^{(m)}(\boldsymbol{x}_{i_j}, \boldsymbol{\zeta}_t)]\right| + \left|u_{k,t}^{(m)}\right|.$$

Let

$$w_{k,t}^{(m)} = \begin{cases} u_{k,t}^{(m)}, & \text{if } u_{k,t}^{(m)} < \infty \\ 0, & \text{otherwise.} \end{cases}$$

Note that $l_{k,t}^{(m)}$ is finite and bounded from above and below by $r^{(m)} \leq l_{k,t}^{(m)} < u_{k,t}^{(m)}$ for all $k, t, m$, where $r^{(m)}$ is the $m^{\text{th}}$ dimension of the reference point.

Hence, we can express the bound in (13) as

$$\tilde{a}_{k,m,j,X_j}(\boldsymbol{x}_{\text{cand}}, \boldsymbol{\zeta}_t) \leq |l_{k,t}^{(m)}| + |w_{k,t}^{(m)}| + \left| \min \left[ f^{(m)}(\boldsymbol{x}_{i_1}, \boldsymbol{\zeta}_t), \dots, f^{(m)}(\boldsymbol{x}_{i_j}, \boldsymbol{\zeta}_t) \right] \right|$$
$$\leq |l_{k,t}^{(m)}| + |w_{k,t}^{(m)}| + \sum_{i_1, \dots, i_j} \left| f^{(m)}(\boldsymbol{x}_{i_j}, \boldsymbol{\zeta}_t) \right|.$$

(14)

Note that we can bound $\sum_{i_1, \dots, i_j} \left| f^{(m)}(\boldsymbol{x}_{i_j}, \boldsymbol{\zeta}_t) \right|$ by

$$\sum_{i_1, \dots, i_j} \left| f^{(m)}(\boldsymbol{x}_{i_j}, \boldsymbol{\zeta}_t) \right| \leq |X_j| \left( \| \mu^{(m)}(\boldsymbol{x}_{\text{cand}}, \boldsymbol{x}_{\text{prev}}) \| + \| L^{(m)}(\boldsymbol{x}_{\text{cand}}, \boldsymbol{x}_{\text{prev}}) \| \| \boldsymbol{\zeta}_t \| \right).$$

Substituting this into (14) yields

$$|\tilde{a}_{k,m,j,X_j}(\boldsymbol{x}_{\text{cand}}, \boldsymbol{\zeta}_t)| \leq |l_{k,t}^{(m)}| + |w_{k,t}^{(m)}| + |X_j| \left( \| \mu^{(m)}(\boldsymbol{x}_{\text{cand}}, \boldsymbol{x}_{\text{prev}}) \| + \| L^{(m)}(\boldsymbol{x}_{\text{cand}}, \boldsymbol{x}_{\text{prev}}) \| \| \boldsymbol{\zeta}_t \| \right)$$

(15)

for all $k, m, j, X_j$.

Because of our assumptions of that $\mathcal{X}$ is compact and that the mean and covariance functions are continuously differentiable, $\mu(\boldsymbol{x}_{\text{cand}}, \boldsymbol{x}_{\text{prev}}), L(\boldsymbol{x}_{\text{cand}}, \boldsymbol{x}_{\text{prev}}), \nabla_{\boldsymbol{x}_{\text{cand}}} \mu(\boldsymbol{x}_{\text{cand}}, \boldsymbol{x}_{\text{prev}})$, and $\nabla_{\boldsymbol{x}_{\text{cand}}} L(\boldsymbol{x}_{\text{cand}}, \boldsymbol{x}_{\text{prev}})$ are uniformly bounded. Hence, there exist $C_1, C_2 < \infty$ such that

$$|\tilde{a}_{k,m,j,X_j}(\boldsymbol{x}_{\text{cand}}, \boldsymbol{\zeta}_t)| \leq C_1 + C_2 \| \boldsymbol{\zeta}_t \|$$

for all $k, m, j, X_j$.

Consider the $M = 2$ case. Omitting the indices $k, t, j, X_j$ for brevity, we have

$$\left| \tilde{A}(\boldsymbol{x}_{\text{cand}}, \boldsymbol{\zeta}_t) - \tilde{A}(\boldsymbol{y}_{\text{cand}}, \boldsymbol{\zeta}_t) \right|$$
$$= \left| \tilde{a}_1(\boldsymbol{x}_{\text{cand}}, \boldsymbol{\zeta}_t) \tilde{a}_2(\boldsymbol{x}_{\text{cand}}, \boldsymbol{\zeta}_t) - \tilde{a}_1(\boldsymbol{y}_{\text{cand}}, \boldsymbol{\zeta}_t) \tilde{a}_2(\boldsymbol{y}_{\text{cand}}, \boldsymbol{\zeta}_t) \right|$$
$$= \left| \tilde{a}_1(\boldsymbol{x}_{\text{cand}}, \boldsymbol{\zeta}_t) \left( \tilde{a}_2(\boldsymbol{x}_{\text{cand}}, \boldsymbol{\zeta}_t) - \tilde{a}_2(\boldsymbol{y}_{\text{cand}}, \boldsymbol{\zeta}_t) \right) + \tilde{a}_2(\boldsymbol{y}_{\text{cand}}, \boldsymbol{\zeta}_t) \left( \tilde{a}_1(\boldsymbol{x}_{\text{cand}}, \boldsymbol{\zeta}_t) - \tilde{a}_1(\boldsymbol{y}_{\text{cand}}, \boldsymbol{\zeta}_t) \right) \right|$$
$$\leq \left| \tilde{a}_1(\boldsymbol{x}_{\text{cand}}, \boldsymbol{\zeta}_t) \right| \left| \tilde{a}_2(\boldsymbol{x}_{\text{cand}}, \boldsymbol{\zeta}_t) - \tilde{a}_2(\boldsymbol{y}_{\text{cand}}, \boldsymbol{\zeta}_t) \right| + \left| \tilde{a}_2(\boldsymbol{y}_{\text{cand}}, \boldsymbol{\zeta}_t) \right| \left| \tilde{a}_1(\boldsymbol{x}_{\text{cand}}, \boldsymbol{\zeta}_t) - \tilde{a}_1(\boldsymbol{y}_{\text{cand}}, \boldsymbol{\zeta}_t) \right|.$$

(16)

Using (15), we can bound $|\tilde{a}_{k,m,j,X_j}(\boldsymbol{x}_{\text{cand}}, \boldsymbol{\zeta}_t) - \tilde{a}_{kmjX_j}(\boldsymbol{y}_{\text{cand}}, \boldsymbol{\zeta}_t)|$ by

$$|\tilde{a}_{k,t,m,j,X_j}(\boldsymbol{x}_{\text{cand}}, \boldsymbol{\zeta}_t) - \tilde{a}_{k,t,m,j,X_j}(\boldsymbol{y}_{\text{cand}}, \boldsymbol{\zeta}_t)|$$
$$\leq \sum_{i_1,\ldots,i_j} \left| S_{\{i_j,m\}}(\mu(\boldsymbol{x}_{\text{cand}}, \boldsymbol{x}_{\text{prev}}) + L(\boldsymbol{x}_{\text{cand}}, \boldsymbol{x}_{\text{prev}})\boldsymbol{\zeta}_t) - S_{\{i_j,m\}}(\mu(\boldsymbol{y}_{\text{cand}}, \boldsymbol{x}_{\text{prev}}) + L(\boldsymbol{y}_{\text{cand}}, \boldsymbol{x}_{\text{prev}})\boldsymbol{\zeta}_t) \right|$$
$$\leq |X_j|\Big(\|\mu(\boldsymbol{x}_{\text{cand}}, \boldsymbol{x}_{\text{prev}}) - \mu(\boldsymbol{y}_{\text{cand}}, \boldsymbol{x}_{\text{prev}})\| + \|L(\boldsymbol{x}_{\text{cand}}, \boldsymbol{x}_{\text{prev}}) - L(\boldsymbol{y}_{\text{cand}}, \boldsymbol{x}_{\text{prev}})\|\|\boldsymbol{\zeta}_t\|\Big).$$

Since $\mu$ and $L$ have uniformly bounded gradients with respect to $\boldsymbol{x}_{\text{cand}}$ and $\boldsymbol{y}_{\text{cand}}$, they are Lipschitz. Therefore, there exist $C_3, C_4 < \infty$ such that

$$|\tilde{a}_{k,t,m,j,X_j}(\boldsymbol{x}_{\text{cand}}, \boldsymbol{\zeta}_t) - \tilde{a}_{k,t,m,j,X_j}(\boldsymbol{y}_{\text{cand}}, \boldsymbol{\zeta}_t)| \leq (C_3 + C_4\|\boldsymbol{\zeta}_t\|)\|\boldsymbol{x}_{\text{cand}} - \boldsymbol{y}_{\text{cand}}\| \qquad (17)$$

for all $\boldsymbol{x}_{\text{cand}}, \boldsymbol{y}_{\text{cand}}, k, t, m, j, X_j$.

Substituting (17) into (16), we have

$$\left| \tilde{A}(\boldsymbol{x}_{\text{cand}}, \boldsymbol{\zeta}_t) - \tilde{A}(\boldsymbol{y}_{\text{cand}}, \boldsymbol{\zeta}_t) \right| \leq 2\Big( C_1 C_3 + (C_1 C_4 + C_2 C_3)\|\boldsymbol{\zeta}_t\| + C_2 C_4 \|\boldsymbol{\zeta}_t\|^2 \Big)\|\boldsymbol{x}_{\text{cand}} - \boldsymbol{y}_{\text{cand}}\|$$

The $M > 2$ is very similar to the $M = 2$ case in (16) albeit with more complex expansions. Similarly, There exist $C < \infty$ such that

$$\left| \tilde{A}(\boldsymbol{x}_{\text{cand}}, \boldsymbol{\zeta}_t) - \tilde{A}(\boldsymbol{y}_{\text{cand}}, \boldsymbol{\zeta}_t) \right| \leq C \sum_{m=1}^{M} \|\boldsymbol{\zeta}_t\|^m \|\boldsymbol{x}_{\text{cand}} - \boldsymbol{y}_{\text{cand}}\|$$

Let us define $\ell(\boldsymbol{\zeta}_t) := C \sum_{m=1}^{M} \|\boldsymbol{\zeta}_t\|^m$. Note that $\ell(\boldsymbol{\zeta}_t)$ is integrable because all absolute moments exist for the Gaussian distribution. Since this satisfies the criteria for Theorem 3 in Balandat et al. [2], the theorem holds for qNEHVI. □

### E.1 Unbiased Gradient estimates from the MC formulation

As noted in Section 7, we can show the following (note that this result is not actually required for Theorem 1):

**Proposition 1.** *Suppose that the GP mean and covariance function are continuously differentiable. Suppose further that the candidate set $\mathcal{X}_{cand}$ has no duplicates, and that the sample-level gradients $\nabla_{\boldsymbol{x}}\text{HVI}(\tilde{f}_t(\boldsymbol{x}))$ are obtained using the reparameterization trick as in Balandat et al. [2]. Then*

$$\mathbb{E}\big[\nabla_{\boldsymbol{x}_{cand}}\hat{\alpha}_{q\text{NEHVI}}^N(\boldsymbol{x}_{cand})\big] = \nabla_{\boldsymbol{x}_{cand}}\alpha_{q\text{NEHVI}}(\boldsymbol{x}_{cand}), \qquad (18)$$

*that is, the averaged sample-level gradient is an unbiased estimate of the gradient of the true acquisition function.*

The proof of Proposition 1 closely follows the proof of Proposition 1 in Daulton et al. [10].

## F  Error Bound on Sequential Greedy Approximation for NEHVI

If the acquisition function $\mathcal{L}(\mathcal{X}_{\text{cand}})$ is a normalized (meaning $\mathcal{L}(\emptyset) = 0$), monotone, submodular (meaning that *the increase* in $\mathcal{L}(\mathcal{X}_{\text{cand}})$ is non-increasing as elements are added to $\mathcal{X}_{\text{cand}}$ set function), then the sequential greedy approximation $\hat{\mathcal{L}}$ of $\mathcal{L}$ enjoys regret of no more than $\frac{1}{e}\mathcal{L}^*$, where $\mathcal{L}^* = \max_{\mathcal{X}_{\text{cand}} \subseteq \mathcal{X}} L(\mathcal{X}_{\text{cand}})$ is the optima of $\mathcal{L}$ [17]. We have $\alpha_{q\text{NEHVI}}(\mathcal{X}_{\text{cand}}) = \mathcal{L}(\mathcal{X}_{\text{cand}}) = \mathbb{E}_{\mathcal{P}}\big[\alpha_{q\text{EHVI}}(\mathcal{X}_{\text{cand}}|\mathcal{P})\big]$. For a fixed, known $\mathcal{P}$, Daulton et al. [10] showed that $\alpha_{q\text{EHVI}}$ is submodular set function. In $\alpha_{q\text{NEHVI}}$, $\mathcal{P}$ is a stochastic, so $\alpha_{q\text{EHVI}}(\mathcal{X}_{\text{cand}}|\mathcal{P})$ is a stochastic submodular set function. Because the expectation of a stochastic submodular function is submodular [1], $\alpha_{q\text{NEHVI}}$ is also submodular. Hence, the sequential greedy approximation of $\alpha_{q\text{NEHVI}}$ enjoys regret of no more than $\frac{1}{e}\alpha_{q\text{NEHVI}}^*$. Using the result from Wilson et al. [56], the MC-based approximation $\hat{\alpha}_{q\text{NEHVI}}(\mathcal{X}_{\text{cand}}) = \sum_{t=1}^{N} \text{HVI}\big[\boldsymbol{f}_t(\mathcal{X}_{\text{cand}})|P_t\big]$ also enjoys the same regret bound because HVI is a normalized submodular set function.[8]

---

[8]Submodularity technically requires a finite search space $\mathcal{X}$, whereas in BO $\mathcal{X}$ is typically an infinite set. Nevertheless in similar scenarios, submodularity has been extended to infinite sets (e.g. Wilson et al. [56]).

# G Experiment Setup

## G.1 Implementation / Code used in the experiments

Our implementations of $q$NEHVI, MESMO, PFES are available in the supplementary files and will be open-sourced under MIT license upon publication. For PESMO, we use the open-source implementation in Spearmint (`https://github.com/HIPS/Spearmint/tree/PESM`), which is licensed by Harvard. For DGEMO, MOEA/D-EGO, and TSEMO we use the open-source implementations available at `https://github.com/yunshengtian/DGEMO/tree/master` under the MIT license. For TS-TCH, $q$EHVI, and qNParEGO we use the open-source implementations in BoTorch, which are available at `https://github.com/pytorch/botorch`) under the MIT license.

For the ABR problem, we use the Park simulator, which is available in open-source at `https://github.com/park-project/park` under the MIT license.

## G.2 Algorithm Details

All methods are initialized with $2(d+1)$ points from a scrambled Sobol sequence. All MC acquisition functions uses $N = 128$ quasi-MC samples [2]. All parallel algorithms using sequential greedy optimization for selecting a batch of candidates points and the base samples are redrawn when selecting candidate $x_i, i = 1, ..., q$.

For EHVI-based methods, we leverage the two-step trick proposed by [58] to perform efficient box decompositions; (i) we find the set of local lower bounds for the maximization problem using Algorithm 5 from Klamroth et al. [30][9], and then (ii) using the local lower bounds as a Pareto frontier for the artificial minimization problem, we compute a box decomposition of the dominated space using Algorithm 1 from Lacour et al. [33].

$q$EHVI uses the IEP for computing joint EHVI over a set of candidates and computes EHVI with respect to the observed Pareto frontier. $q$EHVI-PM-CBD uses the Pareto frontier over the posterior means at the previously evaluated points, providing some amount of regularization with respect to the observed values. In addition, $q$EHVI-PM-CBD uses CBD rather than the IEP, which enables scaling to large batch sizes. $q$NEHVI-1 uses 500 fourier basis functions.

For PFES and MESMO, we use 10 sampled (approximate) functions using RFFs (with 500 basis functions) and optimize each function using 5000 iterations of NSGA-II [11] with a population size of 50. For PFES, we partition the dominated space under each sampled Pareto frontier using the algorithm proposed Lacour et al. [33], which is more efficient and yields fewer hyperrectangles than the Quick Hypervolume algorithm used by the PFES authors [50]. For qNParEGO, we use a similar pruning strategy to that in $q$NEHVI to only integrate over the function values of in-sample points that have positive probability of being best with respect to the sampled scalarization. We use the off-the-shelf implementation of qNParEGO in BoTorch [2], which does not use low-rank Cholesky updates; however, we do note that $q$NPAREGO would likely achieve lower wall times using more efficient linear algebra tricks.

For DGEMO, TSEMO, and MOEA/D-EGO, we use the default settings provided in `https://github.com/yunshengtian/DGEMO/tree/master`.

## G.3 Problem Details

All benchmark problems are treated as maximization problems; the objectives for minimization problems are multiplied by -1 to obtain an equivalent maximization problem.

---

[9]More efficient methods for this step exist (e.g. Dächert et al. [14]), but Klamroth et al. [30] can easily leverage vectorized operations and we find it to be efficient in our experiments.

**BraninCurrin** ($M = 2$, $d = 2$)   The BraninCurrin problem involves optimizing two competing functions used in BO benchmarks: Branin and Currin. The goal is minimize both:

$$f^{(1)}(x_1', x_2') = (x_2 - \frac{5.1}{4\pi^2}x_1^2 + \frac{5}{\pi}x_1 - r)^2 + 10(1 - \frac{1}{8\pi})\cos(x_1) + 10$$

$$f^{(2)}(x_1, x_2) = \left[1 - \exp\left(-\frac{1}{(2x_2)}\right)\right]\frac{2300x_1^3 + 1900x_1^2 + 2092x_1 + 60}{100x_1^3 + 500x_1^2 + 4x_1 + 20}$$

where $x_1, x_2 \in [0, 1]$, $x_1' = 15x_1 - 5$, and $x_2' = 15x_2$.

**DTLZ2** ($M = 2$, $d = 6$)   DTLZ2 [12] is a standard problem from the multi-objective optimization literature. The two objectives are

$$f_1(\boldsymbol{x}) = (1 + g(\boldsymbol{x}_M)) \cos\left(\frac{\pi}{2}x_1\right) \cdots \cos\left(\frac{\pi}{2}x_{M-2}\right) \cos\left(\frac{\pi}{2}x_{M-1}\right)$$

$$f_2(\boldsymbol{x}) = (1 + g(\boldsymbol{x}_M)) \cos\left(\frac{\pi}{2}x_1\right) \cdots \cos\left(\frac{\pi}{2}x_{M-2}\right) \sin\left(\frac{\pi}{2}x_{M-1}\right),$$

where $g(\boldsymbol{x}) = \sum_{x_i \in \boldsymbol{x}_M}(x_i - 0.5)^2$, $\boldsymbol{x} \in [0, 1]^d$, and $\boldsymbol{x}_M$ is the $d - M + 1$ elements of $\boldsymbol{x}$.

**ZDT1** ($M = 2$, $d = 4$)   ZDT1 is a benchmark problem from the multi-objective optimization literature [65]. The goal is minimize the following two objectives

$$f^{(1)}(\boldsymbol{x}) = x_1$$

$$f^{(2)}(\boldsymbol{x}) = g(\boldsymbol{x})\left(1 - \sqrt{\frac{f^{(1)}(\boldsymbol{x})}{g(\boldsymbol{x})}}\right)$$

where $g(\boldsymbol{x}) = 1 + \frac{9}{d-1}\sum_{i=2}^{d} x_i$ and $\boldsymbol{x} = [x_1, ..., x_d] \in [0, 1]^d$.

**VehicleSafety** ($M = 3$, $d = 5$)   The 3 objectives are based on a response surface model that is fit to data collected from a simulator and are given by [52]:

$f_1(\boldsymbol{x}) = 1640.2823 + 2.3573285x_1 + 2.3220035x_2 + 4.5688768x_3 + 7.7213633x_4 + 4.4559504x_5$

$f_2(\boldsymbol{x}) = 6.5856 + 1.15x_1 - 1.0427x_2 + 0.9738x_3 + 0.8364x_4 - 0.3695x_1x_4 + 0.0861x_1x_5$
$\qquad + 0.3628x_2x_4 + 0.1106x_1^2 - 0.3437x_3^2 + 0.1764x_4^2$

$f_3(\boldsymbol{x}) = -0.0551 + 0.0181x_1 + 0.1024x_2 + 0.0421x_3 - 0.0073x_1x_2 + 0.024x_2x_3 - 0.0118x_2x_4$
$\qquad - 0.0204x_3x_4 - 0.008x_3x_5 - 0.0241x_2^2 + 0.0109x_4^2$

where $\boldsymbol{x} \in [1, 3]^5$. We seek to (1) minimize mass (a proxy for fuel efficiency), (2) minimize acceleration (a proxy for passenger trauma) in a full-frontal collision, and (3) minimize the distance that the toe-board intrudes into the cabin (a proxy for vehicle fragility) [52].

**AutoML** ($M = 2$, $d = 8$)   . This experiment considers optimizing predictive performance and latency of a deep neural networks (DNN). Practitioners and researchers across many domains use DNNs for recommendation and recognition tasks in low-latency (e.g. on-device) environments [48], where any increase in prediction time degrades the product experience [42]. Simultaneously, researchers are considering increasingly larger architectures that improve predictive performance [47]. Therefore, a firm may be interesting understanding the set of optimal trade-offs between prediction latency and predictive performance. For a demonstration, we consider optimizing ($d = 8$) hyperparameters of DNN (detailed in Table 1) to minimize out-of-sample prediction error and minimize latency on the MNIST data set [34]. Using a small randomized test set leads to noisy evaluations of predictive performance and latency measurements are often noisy due to unrelated fluctuations in the testing environment. As in previous works, we minimize a logit transformation of the prediction error and minimize a logarithm of the ratio between the latency of a proposed DNN and the latency of the fastest DNN [19, 20, 24]. For each evaluation, we randomly partition the 60,000 examples from the MNIST training set into a set of 50,000 examples for training and 10,000 examples for evaluation. We train each network for 8 epochs using SGD with momentum with mini-batches of 512 examples. The learning rate is decayed after every 30 mini-batch updates using the specified

| PARAMETER | SEARCH SPACE |
|---|---|
| LEARNING RATE ($\log_{10}$ SCALE) | [-5.0, -1.0] |
| LEARNING RATE DECAY MULTIPLIER | [0.01, 1.0] |
| DROPOUT RATE | [0.0, 0.7] |
| $L_1$ REGULARIZATION | $[10^{-5}, 0.1]$ |
| $L_2$ REGULARIZATION | $[10^{-5}, 0.1]$ |
| HIDDEN LAYER 1 SIZE | [20, 500] |
| HIDDEN LAYER 2 SIZE | [20, 500] |
| HIDDEN LAYER 3 SIZE | [20, 500] |

Table 1: The search space for the AutoML benchmark.

decay multiplier. We use randomized rounding on the integer parameters before evaluation. For evaluating the performance of different BO methods, we estimate the noiseless objectives using the mean objectives across 3 replications. DNNs are implemented in PyTorch using ReLU activations and a softmax output layer. Latency measurements are taken on a CPU (2x Intel Xeon E5-2680 v4 @ 2.40GHz).

**CarSideImpact ($M = 4$, $d = 7$)** A side-impact test is common practice under European Enhanced Vehicle-Safety Committee to uphold vehicle safety standards [13]. In constrast with the previous VehicleSafety problem where we considered a full-frontal collision, we now consider the problem of tuning parameters controlling the structural design the of an automobile in the case of a *side-impact* collision. This problem has been widely used in various works and has previously used stochastic parameters to account for manufacturing error [13]. We use the recent 4-objective version proposed by Tanabe & Ishibuchi [52] where the goal to minimize the weight of the vehicle, passenger trauma (pubic force), and vehicle damage (the average velocity of the V-pillar). The fourth objective is a combination of 10 other measures of the vehicle durability and passenger safety (see [13] for details). The mathematical formulas for a response surface model fit to data collected from a simulator are given below:

$$f^{(1)}(\boldsymbol{x}) = 1.98 + 4.9x_1 + 6.67x_2 + 6.98x_3 + 4.01x_4 + 1.78x_5 + 10^{-5}x_6 + 2.73x_7$$
$$f^{(2)}(\boldsymbol{x}) = 4.72 - 0.5x_4 - 0.19x_2x_3$$
$$f^{(3)}(\boldsymbol{x}) = 0.5(V_{\text{MBP}}(\boldsymbol{x}) + V_{\text{FD}}(\boldsymbol{x}))$$
$$f^{(4)}(\boldsymbol{x}) = -\sum_{i=1}^{10} \max[g_i(\boldsymbol{x}), 0]$$

where

$$g_1(\boldsymbol{x}) = 1 - 1.16 + 0.3717x_2x_4 + 0.0092928x_3$$
$$g_2(\boldsymbol{x}) = 0.32 - 0.261 + 0.0159x_1x_2 + 0.06486x_1 + 0.019x_2x_7 - 0.0144x_3x_5 - 0.0154464x_6$$
$$g_3(\boldsymbol{x}) = 0.32 - 0.214 - 0.00817x_5 + 0.045195x_1 + 0.0135168x_1 - 0.03099x_2x_6$$
$$\qquad + 0.018x_2x_7 - 0.007176x_3 - 0.023232x_3 + 0.00364x_5x_6 + 0.018x_2^2$$
$$g_4(\boldsymbol{x}) = 0.32 - 0.74 + 0.61x_2 + 0.031296x_3 + 0.031872x_7 - 0.227x_2^2$$
$$g_5(\boldsymbol{x}) = 32 - 28.98 - 3.818x_3 + 4.2x_1x_2 - 1.27296x_6 + 2.68065x_7$$
$$g_6(\boldsymbol{x}) = 32 - 33.86 - 2.95x_3 + 5.057x_1x_2 + 3.795x_2 + 3.4431x_7 - 1.45728$$
$$g_7(\boldsymbol{x}) = 32 - 46.36 + 9.9x_2 + 4.4505x_1$$
$$g_8(\boldsymbol{x}) = 4 - f_2(\boldsymbol{x})$$
$$g_9(\boldsymbol{x}) = 9.9 - V_{\text{MBP}}(\boldsymbol{x})$$
$$g_{10}(\boldsymbol{x}) = 15.7 - V_{\text{FD}}(\boldsymbol{x})$$
$$V_{\text{MBP}}(\boldsymbol{x}) = 10.58 - 0.674x_1x_2 - 0.67275x_2$$
$$V_{\text{FD}}(\boldsymbol{x}) = 16.45 - 0.489x_3x_7 - 0.843x_5x_6$$

. The search space is:

$$x_1 \in [0.5, 1.5]$$
$$x_2 \in [0.45, 1.35]$$
$$x_3, x_4 \in [0.5, 1.5]$$
$$x_5 \in [0.875, 2.625]$$
$$x_6, x_7 \in [0.4, 1.2].$$

As in the VehicleSafety problem, we add zero-mean Gaussian noise to each objective with a standard deviation of 1% the range of each objective.

**Constrained BraninCurrin ($M = 2$, $V = 2$, $d = 2$)**  The constrained BraninCurrin problem uses the same objectives as BraninCurrin, but adds the following disk constraint from [21]:

$$c(x_1', x_2') = 50 - (x_1' - 2.5)^2 - (x_2' - 7.5)^2) \geq 0$$

We add zero-mean Gaussian noise to objectives and the constraint slack observations with a standard deviation of 5% of the range of each outcome.

**SphereEllipsoidal ($M = 2$, $d = 5$)**  The SphereEllipsoidal problem is defined over $x \in [-5, 5]^d$ and the objectives are given by [6]:

$$f^{(1)}(x) = \sum_{i=1}^{d}(x_i - x_{\text{opt},i}^{(1)})^2 + f_{\text{opt}}^{(1)}$$

$$f^{(2)}(x) = \sum_{i=1}^{d} 10^{6\frac{i-1}{d-1}} z_i^2 + f_{\text{opt}}^{(2)}$$

where

$$z_i = T_{\text{osz}}(\delta_i)$$
$$\delta_i = x_i - x_{\text{opt},i}^{(2)}$$
$$T_{\text{osz}}(\delta_i) = \text{sign}(\delta_i) e^{\hat{\delta}_i + 0.049\left(\sin\left[c_1(\delta_i)\hat{\delta}_i\right] + \sin\left[c_2(\delta_i)\hat{\delta}_i\right]\right)}$$

and

$$\hat{\delta}_i = \begin{cases} \log(|\delta_i|), & \text{if } \delta_i \neq 0 \\ 0, & \text{otherwise} \end{cases}$$

$$c_1(\delta_i) = \begin{cases} 10, & \text{if } \delta_i \geq 0 \\ 5.5, & \text{otherwise} \end{cases}$$

$$c_2(\delta_i) = \begin{cases} 7.9, & \text{if } \delta_i \geq 0 \\ 3.1, & \text{otherwise.} \end{cases}$$

We set

$$x_{\text{opt}}^{(1)} = [-0.0299, 2.1458, -3.2922, -2.9438, -1.5406]$$
$$x_{\text{opt}}^{(2)} = [2.0611, -1.7655, -0.7754, 1.8775, -3.7657]$$
$$f_{\text{opt}}^{(1)} = 203.71$$
$$f_{\text{opt}}^{(2)} = 135.6$$

. We add zero-mean Gaussian noise to objectives and the constraint slack observations with a standard deviation of 5% of the range of each outcome.

| PROBLEM | REFERENCE POINT |
|---|---|
| BRANINCURRIN | [-18.00, -6.00] |
| ZDT1 | [-1.10, -1.10] |
| DTLZ2 | $[-1.10]^M$ |
| VEHICLESAFETY | [-1698.55, -11.21, -0.29] |
| ABR | [150.00, -3500.00] |
| AUTOML | [-2.45, 0.60] |
| CARSIDEIMPACT | [-45.49, -4.51, -13.34, -10.39] |
| CONSTRAINEDBRANINCURRIN | [-80.00, -12.00] |
| SPHEREELLIPSOIDAL | $[-261.00, -6.77 \cdot 10^6]$ |

Table 2: The reference points for each benchmark problem.

## G.4  Evaluation Details

To compute the log hypervolume difference metric, we use NSGA-II to estimate the true Pareto frontier (except for the ABR and AutoML problems, where evaluations are time-consuming and we instead take the true Pareto frontier to be the Pareto frontier across the estimated objectives across all methods and replications). Using this Pareto frontier, we compute the hypervolume dominated by the true Pareto frontier in order to calculate the log hypervolume difference. For ZDT1, the hypervolume dominated by the true Pareto frontier can be computed analytically. For Constrained BraninCurrin, we evaluate the logarithm of the difference between the hypervolume dominated by the true feasible Pareto frontier and the feasible in-sample Pareto frontier for each method.

For all problems, we selected the reference point based on the component-wise noiseless nadir point $f_{\text{nadir}}(x) = \min_{x \in \mathcal{X}} f(x)$ and the range of the Pareto frontier for each noiseless objective using the common heuristic [55]: $r = f_{\text{nadir}}(x) - \beta * (f_{\text{ideal}}(x) - f_{\text{nadir}}(x))$, where $\beta = 0.1$ and $f_{\text{ideal}}(x) = \max_{x \in \mathcal{X}} f(x)$.

# H  Experiments

## H.1  Wall Time Results

Tables 3 and 4 report the acquisition optimization wall times for each method. On all benchmark problems except CarSideImpact, $q$NEHVI is faster to optimize than MESMO and PFES on a GPU. The wall times for optimizing $q$NEHVI are competitive with those for $q$NParEGO on most benchmark problems and batch sizes; on many problems, $q$NEHVI is often faster than $q$NParEGO. On the problems VehicleSafety and CarSideImpact problems which have 3 and 4 objectives respectively, we observed tractable wall times, even when generating $q = 32$ candidates. The wall time for 3 and 4 objective problems is larger primarily because the box decompositions are more time-consuming to compute and result in more hyperrectangles as the number of objectives increases. Although, $q$EHVI(-PM) is faster for $q = 1$ and $q = 8$ on many problems, it is unable to scale to large batch sizes and ran out of memory for $q = 8$ on CarSideImpact due to the box decomposition having a large number of hyperrectangles.

| CPU | BRANINCURRIN | ZDT1 | ABR | VEHICLESAFETY |
|---|---|---|---|---|
| MESMO ($q$=1) | 21.24 ($\pm$0.02) | 19.76 ($\pm$0.03) | 23.24 ($\pm$0.04) | 28.39 ($\pm$0.07) |
| PFES ($q$=1) | 22.86 ($\pm$0.05) | 39.82 ($\pm$0.14) | 43.03 ($\pm$0.12) | 53.16 ($\pm$0.17) |
| TS-TCH ($q$=1) | **0.51 ($\pm$0.0)** | **0.48 ($\pm$0.0)** | **0.75 ($\pm$0.0)** | **0.67 ($\pm$0.0)** |
| qEHVI-PM-CBD ($q$=1) | 2.34 ($\pm$0.02) | 3.7 ($\pm$0.02) | 3.56 ($\pm$0.03) | 7.82 ($\pm$0.05) |
| qEHVI ($q$=1) | 0.58 ($\pm$0.0) | 0.66 ($\pm$0.01) | 2.98 ($\pm$0.02) | 5.07 ($\pm$0.03) |
| qNEHVI ($q$=1) | 40.55 ($\pm$0.61) | 35.66 ($\pm$0.47) | 62.29 ($\pm$0.97) | 120.43 ($\pm$1.25) |
| qNPAREGO ($q$=1) | 3.19 ($\pm$0.05) | 1.65 ($\pm$0.02) | 6.94 ($\pm$0.06) | 1.05 ($\pm$0.01) |
| qPAREGO ($q$=1) | 0.58 ($\pm$0.01) | 0.7 ($\pm$0.01) | 2.5 ($\pm$0.03) | 0.75 ($\pm$0.01) |
| DGEMO ($q$=1) | 65.28($\pm$0.26) | 76.99($\pm$0.35) | NA | NA |
| DGEMO ($q$=8) | 65.44($\pm$0.63) | 86.97($\pm$0.85) | NA | NA |
| DGEMO ($q$=16) | 66.44($\pm$0.93) | 86.06($\pm$1.21) | NA | NA |
| DGEMO ($q$=32) | 66.66($\pm$1.47) | 84.66($\pm$1.66) | NA | NA |
| TSEMO ($q$=1) | 3.02($\pm$0.01) | 2.98($\pm$0.01) | NA | 3.61($\pm$0.01) |
| TSEMO ($q$=8) | 3.53($\pm$0.01) | 3.48($\pm$0.01) | NA | 7.45($\pm$0.1) |
| TSEMO ($q$=16) | 3.77($\pm$0.02) | 3.74($\pm$0.02) | NA | 11.06($\pm$0.28) |
| TSEMO ($q$=32) | 4.29($\pm$0.03) | 4.22($\pm$0.02) | NA | 16.3($\pm$0.68) |
| MOEA/D-EGO ($q$=1) | 57.79($\pm$0.17) | 58.1($\pm$0.17) | NA | 71.0($\pm$0.21) |
| MOEA/D-EGO ($q$=8) | 63.56($\pm$0.18) | 63.57($\pm$0.17) | NA | 77.56($\pm$0.22) |
| MOEA/D-EGO ($q$=16) | 64.0($\pm$0.18) | 63.99($\pm$0.19) | NA | 78.03($\pm$0.25) |
| MOEA/D-EGO ($q$=32) | 64.09($\pm$0.26) | 63.9($\pm$0.24) | NA | 77.77($\pm$0.35) |
| **GPU** | BRANINCURRIN | ZDT1 | ABR | VEHICLESAFETY |
| MESMO ($q$=1) | 19.9 ($\pm$0.04) | 19.92 ($\pm$0.04) | 21.54 ($\pm$0.08) | 24.57 ($\pm$0.09) |
| PFES ($q$=1) | 21.68 ($\pm$0.07) | 45.9 ($\pm$0.17) | 43.3 ($\pm$0.13) | 47.25 ($\pm$0.16) |
| TS-TCH ($q$=1) | 0.88 ($\pm$0.01) | 0.94 ($\pm$0.01) | 1.08 ($\pm$0.01) | 1.04 ($\pm$0.01) |
| TS-TCH ($q$=8) | 1.85 ($\pm$0.03) | 2.01 ($\pm$0.03) | 2.99 ($\pm$0.04) | 2.32 ($\pm$0.05) |
| TS-TCH ($q$=16) | 3.08 ($\pm$0.08) | 3.29 ($\pm$0.1) | 4.28 ($\pm$0.08) | 3.54 ($\pm$0.09) |
| TS-TCH ($q$=32) | 5.25 ($\pm$0.15) | 5.41 ($\pm$0.16) | 7.23 ($\pm$0.2) | 6.41 ($\pm$0.23) |
| qEHVI-PM-CBD ($q$=1) | 2.17($\pm$0.01) | 2.12($\pm$0.02) | 3.59($\pm$0.02) | 51.11($\pm$0.28) |
| qEHVI-PM-CBD ($q$=8) | 39.56($\pm$0.79) | 30.83($\pm$1.35) | 36.2($\pm$0.73) | 716.03($\pm$13.44) |
| qEHVI-PM-CBD ($q$=16) | 82.91($\pm$2.42) | 67.3($\pm$3.88) | 70.02($\pm$1.64) | 1410.79($\pm$41.72) |
| qEHVI-PM-CBD ($q$=32) | 147.81($\pm$6.85) | 105.74($\pm$8.55) | 251.97($\pm$12.69) | 2570.95($\pm$116.61) |
| qEHVI ($q$=1) | 0.72 ($\pm$0.01) | 0.99 ($\pm$0.02) | 3.67 ($\pm$0.02) | 3.96 ($\pm$0.05) |
| qEHVI ($q$=8) | 18.12 ($\pm$1.03) | 18.05 ($\pm$0.86) | 40.55 ($\pm$0.58) | 71.49 ($\pm$2.04) |
| qNEHVI ($q$=1) | 6.15 ($\pm$0.06) | 5.75 ($\pm$0.04) | 7.72 ($\pm$0.09) | 20.81 ($\pm$0.11) |
| qNEHVI ($q$=8) | 48.19 ($\pm$1.2) | 46.74 ($\pm$0.83) | 49.7 ($\pm$0.79) | 168.63 ($\pm$2.49) |
| qNEHVI ($q$=16) | 102.87 ($\pm$4.02) | 95.6 ($\pm$2.62) | 93.14 ($\pm$1.72) | 289.02 ($\pm$5.82) |
| qNEHVI ($q$=32) | 177.56 ($\pm$7.81) | 190.59 ($\pm$6.07) | 181.97 ($\pm$4.77) | 546.83 ($\pm$16.09) |
| qNEHVI-1 ($q$=1) | **0.32($\pm$0.0)** | **0.24($\pm$0.0)** | **0.56($\pm$0.0)** | **0.92($\pm$0.0)** |
| qNEHVI-1 ($q$=8) | 2.43($\pm$0.02) | 2.11($\pm$0.03) | 4.55($\pm$0.06) | 7.1($\pm$0.1) |
| qNEHVI-1 ($q$=16) | 4.97($\pm$0.07) | 3.73($\pm$0.06) | 8.73($\pm$0.14) | 14.77($\pm$0.31) |
| qNEHVI-1 ($q$=32) | 9.03($\pm$0.18) | 8.18($\pm$0.24) | 17.15($\pm$0.41) | 34.99($\pm$1.46) |
| qNPAREGO ($q$=1) | 2.39 ($\pm$0.04) | 1.84 ($\pm$0.04) | 6.47 ($\pm$0.05) | 0.9 ($\pm$0.02) |
| qNPAREGO ($q$=8) | 47.05 ($\pm$1.74) | 52.99 ($\pm$1.94) | 74.72 ($\pm$1.9) | 45.56 ($\pm$1.17) |
| qNPAREGO ($q$=16) | 118.73 ($\pm$5.53) | 116.68 ($\pm$5.51) | 116.79 ($\pm$3.19) | 91.3 ($\pm$3.83) |
| qNPAREGO ($q$=32) | 306.17 ($\pm$17.81) | 279.01 ($\pm$17.72) | 240.56 ($\pm$6.44) | 188.42 ($\pm$13.42) |
| qPAREGO ($q$=1) | 0.81 ($\pm$0.02) | 1.05 ($\pm$0.03) | 4.39 ($\pm$0.05) | 0.79 ($\pm$0.02) |
| qPAREGO ($q$=8) | 13.01 ($\pm$0.53) | 16.4 ($\pm$0.72) | 31.02 ($\pm$0.81) | 12.64 ($\pm$0.84) |
| qPAREGO ($q$=16) | 34.34 ($\pm$2.12) | 43.66 ($\pm$3.12) | 66.85 ($\pm$3.13) | 36.68 ($\pm$4.48) |
| qPAREGO ($q$=32) | 139.73 ($\pm$25.22) | 108.25 ($\pm$6.94) | 122.37 ($\pm$6.12) | 107.34 ($\pm$14.76) |

Table 3: Acquisition function optimization wall time (including box decompositions) in seconds on a CPU (2x Intel Xeon E5-2680 v4 @ 2.40GHz) and a Tesla V100 SXM2 GPU (16GB RAM). The mean and two standard errors are reported. DGEMO, TSEMO, and MOEA/D-EGO are omitted for ABR because they have package requirements that are not easily compatible with our distributed evaluation pipeline and ABR evaluations are prohibitively slow without distributed evaluations. DGEMO is omitted for VehicleSafety because the open-source implementation raises an consistently raises an exception in the graph cutting algorithm with this problem.

## H.2 Scaling to large batch sizes with CBD

In Figure 6, we provide results demonstrating the CBD approach enables scaling to large batch sizes, even with 3 or 4 objectives, whereas the IEP wall times grow exponentially with the batch size and the IEP overflows GPU memory even with modest batch sizes.

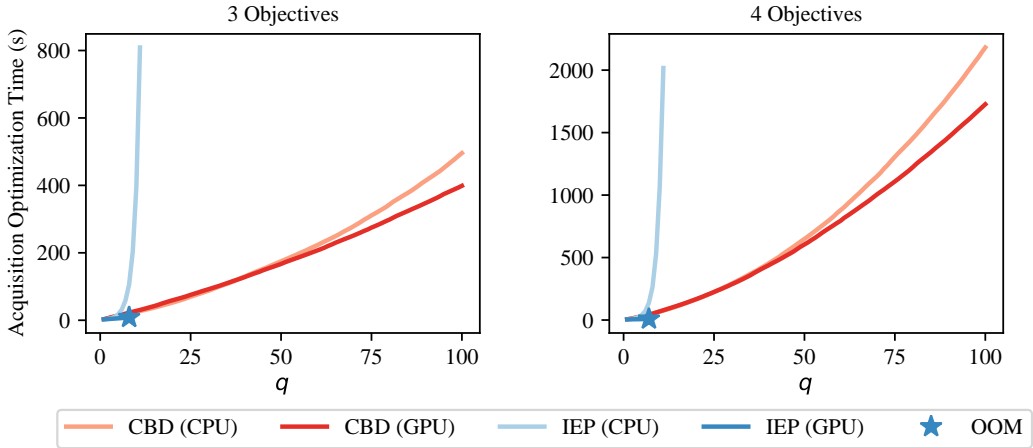

Figure 6: Acquisition optimization wall time under a sequential greedy approximation using L-BFGS-B for three and four objectives. CBD enables scaling to much larger batch sizes $q$ than using the inclusion-exclusion principle (IEP) and avoids running out-of-memory (OOM) on a GPU. Independent GPs are used for each outcome and are initialized with 20 points from the Pareto frontier of the 6-dimensional DTLZ2 problem [12] with 3 objectives (left) and 4 objectives (right). Wall times were measured on a Tesla V100 SXM2 GPU (16GB GPU RAM) and a Intel Xeon Gold 6138 CPU @ 2GHz CPU (251GB RAM).

## H.3 Additional Empirical Results

The additional optimization performance results in the appendix demonstrate that $q$NEHVI-based algorithms are consistently the top performer. The only case where $q$NEHVI(-1) is outperformed is in the sequential setting on the CarSideImpact problem in Figure 9(a), where $q$EHVI performs best However, as show in Figure 9(b) and Figure 9(c), $q$NEHVI enables scaling to large batch sizes, whereas $q$EHVI runs out of memory on a GPU for $q = 8$. Therefore, in a practical setting where vehicles are manufactured and test in parallel, $q$NEHVI would be the best choice.

Figure 11 shows that $q$NEHVI achieves solid performance anytime throughout the learning curve in the sequential setting, and Figure 13 shows that $q$NEHVI-based variants consistently achieves the best performance for various $q$ with a fixed budget of 224 function evaluations. Although, $q$NEHVI-1 does not consistently perform better than $q$NEHVI, $q$NEHVI-1 achieves very little degradation of sample complexity as the batch size $q$ increases.

## H.4 Better performance from $q$EHVI with a larger batch size

Interestingly, on many test problems $q$EHVI performs better with $q = 8$ than $q = 1$. In the case of BraninCurrin and ConstrainedBraninCurrin, $q$ParEGO also improves as $q$ increases. Since this phenomenon is not observed with the noisy acquisition functions ($q$NEHVI, $q$NParEGO), we hypothesize that it may be the case that sequential data collection results in a difficult to optimize acquisition surface and that integrating over the in-sample points leads to a smoother acquisition surface that results in improved sequential optimization. The acquisition functions that do not account for noise may be misled by the noise in sequential setting, but using a larger batch size (within limits) may help avoid the issue of not properly accounting for noise.

## H.5 Performance over Higher Dimensional Spaces

$q$NEHVI-1 relies on approximate GP sample paths (RFFs). Although we find that $q$NEHVI-1 performs very well on many low-dimensional problems (see Figure 4), Figures 8 and 7 show that $q$NEHVI-1 does not perform as well a $q$NEHVI on higher dimensional problems. We hypothesize that the RFF approximation degrades in higher dimensional search spaces leading to poor optimization performance relative $q$NEHVI, which uses exact GP samples. It is likely that using 500 Fourier basis functions leads to large approximation error on high dimensional search spaces. Further study is needed to examine whether robust performance can be achieved by increasing the number of basis functions. As shown in Figure 7, $q$NEHVI consistently outperforms all tested methods regardless of the dimension of the design space.

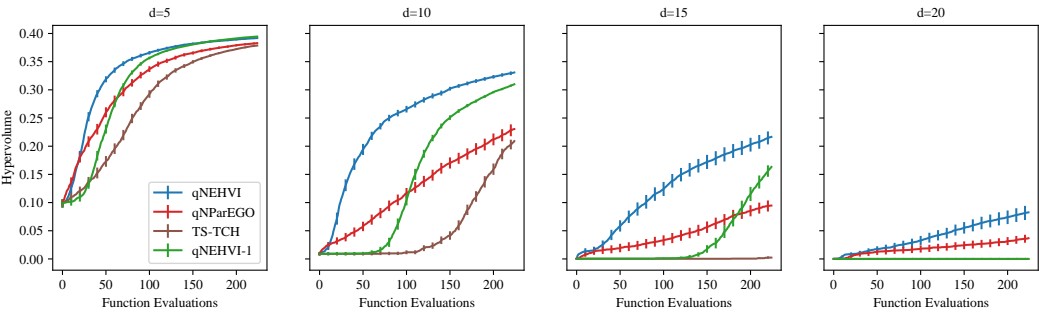

Figure 7: Sequential optimization performance 2-objective DTLZ2 with $\sigma = 5\%$ problems as the dimension of the search space increases from $d = 5$ to $d = 20$.

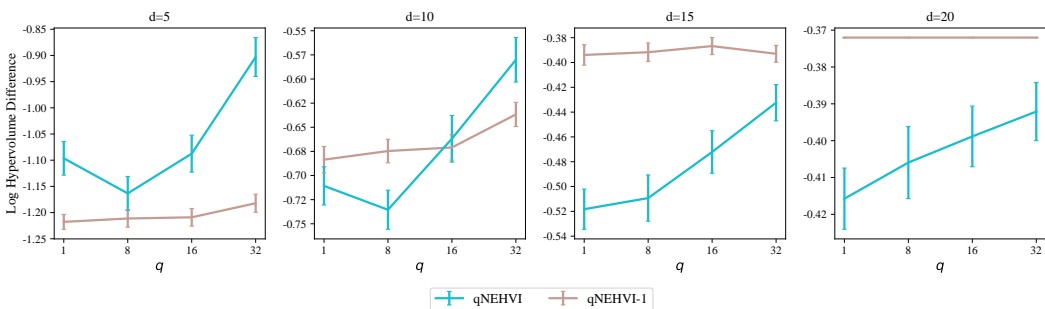

Figure 8: A comparison of the final optimization performance of $q$NEHVI-1, a single sample path approximation of $q$NEHVI, and $q$NEHVI on 2-objective DTLZ2 problems with input dimensions between 5 and 20 under different batch sizes $q$. $q$NEHVI-1 is very effective on lower dimensional problems, but does not perform as well as $q$NEHVI on higher dimensional problems.

| CPU | DTLZ2 | AUTOML | CARSIDEIMPACT | CONSTRAINED BRANINCURRIN |
|---|---|---|---|---|
| MESMO ($q$=1) | 27.79($\pm$0.07) | 37.86 ($\pm$0.08) | 33.31 ($\pm$0.06) | NA |
| PFES ($q$=1) | 69.85($\pm$0.21) | 101.24 ($\pm$0.29) | 102.55 ($\pm$0.34) | NA |
| TS-TCH ($q$=1) | **0.72($\pm$0.0)** | **0.93 ($\pm$0.0)** | **1.27 ($\pm$0.01)** | NA |
| $q$EHVI-PM-CBD ($q$=1) | 3.98($\pm$0.02) | 5.76 ($\pm$0.05) | 83.14 ($\pm$0.74) | 10.27 ($\pm$0.06) |
| $q$EHVI ($q$=1) | 2.74($\pm$0.01) | 5.05 ($\pm$0.04) | 96.19 ($\pm$0.9) | 3.26 ($\pm$0.05) |
| $q$NEHVI ($q$=1) | 54.04($\pm$0.67) | 22.71 ($\pm$0.48) | 541.13 ($\pm$6.83) | 267.67 ($\pm$4.09) |
| $q$NPAREGO ($q$=1) | 21.39($\pm$0.2) | 5.6 ($\pm$0.07) | 3.38 ($\pm$0.05) | 12.05 ($\pm$0.17) |
| $q$PAREGO ($q$=1) | 2.33($\pm$0.02) | 3.06 ($\pm$0.03) | 1.91 ($\pm$0.02) | **1.56 ($\pm$0.03)** |
| DGEMO ($q$=1) | 84.48($\pm$0.64) | NA | NA | NA |
| DGEMO ($q$=8) | 65.99($\pm$0.51) | NA | NA | NA |
| DGEMO ($q$=16) | 69.57($\pm$0.59) | NA | NA | NA |
| DGEMO ($q$=32) | 72.82($\pm$0.8) | NA | NA | NA |
| TSEMO ($q$=1) | 3.1($\pm$0.01) | NA | 14.91($\pm$0.16) | NA |
| TSEMO ($q$=8) | 3.58($\pm$0.01) | NA | 100.78 ($\pm$3.75) | NA |
| TSEMO ($q$=16) | 3.87($\pm$0.02) | NA | 188.44($\pm$9.88) | NA |
| TSEMO ($q$=32) | 4.55($\pm$0.03) | NA | 326.51($\pm$24.03) | NA |
| MOEA/D-EGO ($q$=1) | 59.3($\pm$0.18) | NA | 79.87($\pm$0.16) | NA |
| MOEA/D-EGO ($q$=8) | 65.5($\pm$0.18) | NA | 85.34($\pm$0.21) | NA |
| MOEA/D-EGO ($q$=16) | 65.81($\pm$0.2) | NA | 85.18($\pm$0.3) | NA |
| MOEA/D-EGO ($q$=32) | 65.44($\pm$0.3) | NA | 84.8($\pm$0.39) | NA |
| **GPU** | DTLZ2 | AUTOML | CARSIDEIMPACT | CONSTRAINED BRANINCURRIN |
| MESMO ($q$=1) | 17.01($\pm$0.13) | 28.83 ($\pm$0.2) | 26.0 ($\pm$0.06) | NA |
| PFES ($q$=1) | 46.63($\pm$0.51) | 85.73 ($\pm$0.4) | 55.41 ($\pm$0.16) | NA |
| TS-TCH ($q$=1) | 0.84($\pm$0.01) | 1.49 ($\pm$0.01) | 2.17 ($\pm$0.01) | NA |
| TS-TCH ($q$=8) | 3.71($\pm$0.07) | 3.64 ($\pm$0.1) | 4.15 ($\pm$0.07) | NA |
| TS-TCH ($q$=16) | 5.85($\pm$0.14) | 6.16 ($\pm$0.21) | 6.9 ($\pm$0.19) | NA |
| TS-TCH ($q$=32) | 9.33($\pm$0.3) | 9.47 ($\pm$0.47) | 10.2 ($\pm$0.4) | NA |
| $q$EHVI-PM-CBD ($q$=1) | 4.29($\pm$0.01) | 5.26($\pm$0.05) | 52.7($\pm$0.41) | 15.89($\pm$0.08) |
| $q$EHVI-PM-CBD ($q$=8) | 40.85($\pm$0.41) | 39.55($\pm$0.8) | 460.18($\pm$9.76) | 135.89($\pm$1.43) |
| $q$EHVI-PM-CBD ($q$=16) | 93.39($\pm$1.87) | 71.8($\pm$1.81) | 866.12($\pm$26.46) | 314.42($\pm$5.44) |
| $q$EHVI-PM-CBD ($q$=32) | 194.12($\pm$5.6) | 143.83($\pm$4.88) | 1682.28($\pm$72.5) | 823.01($\pm$24.48) |
| $q$EHVI ($q$=1) | 2.93($\pm$0.02) | 4.67 ($\pm$0.1) | 9.63 ($\pm$0.05) | 5.69 ($\pm$0.11) |
| $q$EHVI ($q$=8) | 39.64($\pm$0.57) | 104.48 ($\pm$1.34) | OOM | 68.95 ($\pm$2.57) |
| $q$NEHVI ($q$=1) | 4.91($\pm$0.01) | 7.95 ($\pm$0.1) | 82.66 ($\pm$0.63) | 20.47 ($\pm$0.12) |
| $q$NEHVI ($q$=8) | 39.96($\pm$0.35) | 67.28 ($\pm$1.87) | 683.06 ($\pm$13.82) | 168.04 ($\pm$1.85) |
| $q$NEHVI ($q$=16) | 74.41($\pm$0.63) | 145.66 ($\pm$4.45) | 1289.4 ($\pm$36.81) | 362.15 ($\pm$9.08) |
| $q$NEHVI ($q$=32) | 142.18($\pm$1.59) | 247.92 ($\pm$11.93) | 2480.41 ($\pm$102.38) | 654.66 ($\pm$23.48) |
| $q$NEHVI-1 ($q$=1) | **0.42($\pm$0.0)** | **0.53($\pm$0.0)** | **2.11($\pm$0.01)** | **1.57($\pm$0.02)** |
| $q$NEHVI-1 ($q$=8) | 3.26($\pm$0.03) | 4.55($\pm$0.09) | 16.88($\pm$0.2) | 11.01($\pm$0.16) |
| $q$NEHVI-1 ($q$=16) | 6.34($\pm$0.04) | 7.22($\pm$0.19) | 34.78($\pm$0.63) | 20.56($\pm$0.4) |
| $q$NEHVI-1 ($q$=32) | 14.85($\pm$0.45) | 12.66($\pm$0.54) | 67.27($\pm$1.61) | 40.6($\pm$1.1) |
| $q$NPAREGO ($q$=1) | 6.41($\pm$0.03) | 4.86 ($\pm$0.1) | 3.25 ($\pm$0.06) | 6.17 ($\pm$0.07) |
| $q$NPAREGO ($q$=8) | 57.17($\pm$0.43) | 48.62 ($\pm$1.18) | 30.65 ($\pm$0.92) | 38.66 ($\pm$0.95) |
| $q$NPAREGO ($q$=16) | 114.91($\pm$1.3) | 122.09 ($\pm$3.5) | 81.04 ($\pm$3.44) | 84.27 ($\pm$3.04) |
| $q$NPAREGO ($q$=32) | 263.56($\pm$4.03) | 275.41 ($\pm$8.29) | 219.98 ($\pm$9.75) | 199.6 ($\pm$9.68) |
| $q$PAREGO ($q$=1) | 2.62($\pm$0.03) | 3.31 ($\pm$0.07) | 3.03 ($\pm$0.06) | 2.83 ($\pm$0.08) |
| $q$PAREGO ($q$=8) | 25.75($\pm$0.66) | 33.84 ($\pm$1.76) | 22.0 ($\pm$0.88) | 20.28 ($\pm$1.18) |
| $q$PAREGO ($q$=16) | 67.89($\pm$2.27) | 77.25 ($\pm$3.9) | 56.09 ($\pm$3.09) | 50.06 ($\pm$3.84) |
| $q$PAREGO ($q$=32) | 159.98($\pm$7.41) | 217.55 ($\pm$19.15) | 139.25 ($\pm$9.41) | 135.39 ($\pm$13.04) |

Table 4: Acquisition function optimization wall time (including box decompositions) in seconds on a CPU (2x Intel Xeon E5-2680 v4 @ 2.40GHz) and a Tesla V100 SXM2 GPU (16GB RAM). The mean and two standard errors are reported. DGEMO, TSEMO, MOEA/D-EGO are omitted for AutoML because they have package requirements that are not easily compatible with our distributed training and evaluation pipeline, and they are omitted for ConstrainedBraninCurrin because they do not support constraints in the open-source implementation at `https://github.com/yunshengtian/DGEMO/tree/master`. DGEMO is omitted on CarSideImpact because the open-source implementation does not support more than 3 objectives.

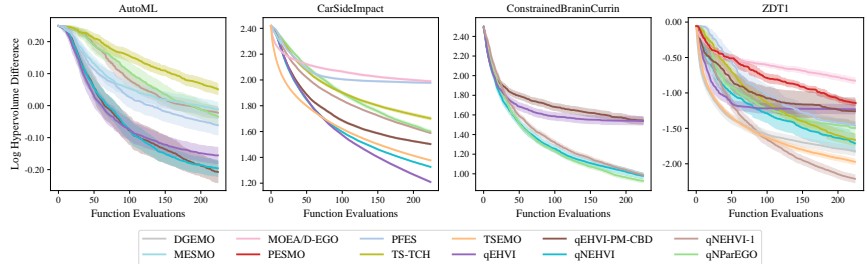

(a) Sequential Optimization performance on additional benchmark problems.

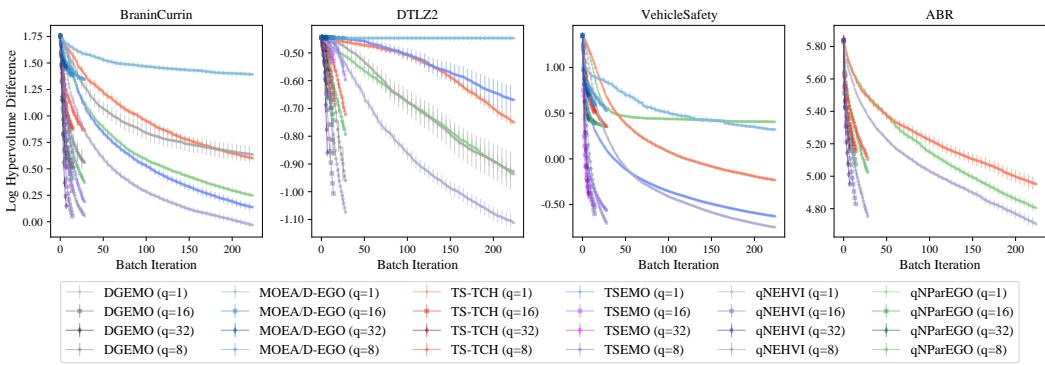

(b) Parallel optimization performance vs batch iterations (1/2).

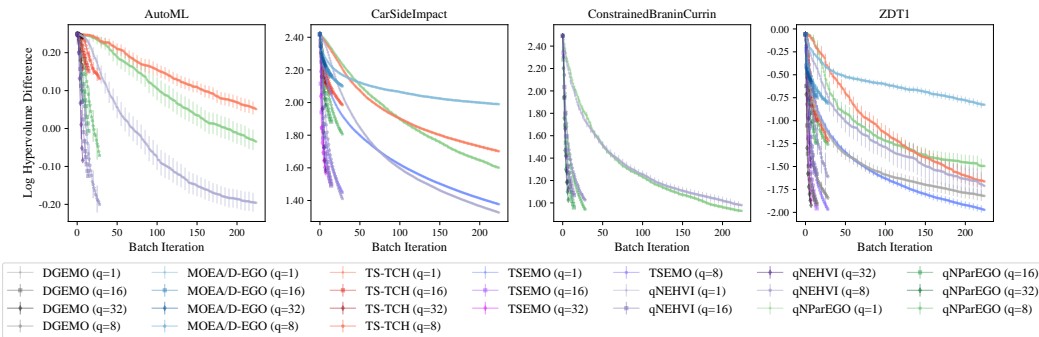

(c) Parallel optimization performance vs batch iterations (2/2).

Figure 9: Optimization performance on additional problems. (a) Sequential optimization performance. (b) and (c) Optimization performance of batch acquisition functions using various $q$ over the number of BO iterations. To improve readability, we omit $q$EHVI(-PM) in this figure because the IEP cannot scale beyond $q = 8$ because of the exponential time and space complexity (running it on a GPU runs out of memory and running it on a CPU results in prohibitively slow wall times). See Figure 10 for results using $q$EHVI(-PM).

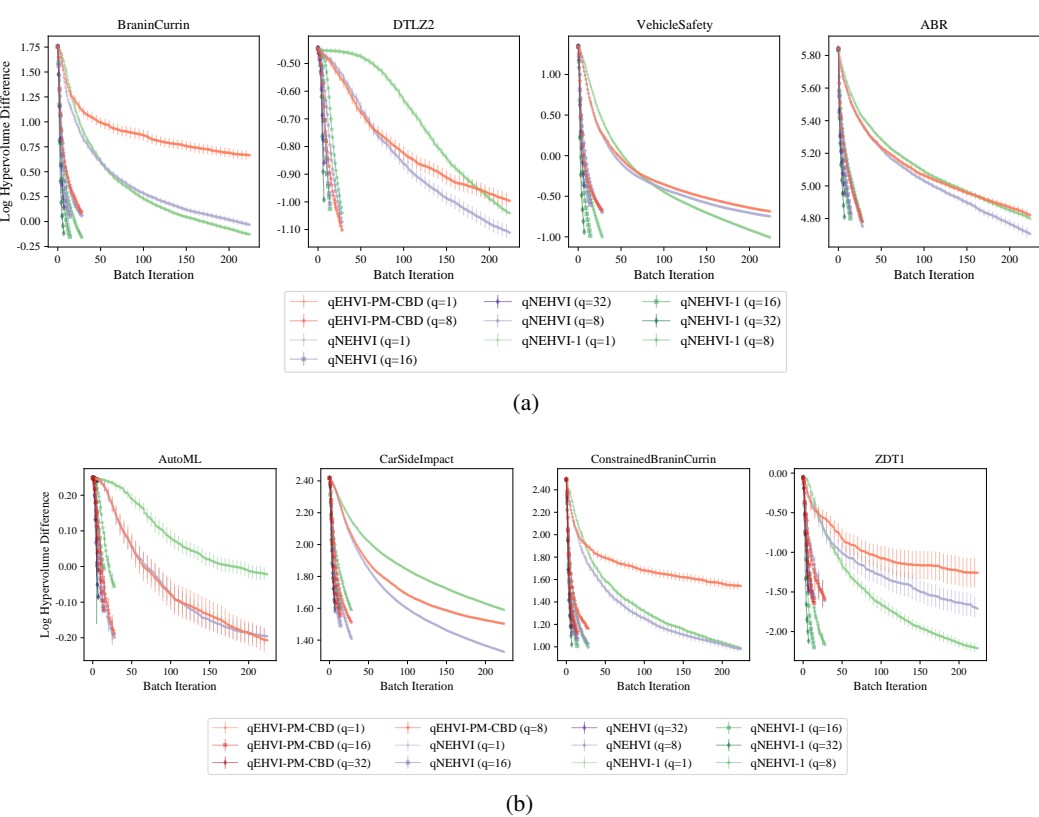

Figure 10: Optimization performance of $q$NEHVI under various batch sizes $q$ vs $q$EHVI(-PM). Note that using the IEP, $q$EHVI(-PM) cannot scale beyond $q = 8$ because of the exponential time and space complexity (running it on a GPU runs out of memory and running it on a CPU results in prohibitively slow wall times).

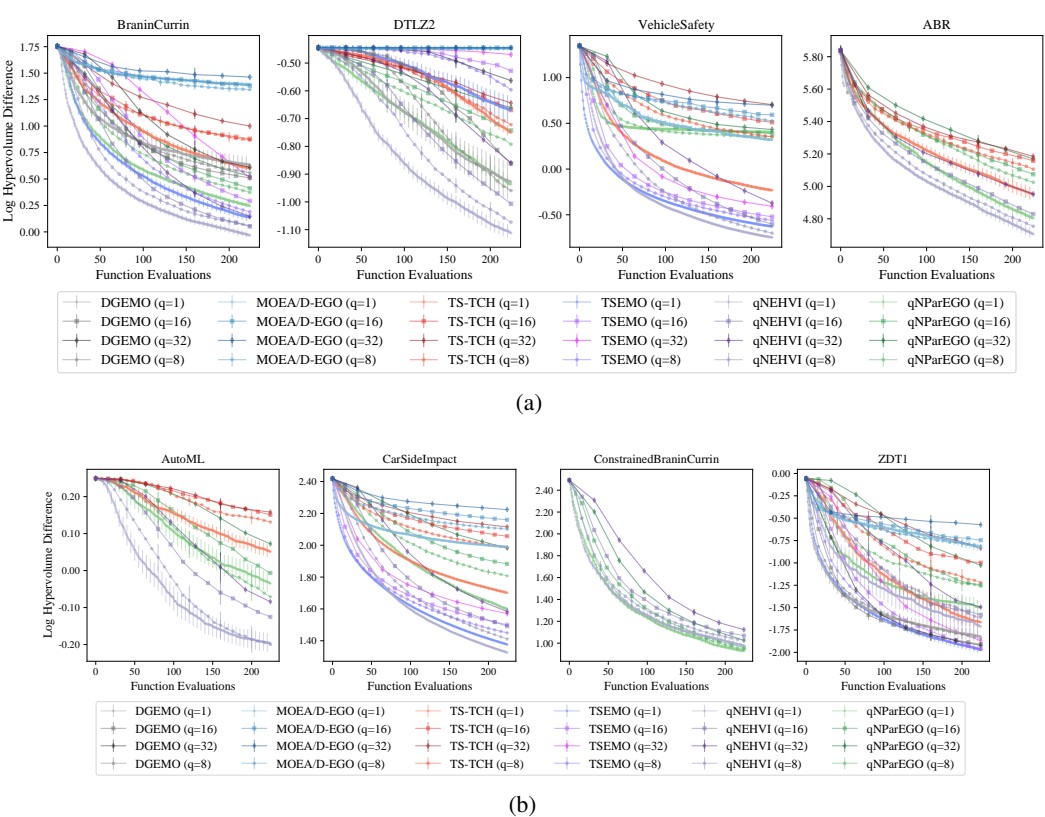

Figure 11: Anytime optimization performance of batch acquisition functions using various $q$ over the number of function evaluations. To improve readability, we omit $q$EHVI(-PM) in this figure because the IEP cannot scale beyond $q = 8$ because of the exponential time and space complexity (running it on a GPU runs out of memory and running it on a CPU results in prohibitively slow wall times).

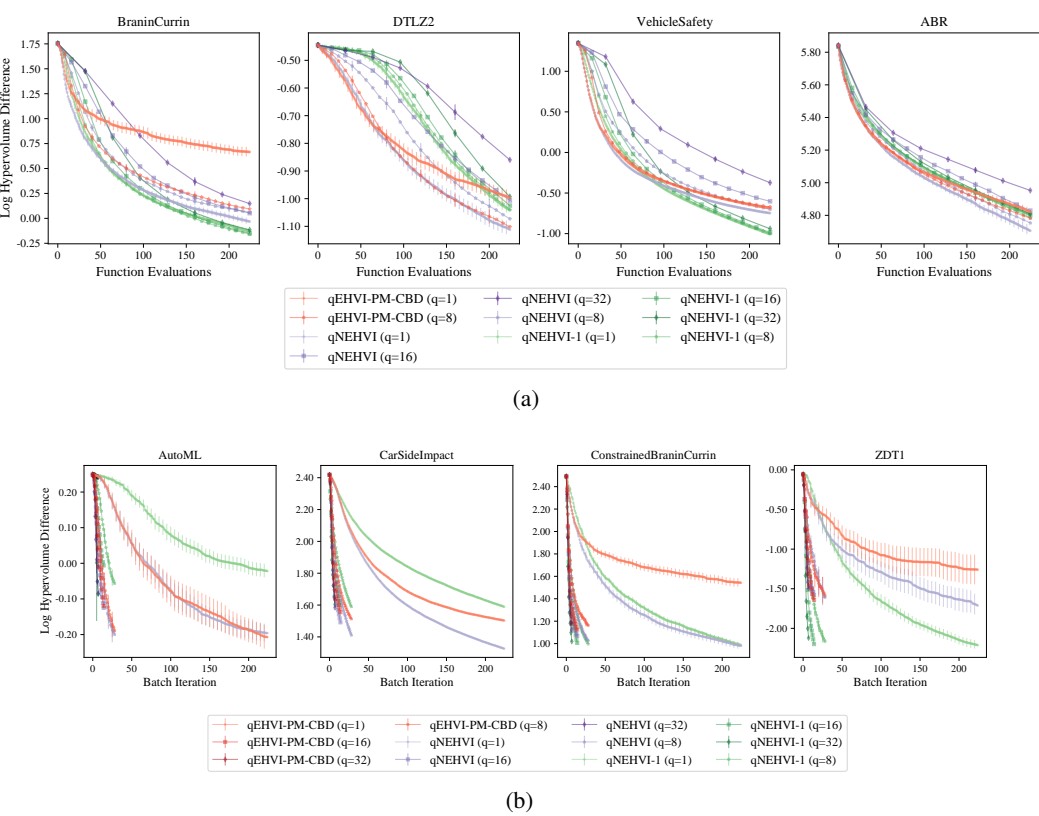

Figure 12: Anytime optimization performance of batch EHVI-based acquisition functions using various $q$ over the number of function evaluations.

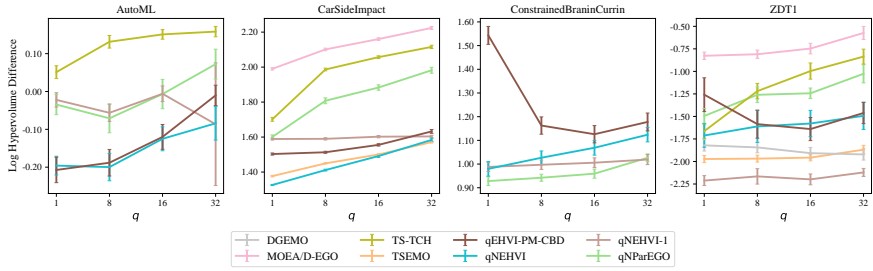

Figure 13: Final log hypervolume difference with various $q$ under a budget of 224 function evaluations. Smaller log hypervolume differences are better.

## H.6 Optimization Performance under Increasing Noise Levels

Figure 14 shows the sequential optimization performance of $q$NEHVI and $q$NEHVI-1 relative to $q$EHVI and $q$NParEGO under increasing noise levels. $q$NEHVI-1 achieves the best final hypervolume when the noise standard deviation $\sigma$ is less than $15\%$ of the range of each objective, but performs worse than $q$NEHVI earlier in the optimization. $q$NEHVI is the top performer in high-noise environments. We observe that all methods degrade as the noise level increases, however $q$NEHVI consistently exhibits excellent performance relative to other methods and only $q$NEHVI-1 is competitive and only in the low-noise regime.

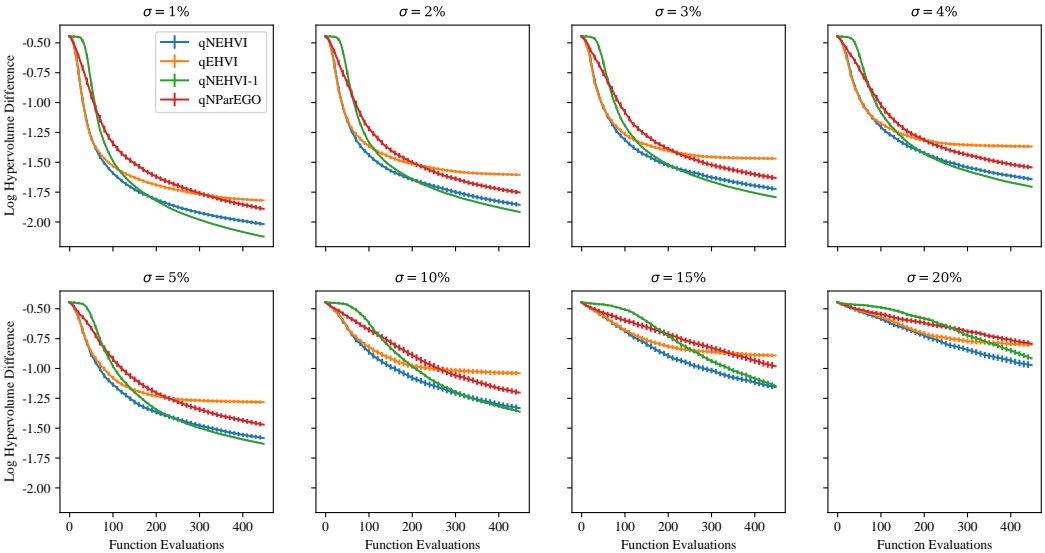

Figure 14: Sequential optimization performance under increasing noise levels on a DTLZ2 problem ($d = 6$, $M = 2$). $\sigma$ is the noise standard deviation, which we define as a percentage of the range of each objective over the entire search space. A noise level of $20\%$ is very high; for comparison, previous work on noisy MOBO has only considered noise levels of $1\%$ [24].

## H.7 Optimization Performance on Noiseless Benchmarks

We include a comparison of optimization performance on *noiseless benchmarks*. Figure 15 shows that $q$NEHVI performs competitively with $q$EHVI(-PM-CBD) and outperforms DGEMO, TS-TCH and $q$NParEGO across all benchmark problems. $q$NEHVI-1 is also a top performer on noiseless problems and both $q$NEHVI and $q$NEHVI-1 show little degradation in performance with increasing levels of parallelism.

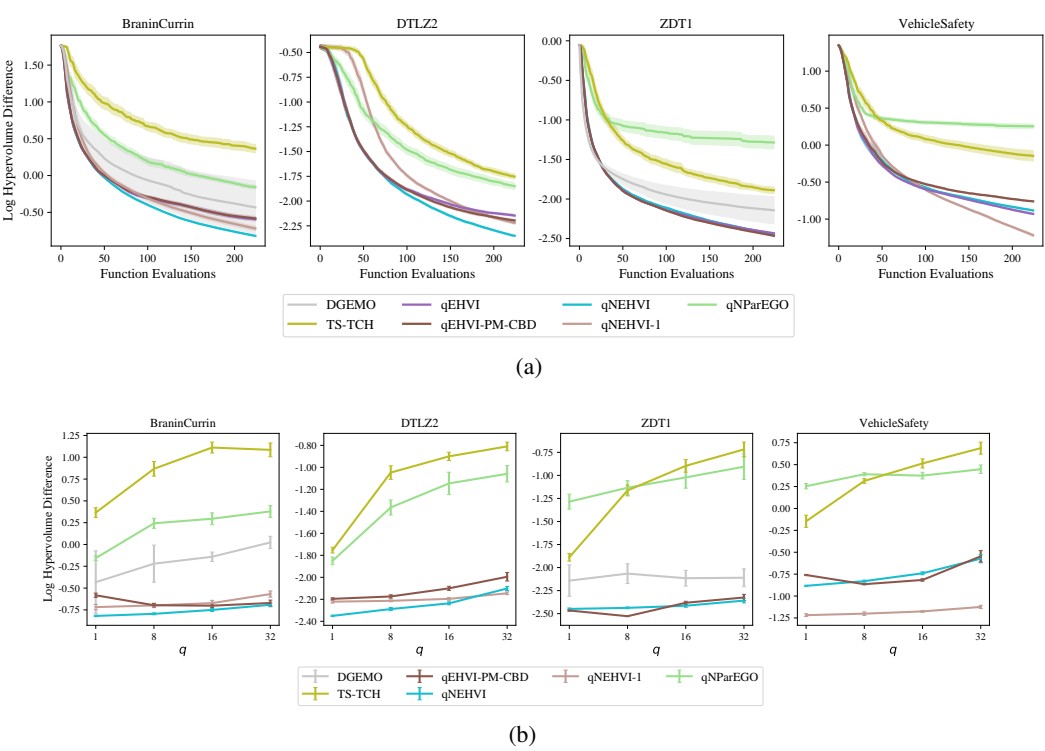

(a)

(b)

Figure 15: Sequential (a) and parallel (b) optimization Performance on *noiseless* benchmarks.

## H.8 Performance of $q$NEHVI-1 on 5-Objective Optimization

We demonstrate that $q$NEHVI-1 enables scaling to 5-objective problems. To our knowledge, no previous methods leveraging EHVI or HVI (e.g. DGEMO, TSEMO) considers 5-objective problems because of the super-polynomial complexity of the hypervolume indicator. Nevertheless, we show that using CBD and a single sample path approximation, $q$NEHVI-1 can be used for 5-objective optimization. As shown in Figure 16, $q$NEHVI-1 outperforms $q$NParEGO and Sobol search. $q$NEHVI-1 takes on average 73.53 seconds (with an SEM of 1.74 seconds) to generate each candidate, whereas $q$NParEGO takes 11.37 seconds (with an SEM of 0.97 seconds).

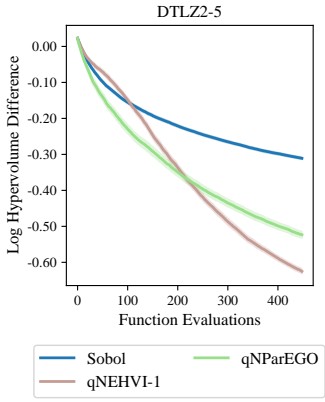

Figure 16: Optimization performance on a 5-objective DTLZ2 problem ($d = 6$) with $\sigma = 5\%$.

## H.9    Performance compared against a Multi-Objective CMA-ES

CMA-ES is an evolutionary strategy that is a strong method in single objective optimization, and many works have proposed extensions of CMA-ES to the multi-objective setting [26, 54]. We compare $q$NEHVI against the COMO-CMA-ES algorithm, which has been shown to outperform MO-CMA-ES on a variety of problems [54].[10]. We evaluate performance on the SphereEllipsoidal function from Bi-objective Black-Box Optimization Benchmarking Test Suite [6], and we add zero-mean Gaussian noise to each objective with $\sigma = 5\%$ of the range of each objective. We run COMO-CMA-ES with 5 kernels, the same initial quasi-random design as the BO methods, a population size of 10, and an initial step size of 0.2. As shown in Figure 17, the BO methods vastly outperform COMO-CMA-ES. $q$NEHVI and $q$NParEGO perform best and are closely followed by $q$NEHVI-1.

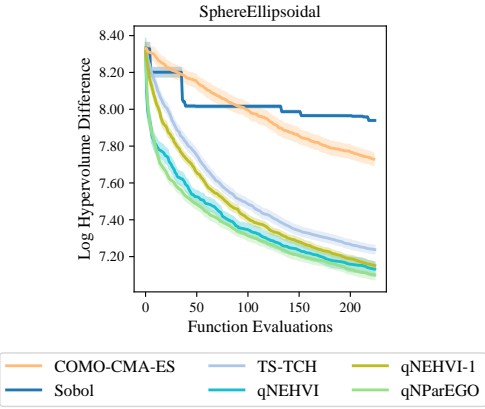

Figure 17:  Optimization performance on a 2-objective Sphere-Ellipsoidal problem ($d = 5$) with $\sigma = 5\%$.

## H.10    Importance of Accounting for Noise in DGEMO

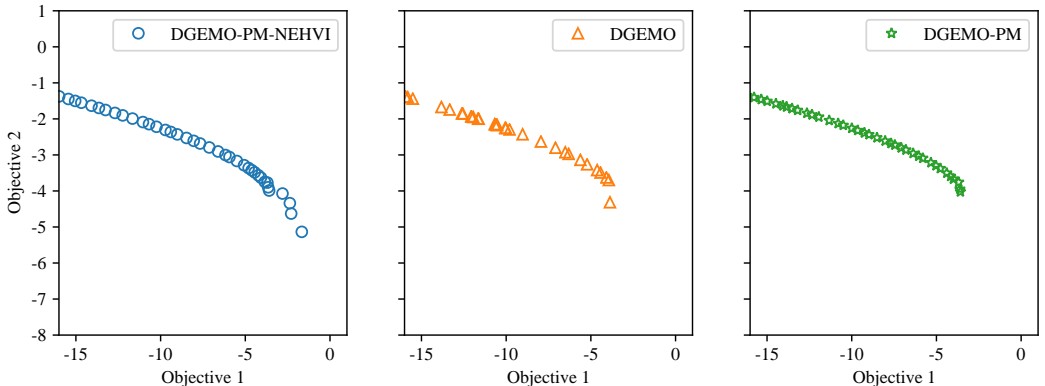

Figure 18:  An illustration of the effect of noisy observations on the true noiseless Pareto frontiers identified by DGEMO (right) DGEMO-PM-NEHVI (left, see Appendix H.10). Both algorithms are tested on a BraninCurrin synthetic problem, where observations are corrupted with zero-mean, additive Gaussian noise with a standard deviation of 5% of the range of respective objective. All methods use sequential ($q = 1$) optimization.

---

[10]COMO-CMA-ES is also the only multi-objective CMA-ES that we could find with an open-source Python implementation. We use the implementation available at `https://github.com/CMA-ES/pycomocma` under the BSD 3-clause license.

Similar to EHVI, DGEMO relies on the observed (noisy) Pareto frontier for batch selection. The right plot in Figure 18 shows that DGEMO exhibits the same clumping behavior in objective space in the noisy setting as EHVI. While DGEMO's diversity constraints (with respect to the input parameters) make it slightly more robust to noise, the solutions are clustered and the bottom right corner of the Pareto frontier is not identified. In an attempt to mitigate these issues, we propose an augmented version of DGEMO, which we call DGEMO-PM-NEHVI, as follows: (i) we use the posterior mean at the previously evaluated points to estimate the in-sample Pareto frontier, which we hope will improve robustness to noise when selecting a discrete set of potential candidates using the DGEMO's first-order approximation of the Pareto frontier, and (ii) we use $q$NEHVI rather than HVI under the posterior mean as the batch selection criterion over the discrete set, subject to DGEMO's diversity constraints. We find that using $q$NEHVI to integrate over the uncertainty in the Pareto frontier over the previously evaluated points, results in identifying higher quality Pareto frontiers as shown in Figure 18. We also include DGEMO-PM, which uses (i) but not (ii) for completeness. Not only does DGEMO-PM-NEHVI identify much more diverse solutions that provide better coverage across the Pareto frontier, but DGEMO-PM-NEHVI also identifies much better solutions on the lower right portion of the Pareto frontier than DGEMO and DGEMO-PM. DGEMO-PM performs much better than DGEMO and is competitively with DGEMO-PM-NEHVI, which we speculate is because DGEMO uses a first-order approximation of the Pareto frontier (using the observed values or using the posterior mean for -PM variants) to generate a discrete set of candidates. Using the posterior mean in this step is important for regularizing against extreme observed values due to noise. $q$NEHVI is only used as a filtering criterion for batch selection over that discrete set of candidates, subject to DGEMO's diversity constraints. Hence, $q$NEHVI has limited control over the batch selection procedure.

DGEMO's first-order approximation fundamentally does not account for uncertainty in the Pareto frontier over previously evaluated points. Although one could integrate over the uncertainty in the in-sample Pareto frontier by generating a first-order approximation of the Pareto frontier under different sample paths, the graph cut algorithm would yield different families under each sample path. It is unclear how to set the diversity constraints in that setting. We leave this for future work.

# I  Noisy Outcome Constraints

While the focus of this work is on developing a scalable parallel hypervolume-based acquisition function for noisy settings, our MC-based approach naturally lends itself to support for constraints.

## I.1  Derivation of Constrained NEHVI

The NEHVI formulation in (2) can be extended to handle noisy observations of outcome constraints. We consider the scenario where we receive noisy observations of $M$ objectives $\boldsymbol{f}(\boldsymbol{x}) \in \mathbb{R}^M$ and $V$ constraints $\boldsymbol{c}^{(v)} \in \mathbb{R}^V$, all of which are assumed to be "black-box": $\mathcal{D}_n = \{\boldsymbol{x}_i, \boldsymbol{y}_i, \boldsymbol{b}_i\}_{i=1}^n$ where $\begin{bmatrix} \boldsymbol{y}_i \\ \boldsymbol{b}_i \end{bmatrix} \sim \mathcal{N}\left( \begin{bmatrix} \boldsymbol{f}(\boldsymbol{x}_i) \\ \boldsymbol{c}(\boldsymbol{x}_i) \end{bmatrix}, \Sigma_i \right)$, $\Sigma_i \in \mathbb{R}^{(M+V) \times (M+V)}$. We assume, without loss of generality, that $\boldsymbol{c}^{(v)}$ is feasible iff $\boldsymbol{c}^{(v)} \geq 0$. In the constrained optimization setting, we aim to identify the a finite approximate feasible Pareto set

$$\mathcal{P}_{\text{feas}} = \{\boldsymbol{f}(\boldsymbol{x}) \mid \boldsymbol{x} \in X_n, \boldsymbol{c}(\boldsymbol{x}) \geq \boldsymbol{0}, \nexists \ \boldsymbol{x}' : \boldsymbol{c}(\boldsymbol{x}') \geq \boldsymbol{0} \ s.t. \ \boldsymbol{f}(\boldsymbol{x}') \succ \boldsymbol{f}(\boldsymbol{x})\}$$

of the true feasible Pareto set

$$\mathcal{P}_{\text{feas}}^* = \{\boldsymbol{f}(\boldsymbol{x}) \ s.t. \ \boldsymbol{c}(\boldsymbol{x}) \geq \boldsymbol{0}, \nexists \ \boldsymbol{x}' : \boldsymbol{c}(\boldsymbol{x}') \geq \boldsymbol{0} \ s.t. \ \boldsymbol{f}(\boldsymbol{x}') \succ \boldsymbol{f}(\boldsymbol{x})\}.$$

The natural improvement measure in the constrained setting is *feasible* HVI, which we define for a single candidate point $\boldsymbol{x}$ as

$$\text{HVI}_{\text{C}}(\boldsymbol{f}(\boldsymbol{x}), \boldsymbol{c}(\boldsymbol{x})|\mathcal{P}_{\text{feas}}) := \text{HVI}[\boldsymbol{f}(\boldsymbol{x})|\mathcal{P}_{\text{feas}}] \cdot \mathbb{1}[\boldsymbol{c}(\boldsymbol{x}) \geq \boldsymbol{0}].$$

Taking the expectation over $\text{HVI}_{\text{C}}$ gives the constrained expected hypervolume improvement:

$$\alpha_{\text{EHVI}_{\text{c}}}(\boldsymbol{x}) = \int \text{HVI}_{\text{C}}(\boldsymbol{f}(\boldsymbol{x}), \boldsymbol{c}(\boldsymbol{x})|\mathcal{P}_{\text{feas}})p(\boldsymbol{f}, \boldsymbol{c}|\mathcal{D})d\boldsymbol{f}d\boldsymbol{c} \tag{19}$$

For brevity, we define $\mathcal{C}_n = \boldsymbol{c}(X_n), \mathcal{F}_n = \boldsymbol{f}(X_n)$. The *noisy expected hypervolume improvement* is then defined as:

$$\alpha_{\text{NEHVI}_c}(\boldsymbol{x}) = \int \alpha_{\text{EHVI}_c}(\boldsymbol{x}|\mathcal{P}_{\text{feas}})p(\mathcal{F}_n, C_n|\mathcal{D}_n)d\mathcal{F}_n d\mathcal{C}_n. \tag{20}$$

Performing feasibility-weighting on the sample-level allows us to include such auxiliary outcome constraints into the full Monte Carlo formulation given in (3) in a straightforward way:

$$\hat{\alpha}_{\text{NEHVI}c}(\boldsymbol{x}) = \frac{1}{N}\sum_{t=1}^{N}\sum_{k=1}^{K_t}\left[\prod_{m=1}^{M}[z_{k,t}^{(m)} - l_{k,t}^{(m)}]_+ \prod_{v=1}^{V}\mathbb{1}[c_t^{(v)}(\boldsymbol{x}) \geq 0]\right]$$

where $z_{k,t}^{(m)} := \min\left[u_{k,t}^{(m)}, \tilde{f}_t^{(m)}(\boldsymbol{x})\right]$ and $l_{k,t}^{(m)}, u_{k,t}^{(m)}$ are the $m^{\text{th}}$ dimension of the lower and upper vertices of the rectangle $S_{k,t}$ in the non-dominated partitioning $\{S_{1,t}, ..., S_{K_t,t}\}$ under the feasible sampled Pareto frontier

$$\mathcal{P}_{\text{feas},t} = \mathcal{P}_{\text{feas}} = \{\tilde{\boldsymbol{f}}_t(\boldsymbol{x}) \mid \boldsymbol{x} \in X_n, \tilde{\boldsymbol{c}}_t(\boldsymbol{x}) \geq \boldsymbol{0}, \not\exists \ \boldsymbol{x}' : \tilde{\boldsymbol{c}}_t(\boldsymbol{x}') \geq \boldsymbol{0} \ s.t. \ \tilde{\boldsymbol{f}}_t(\boldsymbol{x}') \succ \tilde{\boldsymbol{f}}_t(\boldsymbol{x})\}.$$

In this formulation, the $\prod_{v=1}^{V}\mathbb{1}[c_t^{(v)}(\boldsymbol{x}) \geq 0]$ indicates feasibility of the $t$-th sample.

To permit gradient-based optimization via exact sample-path gradients, we replace the indicator function (which is non-differentiable) with a differentiable sigmoid approximation with a temperature parameter $\tau$, which becomes exact as $\tau \to \infty$:

$$\mathbb{1}[c^{(v)}(\boldsymbol{x}) \geq 0] \approx s(c^{(v)}(\boldsymbol{x}); \tau) := \frac{1}{1 + \exp(-c^{(v)}(\boldsymbol{x})/\tau)} \tag{21}$$

Hence,

$$\hat{\alpha}_{\text{NEHVI}c}(\boldsymbol{x}) \approx \frac{1}{N}\sum_{t=1}^{N}\sum_{k=1}^{K_t}\left[\prod_{m=1}^{M}[z_{k,t}^{(m)} - l_{k,t}^{(m)}]_+ \prod_{v=1}^{V}s(c_t^{(v)}(\boldsymbol{x}), \tau)\right]$$

## I.2 Derivation of Parallel, Constrained NEHVI

The constrained NEHVI can be extended to the parallel setting in a straightforward fashion. The joint constrained hypervolume improvement of a set of points $\{\boldsymbol{x}_i\}_{i=1}^{q}$ is given by

$$\text{HVI}_{\text{C}}(\{\boldsymbol{f}(\boldsymbol{x}_i), \boldsymbol{c}(\boldsymbol{x}_i)\}_{i=1}^{q}) = \sum_{k=1}^{K}\sum_{j=1}^{q}\sum_{X_j \in \mathcal{X}_j}(-1)^{j+1}\left[\left(\prod_{m=1}^{M}[z_{k,X_j}^{(m)} - l_k^{(m)}]_+\right)\prod_{\boldsymbol{x}' \in X_j}\prod_{v=1}^{V}\mathbb{1}[c^{(v)}(\boldsymbol{x}') \geq 0]\right].$$

and the constrained $q$EHVI is [10]:

$$\alpha_{q\text{EHVI}_c}(\mathcal{X}_{\text{cand}}|\mathcal{P}_{\text{feas}}) = \int \text{HVI}_{\text{C}}(\boldsymbol{f}(\mathcal{X}_{\text{cand}}), \boldsymbol{c}(\mathcal{X}_{\text{cand}})|\mathcal{P}_{\text{feas}})p(\boldsymbol{f}, \boldsymbol{c}|\mathcal{D}_n)d\boldsymbol{f}d\boldsymbol{c}$$

Hence, the constrained $q$NEHVI is given by:

$$\begin{aligned}\alpha_{q\text{NEHVI}c}(\mathcal{X}_{\text{cand}}) &= \int \alpha_{q\text{EHVI}_c}(\mathcal{X}_{\text{cand}}|\mathcal{P}_{\text{feas}})p(\mathcal{F}_n, C_n|\mathcal{D}_n)d\mathcal{F}_n d\mathcal{C}_n \\ &= \int \text{HVI}_{\text{C}}(\boldsymbol{f}(\mathcal{X}_{\text{cand}}), \boldsymbol{c}(\mathcal{X}_{\text{cand}})|\mathcal{P}_{\text{feas}})p(\mathcal{F}_n, C_n|\mathcal{D}_n)d\mathcal{F}_n d\mathcal{C}_n\end{aligned} \tag{22}$$

Using MC integration for the integral in (22), we have

$$\hat{\alpha}_{q\text{NEHVI}c}(\mathcal{X}_{\text{cand}}) = \frac{1}{N}\sum_{t=1}^{N}\text{HVI}_c(\tilde{\boldsymbol{f}}_t(\mathcal{X}_{\text{cand}}), \tilde{\boldsymbol{c}}_t(\mathcal{X}_{\text{cand}})|\mathcal{P}_{\text{feas},t}). \tag{23}$$

Under the CBD formulation, the constrained $q$NEHVI is given by

$$\begin{aligned}\hat{\alpha}_{q\text{NEHVI}}(\{\boldsymbol{x}_1, ..., \boldsymbol{x}_i\}) = &\frac{1}{N}\sum_{t=1}^{N}\text{HVI}_{\text{C}}\left(\{\tilde{\boldsymbol{f}}_t(\boldsymbol{x}_j), \tilde{\boldsymbol{c}}_t(\boldsymbol{x}_j)\}_{j=1}^{i-1}\} \mid \mathcal{P}_{\text{feas},t}\right) \\ &+ \frac{1}{N}\sum_{t=1}^{N}\text{HVI}_{\text{C}}\left(\tilde{\boldsymbol{f}}_t(\boldsymbol{x}_i), \tilde{\boldsymbol{c}}_t(\boldsymbol{x}_i) \mid \mathcal{P}_{\text{feas},t} \cup \{\tilde{\boldsymbol{f}}_t(\boldsymbol{x}_j), \tilde{\boldsymbol{c}}_t(\boldsymbol{x}_j)\}_{j=1}^{i-1}\}\right).\end{aligned} \tag{24}$$

As in (6), the first term is a constant when generating candidate $i$ and the second term is the NEHVI of $x_i$.

## J Evaluating Methods on Noisy Benchmarks

Given noisy observations, we can no longer compute the true Pareto frontier over the in-sample points $X_n$. Moreover, the subset of Pareto optimal designs $\mathcal{X}_n^* = \{x \in X_n, \nexists\, x' \in X_n \ s.t. \ f(x') \succ f(x)\}$ from the previously evaluated points may not be identified due to noise. For the previous results reported in this paper, we evaluate each method according to hypervolume dominated by the true unknown Pareto frontier of the noiseless objectives over the *in-sample* points. In practice, decision-makers would often select one of the in-sample points according to their preferences. If the decision maker only has noisy observations, selecting an in-sample point may be preferable to evaluating a new out-of-sample point according to the model's beliefs. An alternative evaluation method would be use the model's posterior mean to identify what it believes is the Pareto optimal set of in-sample designs. The hypervolume dominated by the true Pareto frontier of noiseless objectives over that set of selected designs could be computed and used for comparing the performance of different methods. Results using this procedure are shown in Figure 19. The quality of the Pareto set depends on the model fit. Several methods have worse performance over time (e.g. $q$EHVI and $q$PAREGO on the ZDT1 problem), likely due to the collection of outlier observations that degrade the model fit. Nevertheless, $q$NEHVI consistently has the strongest performance.

An alternative to the *in-sample* evaluation techniques described above would be to use the model to identify the Pareto frontier across the entire search space (in-sample or *out-of-sample*). For example, Hernández-Lobato et al. [24] used NSGA-II to optimize the model's posterior mean and identify the model estimated Pareto frontier. For benchmarking purposes on expensive-to-evaluate functions (e.g. in AutoML or ABR), this is prohibitively expensive. Moreover, such a method is less appealing in practice because a decision-maker would have to select out-of-sample points according to the posterior mean and then evaluate a set of preferred designs on the noisy objective to verify that the model predictions are fairly accurate at those out-of-sample designs. Therefore, in this work we evaluate methods based on the in-sample designs.

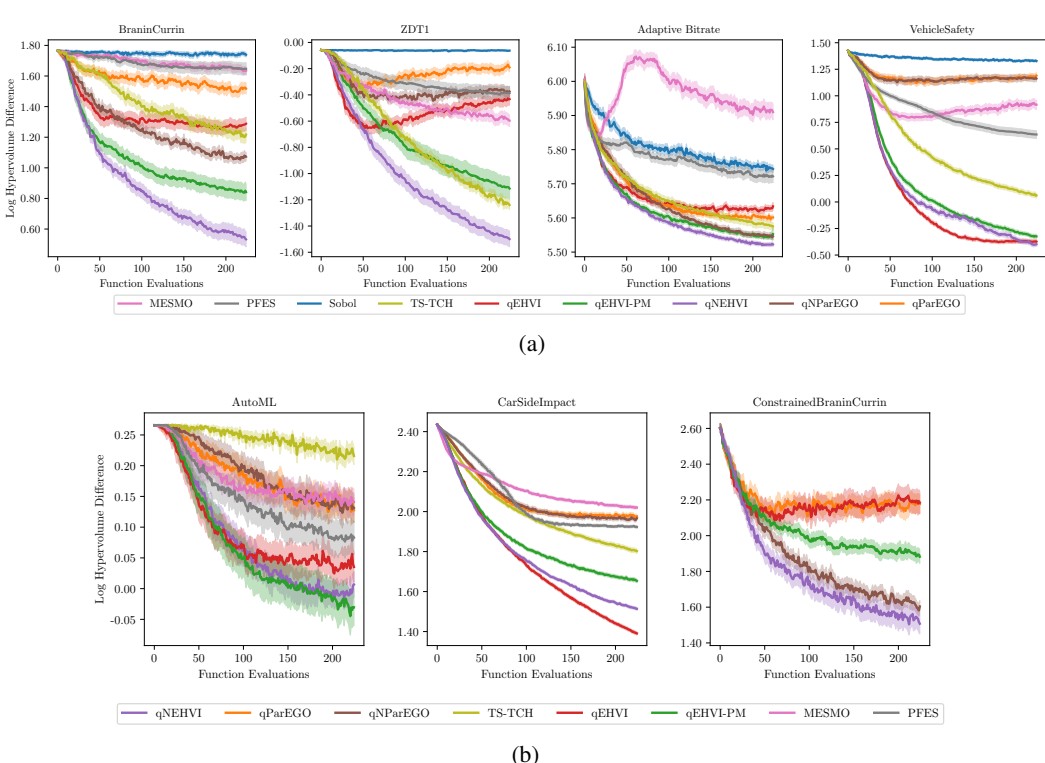

Figure 19: Sequential optimization performance using based on the model-identified Pareto set across in-sample points.