# OpenReview forum: "Parallel Bayesian Optimization of Multiple Noisy Objectives with Expected Hypervolume Improvement"
_NeurIPS.cc/2021/Conference — NeurIPS 2021 Poster_

### Official Review · Reviewer_dGcJ · 2021-07-02

**Rating:** 7
**Confidence:** 3

**Summary:**

The authors describe an BO algorihm for multi-objective optimization using hypervolume improvement. The main contribution is the incorporation of noise into the estimation of the improvement measure. The algorithm is evaluated on four functions and theoretical analysis is performed. In the experiments, the algorithm performed best on the set of functions.

**Limitations And Societal Impact:**

limitations are discussed.

**Main Review:**

strengths:
- Straight forward idea that is easy to understand
- Theoretical and empirical analysis

weaknesses
- The evaluation section features significantly more algorithms than objective functions that are evaluated.
- Some of the literature/contributions in ES-based MOO are missing
- In parts the paper feels like it is using much too much text to present a small(but neat!) idea.

Details:
- I would have liked to see some more functions to evaluate. An easy target would have been parts of the BBOB-biobj benchmark (with and w/o artificial simulated noise). The noise-free version would give an indication of how well the algorithm is positioned within the overall segment and would enable comparisons to strong noise-free baselines like the UP-MO-CMA-ES. This would also give strong indications of how more difficult the functions become under the chosen level of noise.

- The idea of doing greedy optimization for hypervolume improvement in section 5.1 is not new. The MO-CMA-ES is doing that (under the name hypervolume contribution, see[*]) to select which subsets of point to keep in the pareto front. Otherwise the adaptation to the noisy case seems to follow straight forward from the approach.

- I feel that the presentation could be improved and shortened a lot. e.g. the rhs of (3) feels unnecessary since it just spells out how the HVI is computed by the algorithm. The paragraph "Conditional Posterior Sampling" reads like standard optimization techniques used to acquire GP samples and has nothing to do with CBD. 5.1 similarly seems to be completely independent of CBD and any algorithm for computing the HVI could be inserted here. Moreover, it feels like the novel core of 5.1 is just a short addendum to 4.2. This is a bit of a shame because some of the "meat" of the paper has been moved to the appendix to make space for these computational basics which do not serve the presentation of the core idea.

I do think the idea is good and if i have not missed any important prior work, it does present a good contribution to the field (even though I am not an expert in the BO side of MOO). But I think, the authors should take some time to add a few more results and to slightly clean up the paper to make the core idea more prominent.

Note: Contrary to what the checklist states, i could not find any mention of compute time spent on the experiments. There is a statement of higher efficiency i have found, but no measurement of approximate compute resources required to repeat the experiments. If I missed it, maybe the authors can provide me a hint to where it is written.

[*]Thomas Voß, Nikolaus Hansen, Christian Igel: Improved Step Size Adaptation for the MO-CMA-ES



**Time Spent Reviewing:**

3

---

> ### Author Response · Authors · 2021-08-10
> **To Reviewer dGcJ**
>
> We strive to develop simple, accessible, and practically relevant ideas that can contribute to further work in the field, and find your recognition to be very encouraging. We appreciate your perspective as an expert in MOO working outside of the BayesOpt field. The number of benchmarks we include with our paper is commensurate and in many cases above the number considered in MOBO papers published at NeurIPS/ICML.
>
> > The evaluation section features significantly more algorithms than objective functions that are evaluated.
>
> The number of methods we compare against is quite extensive, and in nearly all cases we show superior anytime and final optimization performance. We view the number of other methods considered to be a strength, rather than weakness, and would like to better understand your concerns regarding considering more methods than benchmarks (we included 7 problems in the submission).
>
> > I would have liked to see some more functions to evaluate. An easy target would have been parts of the BBOB-biobj benchmark (with and w/o artificial simulated noise). The noise-free version would give an indication of how well the algorithm is positioned within the overall segment and would enable comparisons to strong noise-free baselines like the UP-MO-CMA-ES. This would also give strong indications of how more difficult the functions become under the chosen level of noise.
>
> Nonetheless, we are happy to include more benchmarks in the appendix (or potentially some summary statistics of a meta-analysis of the results for BBO-biobj, which we were previously unfamiliar with). If you have any suggestions for specific benchmark problems to include, we can include them in the final version of our work.
>
> Thank you for pointing us to the work on MO-CMA-ES by Voß, Hansen, and Igel. As far as we can tell, no MOBO papers have compared against MO-CMA-ES before, perhaps because it is designed for very large numbers of function evaluations. While our time to test with many new benchmarks during the rebuttal period is limited, we will add a test problem from BBO-biobj to the final manuscript to compare against MO-CMA-ES. We reviewed the BBO-biobj codebase (written in C) and time did not permit us to verify that a reimplementation in our benchmarking code of the 15 problem instances for a BBO-biobj function matched the instances used in the original C code.
>
> The reviewer also asks about how the performance varies with noise levels. In the supplementary material (Appendix H.6) we show how mild to large levels of noise affect optimization performance on a 2-objective DTLZ2 problem ($d=6$). qNEHVI consistently has better anytime performance relative to both qEHVI and qNParEGO across all noise levels. We can include additional algorithms or ranges of noise if the reviewer does not find this adequate. Finally, the reviewer may be interested in examining the performance of various acquisition functions under the noiseless case in Appendix H.7.
>
> Thank you for the feedback on the presentation and writing, which we address below.
>
> > I feel that the presentation could be improved and shortened a lot. e.g. the rhs of (3) feels unnecessary since it just spells out how the HVI is computed by the algorithm.
>
> We will move the RHS from the main text to the appendix as this notation is cumbersome and not used elsewhere in the main text.
>
> > The paragraph "Conditional Posterior Sampling" reads like standard optimization techniques used to acquire GP samples and has nothing to do with CBD.
>
> We believe that discussing conditional posterior sampling is important because it is essential that the base samples used for in-sample points must be fixed for the CBD approach to work. If the samples were to change, the box decompositions would need to be re-computed. As far as we know, this has not been discussed in the literature. We feel that describing how valid joint samples from the posterior over f(X, X_n) can be drawn is an important detail, but if the reviewers agree that this is something that is better described in the appendix, we would be happy to consider moving these details to the appendix to create more space in the main text.
>
> > The idea of doing greedy optimization for hypervolume improvement in section 5.1 is not new. The MO-CMA-ES is doing that (under the name hypervolume contribution, see[*]) to select which subsets of point to keep in the pareto front.
>
> This is true; even greedy optimization of EHVI is not new. But efficient batch selection using CBD requires sequential greedy batch selection. Sequential greedy batch selection with CBD enables using EHVI with large batch sizes by vastly improving the computational complexity.
>
> > 5.1 similarly seems to be completely independent of CBD and any algorithm for computing the HVI could be inserted here.
>
> We strongly disagree with this statement. S5.1 describes how CBD can be used with sequential greedy batch selection to reduce the complexity of batch EHVI from exponential in the batch size (using the IEP) to polynomial in the batch size. Although we introduce CBD as a computational approach for evaluating NEHVI, it is the crucial component that allows us to perform vastly more efficient sequential greedy batch selection. This section is a core contribution of the paper. IEP and CBD are, as far as we know, the only two differentiable methods for sequential greedy batch selection. One could use another method (e.g. Fonseca et al. 2006 or Russo et al 2012) for computing HVI under each posterior sample, but that would not be differentiable (rendering optimization much harder), and even computing HVI would likely be much slower because alternative algorithms have not been shown to be able to leverage highly parallelized computation (i.e. computing HVI under all posterior samples using SIMD parallelism).
>
> > This is a bit of a shame because some of the "meat" of the paper has been moved to the appendix to make space for these computational basics which do not serve the presentation of the core idea.
>
> Can you please clarify what parts you would recommend moving from the appendix to the main text, if we were to consolidate the main text and free up some space? We value the reviewers’ advice on how to best present the material to readers with fresh eyes.

---

> > ### Author Response · Authors · 2021-08-19
> > **Response to Reviewer dGcJ: follow-up**
> >
> > Hi Reviewer dGcJ, we would be grateful if you can confirm whether our response has addressed your concerns. To recap our response, we:
> >
> > 1. Addressed the question of how qNEHVI performs under various noise levels by providing additional results in low-noise ($\sigma=1-4$% of the range of each objective) regimes (see response to Reviewer 6KXs) and referring you to our existing study of various noise levels ($\sigma=5-30$% of the range of each objective) in Appendix H.6 and noiseless benchmarks in Appendix H.7. We find that our method performs best in all noise settings.
> > 2. Stated that we will add problems from BBO-biobj in our revision.
> > 3. Clarified why CBD is crucial for efficient batch candidate selection
> > 4. Asked for clarification on what "meat" from the appendix you would like to see in the main text.
> >
> > Please let us know your thoughts on (4) and if any issues remain. Thanks!

---

### Official Review · Reviewer_LnaP · 2021-07-08

**Rating:** 6
**Confidence:** 5

**Summary:**

The authors propose a version of a parallel Expected Hypervolume Improvement acquisition function for multi-objective Bayesian optimization. They show improved scaling over existing methods on a variety of benchmarks.

**Limitations And Societal Impact:**

Yes

**Main Review:**

The technical contribution is incremental but solid. Some claims must be clarified, as well as highlighting when simplifications are done for speeding up computations (e.g., not just in appendix). The empirical results are strong.

Detailed comments

In the abstract, the authors “argue that, even in the noiseless setting, the problem of generating multiple candidates in parallel can be reduced to that of handling uncertainty in the Pareto frontier.” but the meaning is unclear in the rest. Note that there are works on quantifying the uncertainty in the Pareto front, see e.g.,
- Calandra, R., Peters, J., & Deisenrothy, M. (2014). Pareto front modeling for sensitivity analysis in multi-objective bayesian optimization. In NIPS Workshop on Bayesian Optimization (Vol. 5).
- Binois, M., Ginsbourger, D., & Roustant, O. (2015). Quantifying uncertainty on Pareto fronts with Gaussian process conditional simulations. European journal of operational research, 243(2), 386-394.
In addition, this is seemingly in contradiction with the beginning of Section 5 where sampled Pareto fronts are not updated based on the batch points (in the noisy case).

It is referred several times to the “exponential” complexity of the batch variant, presumably related only to the method in [7]. This should be clarified.

In section 4, it is written “To our knowledge, all previous work on EHVI assumes that observations are noiseless” while there is a dedicated paragraph starting line 128, and just below, line 182, is the obvious fix. Note that reinterpolation is another commonly used option in practice, see, e.g., Forrester, A., Sobester, A., & Keane, A. (2008). Engineering design via surrogate modelling: a practical guide. John Wiley & Sons.


Minor points:
- Figure 1: Add true Pareto front and reference point
- Page 1, line 66: what is a “high-quality implementation”?

**Time Spent Reviewing:**

5

---

> ### Author Response · Authors · 2021-08-10
> **To Reviewer LnaP**
>
> We thank the reviewer for their careful review and will work toward improving the precision and clarity of our presentation. We appreciate that the reviewer finds the technical contributions and empirical results to be solid. Regarding the novelty of the paper, we refer the reviewer to our general response to all reviewers above.
>
> > In the abstract, the authors “argue that, even in the noiseless setting, the problem of generating multiple candidates in parallel can be reduced to that of handling uncertainty in the Pareto frontier.” but the meaning is unclear in the rest
>
> We apologize for lack of clarity with this statement, and we will clarify this statement in our final manuscript. We describe what we mean by this in more detail in S5.1 (around Eq 6). Our method accounts for uncertainty in the function values at the in-sample (i.e. previously evaluated) points due to noise observations. Since the function values are unknown the true Pareto frontier over the true function values at the evaluated designs is also unknown. By integrating over the posterior distribution for the in-sample function values, we are accounting for uncertainty in the Pareto frontier. The problem of generating a batch of new candidates under sequential greedy selection requires handling a very related problem: accounting for uncertainty in the function values corresponding to pending points (i.e. the previously selected points in the batch). This too is an incarnation of handling uncertainty in the Pareto frontier.
>
> > Related work
>
> We thank the reviewer for pointing us to the Calandra et al. and Binois et al. papers, and we will include these references in our final draft. The Calandra paper does an excellent job of motivating the problems posed by noisy observations when attempting to characterize Pareto frontiers. The Binois paper gives an excellent overview of uncertainty around Pareto frontiers and how it can be quantified and monitored during the optimization process. However, unlike either of these works, we provide a solution to performing MOBO in noisy settings, rather than purely reasoning about the uncertainty in the Pareto frontier. This extension is non-trivial, and requires a lot of care to make it scalable for practical settings.
>
> > In addition, this is seemingly in contradiction with the beginning of Section 5 where sampled Pareto fronts are not updated based on the batch points (in the noisy case).
>
> We state in L270 that the Pareto frontiers and box decompositions are updated based on the already selected points in the batch. Can you clarify the source of your confusion please?
>
> > It is referred several times to the “exponential” complexity of the batch variant, presumably related only to the method in [7]. This should be clarified.
>
> To our knowledge, [7] is the only paper that derives true (i.e. characterizing the expected increase in HV under the joint distribution of the q candidates) batch EHVI (although other prior works like DGEMO, TSEMO, or Yang et al. 2019, propose heuristic approaches for producing batches using HVI-based criteria). CBD alleviates this fundamental limitation for batch EHVI and enjoys polynomial complexity with respect to the batch size.
>
> > In section 4, it is written “To our knowledge, all previous work on EHVI assumes that observations are noiseless”...
>
> We will clarify our statement about prior work on EHVI assuming observations are noiseless, since using the model’s posterior mean (or using re-interpolation, which we discuss in L130) does not assume noiseless observations. This statement is intended to say that prior work does not directly account for the noise at the level of the acquisition function itself.
>
> > Minor points
>
> Thanks. We will add the true Pareto frontier and reference point to Figure 1. By "high-quality" implementation, we mean that our code has 100% unit test coverage, strict typing, thorough documentation, support for modern ML frameworks (including GPU and parallel SIMD operations), and will be made available in open-source. In addition, we implemented SoTA box decomposition algorithms for EHVI and PFES. Many previously MOBO works (e.g. the papers corresponding DGEMO, PFES, MESMO, PESMO, TS-TCH) that compare against EHVI use naive brute-force box decomposition algorithms, which yields poor wall times (orders of magnitude slower than ours) for EHVI. [7] uses the box decomposition algorithm from [6] that is vastly outperformed by the SoTA [26, 29, 53]. While there have been significant advances in and many papers on efficient box decomposition algorithms (e.g. [26, 29, 53]), the authors of those works have primarily released C and MatLab code, which is difficult to leverage with many of the popular based BO libraries, which tend to be implemented in python (e.g., GPyOpt. BoTorch, Spearmint, etc). Although implemented in Python, our code leverages SIMD parallelism where possible.

---

> > ### Author Response · Authors · 2021-08-19
> > **To Reviewer LnaP: follow-up**
> >
> > Hi Reviewer LnaP, we would be grateful if you can confirm whether our response has addressed your concerns. To recap our response, we:
> >
> > 1. Responded to concerns about novelty (see comment to All Reviewers).
> > 2. Addressed confusion about CBD, its complexity, and the relationship between accounting for uncertainty in the Pareto frontier and parallel candidate generation. We are thankful that you have pointed out these (and other) points of confusion, and we will clarify these points in our revision.
> > 3. Reviewed the two papers that consider uncertainty in Pareto frontiers that you mentioned and described how those works are related, but that our work tackles a different problem of performing optimization under noisy observations.
> >
> > Please let us know if any issues remain. Thanks!

---

> > > ### Comment · Reviewer_LnaP · 2021-08-24
> > > **Follow-up**
> > >
> > > Dear authors,
> > >
> > > Thank you very much for the reply and clarifications.
> > >
> > > Concerning the confusing point, it is just that L231-232 the samples are said to be fixed and computed once, while L271 they are said to be updated. This can be made clearer with the proposed additional details on Pareto frontier uncertainty. (Also perhaps this is an opportunity to reference Appendix C.1 and C.3?)
> > >
> > > Best regards

---

> > > > ### Author Response · Authors · 2021-08-27
> > > > **Re: Follow-up**
> > > >
> > > > Dear Reviewer LnaP,
> > > >
> > > > Thank you for taking the time to help us understand the point of confusion.
> > > >
> > > > When we say that the samples are said to be fixed and computed once in L231-232, we are referring to computing the sequential acquisition function NEHVI with $q=1$, not the parallel variant qNEHVI with $q >1$. We begin the sentence L231 by saying that this is explicitly for evaluating the integral in (2), which is sequential NEHVI. However, we see how this could be confusing since the previous section (4.2) presents the parallel variant, qNEHVI, and the reader must look at equation (2) to see that we are talking about sequential NEHVI. We will make this clearer in our revision by explicitly stating that this statement applies to optimizing NEHVI, rather than selecting a batch of $q$ points with the parallel variant qNEHVI.
> > > >
> > > > When we say that the box decompositions are update in L271, this is in the context of optimizing a batch of $q$ points using a sequential greedy approach. Optimizing qNEHVI in a sequential greedy fashion involves optimizing a sequence of NEHVI ($q=1$) acquisition functions where the in-sample Pareto frontier, over function values at the previously evaluated points and new points in the batch that have been selected $f(X_n, x_1, ..., x_{i-1})$, changes at each iteration $i=1, ..., q$ as new candidate points are selected. For selecting any individual candidate $x_i$, the box decompositions of the Pareto frontier over the sampled function values of $f(X_n, x_1, ..., x_{i-1})$ are fixed while $x_i$ is optimized, but the box decompositions are updated after $x_i$ is selected. In our revision, we will clarify this point and highlight that for selecting any individual candidate $x_i$, we are optimizing NEHVI with an uncertain Pareto frontier over $f(X_n, x_1, ..., x_{i-1})$. Additionally, we will reference Appendix C2 to provide more details for the reader. Appendix C1 is about the IEP formulation and Appendix C3 is about NEHVI-1, so we believe that referencing C2 would be most useful.

---

### Official Review · Reviewer_35tW · 2021-07-16

**Rating:** 6
**Confidence:** 5

**Summary:**

The paper proposes a novel method to perform multi-objective BO based on Expected HyperVolume Improvement. Specifically, the authors argue that taking into account noisy observations and dealing with that noise improves the optimisation as they introduce a new acquisition function NEHVI. They moreover introduce a parallelisation framework enabling scalable batch acquisition in their novel acquisition function. This paper is well written. The ability to use EHVI in a noisy environment is useful in practice and in my opinion, the improvement in scalability is important for the community. However, I would have liked to see experiments led with varying numbers of objectives as it is known that it’s a main bottleneck of EHVI in practice.

**Limitations And Societal Impact:**

Yes

**Main Review:**

1) The authors interestingly avoid having to decompose their hypervolumes at each iteration using a cache mechanism. I think this is a good direction and really relevant. What is not clear to me, however, is what order of error is introduced using the above mechanism? Can the authors please elaborate a bit on this in their rebuttal?

2) Is the decomposition fixed for each BO step? If so, what is the impact on estimating (2)? It would be interesting to see such an analysis.

3) It is mentioned by the authors that IEP and CBD are mathematically equivalent but can CBD - while producing the exact same acquisition value for a set of points, is significantly cheaper – scale to as many objectives and what is the cost of using CBD over IEP in practice, maybe in terms of convergence of the acquisition optimisation apart from purely memory & wall-time?

4) Related to the above, Theorem 1 relates to NEHVI but not qNEHVI. What is the cost of SAA in that case on the convergence during the acqusition Optimisation?

5) There are many benchmarks in this paper which is great. However, I would really like to see the performance as a function of an increase in the number of objectives as well as in terms of higher dimensions. Can the authors perform such an analysis?

Related Work:
I think the paper covers the most relevant related work. I do, however, think include and benchmarking against the following can aid in further improving the paper:
1. http://proceedings.mlr.press/v80/lyu18a/lyu18a.pdf
2. https://arxiv.org/pdf/2012.03826.pdf


**Time Spent Reviewing:**

3

---

> ### Author Response · Authors · 2021-08-10
> **To Reviewer 35tW**
>
> We thank the reviewer for their encouraging remarks about the practical contributions, as well as the contributions in terms of scalability.
>
> > The authors interestingly avoid having to decompose their hypervolumes at each iteration using a cache mechanism. [..] What is not clear to me, however, is what order of error is introduced using the above mechanism?
>
> There is no approximation error from using CBD. Computing HVI of a single new candidate using CBD is exactly equivalent to IEP and both yield exact computations of HVI. Hence, under sequential greedy candidate selection, the joint HVI from candidates $\mathbf{x}_1, …, \mathbf{x}_i$ for $i=1,..., q$ using the IEP is exactly the same value as HVI computed using CBD. However, the way that the HVI is computed is different. CBD has polynomial complexity w.r.t to $i$ when selecting candidate $i$, whereas the IEP has exponential complexity. Using the IEP, [7] found that sequential greedy batch selection yielded better optimization performance than jointly optimizing $q$ candidates because joint optimization involves a much harder high-dimensional optimization problem. The only error in evaluating NEHVI comes from MC approximation and affects both CBD and IEP formulations. In addition, we provide an theoretical error bound in Appendix F on the difference between (i) the qNEHVI of the set of candidates selected by maximizing the qNEHVI in a sequential greedy fashion, and (ii) the qNEHVI of the set of candidates that has the maximum joint qNEHVI. Our result holds for both qNEHVI and it’s MC estimator as shown in Appendix F.
>
> > Is the decomposition fixed for each BO step? If so, what is the impact on estimating (2)? It would be interesting to see such an analysis.
>
> Box decompositions are computed and cached once per candidate. If $q=1$, this is once per BO iteration, If $q>1$, then the box decompositions are updated after each candidate $\mathbf{x}_i$ for $i=1, …, q-1$ is selected to integrate over the unknown function values for the selected candidate. As described above, there is no approximation error compared to IEP when using a sequential greedy approach, and we provide a theoretical bound on the error from the sequential greedy approach in Appendix F.
>
> > Theorem 1 relates to NEHVI but not qNEHVI. What is the cost of SAA in that case on the convergence during the acqusition Optimisation?
>
> This is incorrect. Theorem 1 applies to both NEHVI and qNEHVI (as noted in L298) and the proof of Theorem 1 in Appendix E is of the general $q>=1$ case. Therefore, there is no additional cost in terms of convergence under the SAA between NEHVI and qNEHVI.
>
>
> > I would really like to see the performance as a function of an increase in the number of objectives as well as in terms of higher dimensions. Can the authors perform such an analysis?
>
> We will include an analysis of how performance is affected by the number of objectives with comparisons to other AFs (see response to reviewer 6KXs). In Appendix H.5, we compare qNEHVI with qNEHVI-1 as the dimension of the search space increases. For a more comprehensive analysis, we add additional comparisons with standard MOBO methods under sequential optimization on DTLZ2 with 2 objectives and dimensions ranging from 5 to 20. Means and 2 SEMs of the HV after 224 evaluations over 100 replications are provided below:
>
> **HV**
> * $d=5$: qNEHVI: 0.381 (+/-0.003), qNParEGO: 0.384 (+/-0.002), TS-TCH: 0.379 (+/-0.001)
> * $d=10$: qNEHVI: 0.311 (+/-0.004), qNParEGO: 0.245 (+/-0.01). TS-TCH: 0.206 (+/-0.008)
> * $d=15$: qNEHVI: 0.19 (+/-0.014), qNParEGO: 0.092 (+/-0.008), TS-TCH: 0.001 (+/-0.001)
> * $d=20$: qNEHVI: 0.073 (+/-0.014), qNParEGO: 0.043 (+/-0.008), TS-TCH: 0.0 (+/-0.0)
>
> Optimization performance degrades for all methods as the dimensionality of the search space increases (with a fixed evaluation budget), but qNEHVI is consistently the top performer and the benefit of qNEHVI is even more pronounced in higher-dimensional search spaces.
>
> > Related Work: I think the paper covers the most relevant related work. I do, however, think include and benchmarking against the following can aid in further improving the paper:
>
> Thanks for pointing out this interesting work by Lyu and the subsequent pre-print that builds on that work. While the MACE approaches leverage MOO in its acquisition function as a way of “hedging” or ensembling single-objective acquisition functions, it is not used to actually solve multi-objective optimization problems, and is therefore not directly applicable to our problem setting. We will cite this work and mention that combining NEHVI and other methods using their proposed ensembling approach would be an interesting area for future work, but investigation into this direction is beyond the scope of this work.

---

> > ### Author Response · Authors · 2021-08-19
> > **To Reviewer 35tW: follow-up**
> >
> > Hi Reviewer 35tW, we would be grateful if you can confirm whether our response has addressed your concerns. To recap our response, we:
> >
> > 1. Clarified that CBD does not introduce additional error and articulated how and when box decompositions are fixed and cached
> > 2. Clarified that the SAA convergence result applies qNEHVI (the proof is the general $q>=1$ case) with no additional cost convergence cost.
> > 3. Demonstrated that qNEHVI-1 performs best (in terms of hypervolume) on a 6-objective problem (see response to Reviewer 6KXs) with reasonable wall times and that qNEHVI scales better than other baselines (with respect to hypervolume) as the dimensionality of the search space increases from 5 to 20.
> >
> > Please let us know if any issues remain. Thanks!

---

> > > ### Comment · Reviewer_35tW · 2021-08-19
> > > **Response**
> > >
> > > I thank the authors for the response that did address my concerns.

---

### Official Review · Reviewer_6KXs · 2021-07-28

**Rating:** 6
**Confidence:** 4

**Summary:**

The paper focuses on the setting of multi-objective Bayesian Optimization (BO), when observations are corrupted by noise. Existing approaches only consider the uncertainty in the new point where the function is being evaluated, but consider the Pareto front from prior observations to be fixed. This work, on the other hand, additionally assumes uncertainty in the Pareto front - this is a natural assumption when observations contain noise. The authors introduce a new acquisition function, NEHVI, which extends the classical expected hypervolume improvement (EHVI) criterion. They also derive a new parallel variant with polynomial complexity, qNEHVI, which can replace the more complex qEHVI variant; this is made possible by re-using cached box decompositions. Various experiments demonstrate strong performance and competitive wall-times.

**Limitations And Societal Impact:**

The authors discuss openly the limitations of their work in Section 8 (Discussion Section).
The authors discuss the potential societal impact of their work in detail in Appendix A.

**Main Review:**

Strengths of this work
- The work is very well motivated. Assuming uncertainty in the Pareto front is not just desirable but also required, in the standard BO setting where observations contain noise.
- The idea of caching the box decompositions is a central contribution of this work. This idea allows the efficient (polynomial) computation of qNEHVI (or qEHVI), which would otherwise be intractable (exponential) is we use the standard inclusion-exclusion principle (IEP).
- The experiments are quite comprehensive, in the sense that they include several competitors from prior literature and a number of synthetic and real benchmarks. qNEHVI with cached box decompositions performs consistently well in terms of Pareto front quality and wall time.
- The CBD-based algorithm offers very visible benefits in terms of execution time compared to the IEP-based algorithm. The latter cannot even handle batch sizes beyond a certain size (e.g., 16 or 32).
- The work seems to be technically sound, and it contains a rigorous sample average approximation convergence result for NEHVI.
- The paper is well written.

Concerns about this work
- My major concern is that this work is more or less an extension of (Daulton, Balandat and Bakshy 2020). That work already introduces some important concepts: (i) the q-expected hypervolume improvement (qEHVI), (ii) box decompositions for the computation of hypervolume improvement, (iii) computing q-HVI with the inclusion-exclusion principle, (iv) (quasi-)MC methods for computing qEHVI, (v) differentiability of the MC-estimator qEHVI, and (vi) sample average approximation convergence results for qEHVI. The major contributions of the current work compared to (Daulton, Balandat and Bakshy 2020) are in my view:
  - Assuming noise/uncertainty in the Pareto front when computing the expected hypervolume improvement.
  - Caching the box decompositions to avoid computing the Pareto front and performing an expensive box decomposition every time we draw new samples (this is a very central idea in this work).
  - Algorithm for efficient sequential greedy batch selection using cached box decompositions (polynomial complexity versus exponential with inclusion-exclusion principle).

The idea of caching the box decompositions is certainly interesting and has the potential to improve efficiency, but overall the novelty is not very strong. To me, this work seems to be mostly about efficient computation of  qEHVI (or the more general qNEHVI).
- Related to the point above, the convergence result is very reminiscent of the convergence proof in (Daulton, Balandat and Bakshy 2020).
- The scope of this work is a bit unclear. On the one hand, this work is about extending EHVI or qEHVI with noise. But on the other hand, it is about efficient computation of both the noiseless and noisy variants. The efficient batch selection algorithm could even be applicable in the original noiseless setting.
- The asymptotic complexities for CBDs or IEP depend exponentially on the number of objectives. This means that even with the efficient algorithm proposed in this work, multi-objective BO over several objectives could be prohibitively expensive. On the other hand, for up to 4 objectives approaches based on trust regions (e.g., (Eriksson et al. 2019) or (Eriksson and Poloczek 2021)) might be able to handle high-D inputs and multiple constraints better. Furthermore, Thompson sampling for evaluation of candidate points could lead to high efficiency. It is not clear how this line of work compares to such approaches in terms of solution quality or efficiency.

Additional questions to authors
- Are there new technical elements in the convergence proof compared to the convergence result in (Daulton, Balandat and Bakshy 2020)?
- Have the authors tried experiments with more than 4 objectives? Could the CBD-based algorithm handle several objectives?
- At what noise levels (e.g., in terms of std) does it start to make a difference whether we use EHVI or NEHVI? Are small noise levels still able to lead to visible errors when using EHVI?

**Time Spent Reviewing:**

6

---

> ### Author Response · Authors · 2021-08-10
> **To Reviewer 6KXs**
>
> > The scope of this work is a bit unclear. On the one hand, this work is about extending EHVI or qEHVI with noise. But on the other hand, it is about efficient computation of both the noiseless and noisy variants. The efficient batch selection algorithm could even be applicable in the original noiseless setting.
>
> We agree that the scope of the paper is more expansive than simply solving EHVI for the noisy case. More generally, our work enlarges the scope and performance of HVI-based methods. qNEHVI has equivalent or better sample complexity with respect to parallelism relative to any recent approach to MOBO in both the noisy and noiseless settings (see Appendix H.7 for noiseless results). qNEHVI has similar performance in the noiseless setting, but as stated earlier, makes large batch optimization feasible. Furthermore, we show that a one-sample approximation of qNEHVI (qNEHVI-1, described in the appendix, C.3), has extremely strong empirical performance with respect to both sample complexity and compute efficiency. qNEHVI can therefore be used as a general-purpose parallel MOBO algorithm, as well as a building block for other algorithms.
>
> Your feedback (and further consideration of the work since submission) has made it clear that our work would benefit from minor adjustments in framing and we will revise the narrative in our final manuscript. Perhaps a more apt title would have been “Scalable Parallel Multi-Objective Bayesian Optimization with Noisy Expected Hypervolume Improvement”.
>
>
> > The asymptotic complexities for CBDs or IEP depend exponentially on the number of objectives. This means that even with the efficient algorithm proposed in this work, multi-objective BO over several objectives could be prohibitively expensive.
>
> Indeed, while the runtime of the box decompositions involved in HVI do have an exponential dependence on the number of objectives, recent developments in HVI computation and our ability to cache these computations make it feasible to scale to >3 objectives; advances in box decomposition algorithms have yielded faster wall times and decompositions with fewer boxes (which speed up subsequent HVI computation, given the box decomposition). Note that many SoTA methods leverage HVI, including DGEMO and TSEMO. In response to your feedback, we have tested qNEHVI-1 with up to 5 objectives and have found superior results relative to scalarization-based methods such as qNParEGO. We only consider qNEHVI-1 because it only requires a single box decomposition per generated candidate (whereas qNEHVI requires one box decomposition per posterior sample). Below we report mean HV (and 2 SEMs) over 100 replications after 448 function evaluations on the DTLZ2 problem (d=6) with 5 objectives. These results use sequential optimization.
>
> **HV**
> * Sobol: 0.957 (+/-0.004)
> * qNParEGO: 1.144 (+/-0.007)
> * qNEHVI-1: 1.208 (+/-0.004)
>
> Below we report the mean generation wall times (and 2 SEMs) on a GPU (see paper for hardware details) over 100 replications:
>
> **Wall times (s)**
> * Sobol: 0.0 (+/-0.0)
> * qNParEGO: 11.374 (+/-0.971)
> * qNEHVI-1: 73.529 (+/-3.481)
>
> We are unaware of any work using exact HVI that has been shown to scale to 5 objectives, which further demonstrates the ways in which our work expands the scope of HVI-based methods. Approximate box decompositions can be used with CBD to scale to even more objectives.
>
> > On the other hand, for up to 4 objectives approaches based on trust regions (e.g., (Eriksson et al. 2019) or (Eriksson and Poloczek 2021)) might be able to handle high-D inputs and multiple constraints better.
>
> TuRBO and SCBO do not support multi-objective optimization, and their generalization to this case is non-trivial. For example, the selection of appropriate trust region centers is not obvious, and it is not obvious how various hyper-parameters like the number of trust regions affect the performance. In addition, it is not clear how well a trust-region-based BO method would work in the multi-objective setting because set of optimal solutions is typically not a singleton as is often the case in the single-objective setting, and therefore, local search may not lead to identifying diverse pareto sets. However, we expect that research into multi-objective trust region methods would likely benefit from our work, but this avenue is beyond the scope of this work.
>
> > Furthermore, Thompson sampling for evaluation of candidate points could lead to high efficiency. It is not clear how this line of work compares to such approaches in terms of solution quality or efficiency.
>
> We show that existing Thompson sampling (TS) methods including random scalarization-based TS-TCH and HVI-based TSEMO are outperformed by CBD-based approaches in terms of solution quality. Furthermore, the TS analog of qNEHVI (which we call qNEHVI-1) exhibits a similar sample complexity as qNEHVI, but is an order of magnitude more compute efficient (see C.3, H.1, and additional results throughout the appendix), making it a natural substitute for other TS-based methods in high-throughput settings.
>
>
> > At what noise levels (e.g., in terms of std) does it start to make a difference whether we use EHVI or NEHVI? Are small noise levels still able to lead to visible errors when using EHVI?
>
> First we refer the review to Appendix H.6 where we perform an analysis of noise levels (standard deviations) ranging from 5-30% of the range of each objective. To address the small noise level question, we have extended the analysis to include noise levels of 1-4% (on the same DTLZ2 problem as in H.6). Even under a 1% noise level, we observe a significant difference in performance between qEHVI and qNEHVI. This gap monotonically increases as a function of the noise level. Mean HV and 2 sems (across 100 replications) are provided below:
>
> **HV**
> * $\sigma=1$%: qNEHVI: 0.412 (+/-0.001), qEHVI: 0.409, (+/-0.0)
> * $\sigma=2$%: qNEHVI: 0.405, (+/-0.001, qEHVI: 0.4, (+/-0.001)
> * $\sigma=3$%: qNEHVI: 0.4, (+/-0.001), qEHVI: 0.389, (+/-0.001)
> * $\sigma=4$%: qNEHVI: 0.392, (+/-0.002), qEHVI: 0.38, (+/-0.001)
>
> We note that recent works that claim to account for noise (e.g. [21]) have not analyzed the effect of the noise level on optimization performance.

---

> > ### Comment · Reviewer_6KXs · 2021-08-17
> > **updated review**
> >
> > I thank the authors for the detailed response. My concerns were addressed, so I have updated my score. I suggest that (i) the authors clarify better the scope and improve the narrative of their work, since the scope is a bit confusing in the current version; (ii) they include the results with 5 objectives (including wall times) as well as the results from the low-noise regime.

---

### Author Response · Authors · 2021-08-10
**All Reviewers**

We thank the reviewers for their feedback and push for clarity on the contributions of the work. Our work is motivated by the problem of dealing with noisy observations and parallel evaluations—including ones involving low signal to noise ratios which are pervasive in real-world applications, such as those in chemistry, materials science, agriculture, internet experiments, and manufacturing.

One concern of the reviewers is about the novelty of the work. Indeed, the work here is cumulative: it builds off of a method, EHVI, which has been studied for over a decade, but has only until recently has shown to be a performant acquisition function by [7] through the use of a differentiable formulation of HV decompositions. Our work borrows ideas and proof techniques from [7], but we explicitly seek to address two critical shortcomings of all previous work on EHVI that are necessary for using EHVI in many practical scenarios: EHVI’s failure to explicitly account for observation noise in the acquisition function, and the infeasibility of using EHVI with large batch sizes. To address these shortcomings, we propose (i) a one-step Bayes optimal method that accounts for observation noise and (ii) CBD, a novel scalable approach to differentiable hypervolume computation whose runtime is polynomial with batch size for noisy and noiseless parallel EHVI. This is in contrast to the IEP approach of [7], which is restricted to very small small batch sizes: <15 in the 2-objective setting (as shown in Figure 2) and <10 in 3-objective and 4-objective settings (as shown in figure 6).

The work presented in our paper pushes the state-of-the-art for parallel noisy and noiseless MOBO, and as we discuss below, can also solve many of the desiderata described by the reviewers, including scaling to more than 3 objectives and high-throughput multi-objective optimization (for more details, see response to 6KXs). We hope that the responses below both clarify the contributions of this work to the community, and the thoughtful technical questions that were raised.

---

### Decision · Program_Chairs · 2021-09-27

**Decision:**

Accept (Poster)

**Comment:**

We thank the authors for the additional clarifications provided in their rebuttal. All reviewers agreed that this work is of practical importance and made solid contributions by building on top of prior work, addressing an issue overlooked in the literature (namely the uncertainty residing in the Pareto front). The authors conducted an extensive set of experiments. They pre-dominantly focus on BO methods; including multi-objective approaches not relying on the BO paradigm could strengthen the paper. It was is also surprising the author missed to include the following reference/baseline:

Daniel Golovin and Qiuyi Zhang. Random Hypervolume Scalarizations for Provable Multi-Objective Black Box Optimization. 2020.